Letter

# Long COVID manifests with T cell dysregulation, inflammation and an uncoordinated adaptive immune response to SARS-CoV-2

Kailin Yin[1,2,9], Michael J. Peluso [3,9], Xiaoyu Luo[1,2], Reuben Thomas[1], Min-Gyoung Shin[1], Jason Neidleman[1,2], Alicer Andrew [1,2], Kyrlia C. Young[1,2], Tongcui Ma[1,2], Rebecca Hoh[3], Khamal Anglin[3], Beatrice Huang [3], Urania Argueta[3], Monica Lopez[3], Daisy Valdivieso [3], Kofi Asare[3], Tyler-Marie Deveau[4], Sadie E. Munter[4], Rania Ibrahim[3], Ludger Ständker[5], Scott Lu[6], Sarah A. Goldberg [6], Sulggi A. Lee[7], Kara L. Lynch[8], J. Daniel Kelly [6], Jeffrey N. Martin[6], Jan Münch [5], Steven G. Deeks[3], Timothy J. Henrich [4] ✉ & Nadia R. Roan [1,2] ✉

Long COVID (LC) occurs after at least 10% of severe acute respiratory syndrome coronavirus 2 (SARS-CoV-2) infections, yet its etiology remains poorly understood. We used 'omic' assays and serology to deeply characterize the global and SARS-CoV-2-specific immunity in the blood of individuals with clear LC and non-LC clinical trajectories, 8 months postinfection. We found that LC individuals exhibited systemic inflammation and immune dysregulation. This was evidenced by global differences in T cell subset distribution implying ongoing immune responses, as well as by sex-specific perturbations in cytolytic subsets. LC individuals displayed increased frequencies of CD4+ T cells poised to migrate to inflamed tissues and exhausted SARS-CoV-2-specific CD8+ T cells, higher levels of SARS-CoV-2 antibodies and a mis-coordination between their SARS-CoV-2-specific T and B cell responses. Our analysis suggested an improper crosstalk between the cellular and humoral adaptive immunity in LC, which can lead to immune dysregulation, inflammation and clinical symptoms associated with this debilitating condition.

Intense efforts are underway to determine the pathophysiology of long COVID (LC), a set of conditions characterized by immune perturbations[1]. T cells have important roles in severe acute respiratory syndrome coronavirus 2 (SARS-CoV-2) immunity and pathogenesis[2–6], yet relatively little is known about their role in LC. Here we used CyTOF, serology, RNA sequencing (RNA-seq), single-cell RNA-seq (scRNA-seq) and plasma proteomics to obtain a deep phenotypic characterization of T cells in a well-matched set of LC and fully recovered (R) individuals to identify unique immune features associated with LC that inform on the mechanistic underpinnings of this condition.

We leveraged a well-characterized cohort (Long-term Impact of Infection with Novel Coronavirus (LIINC)[7]; Supplementary Tables 1–3) to analyze the blood from 27 LC and 16 R individuals, obtained 8 months postinfection (Fig. 1a) before any SARS-CoV-2 vaccination

or reinfection. LC individuals, who consistently exhibited LC symptoms such as fatigue, 'brain fog' and sleep disturbance over 8 months, were 63% female and included 26% previously hospitalized for COVID-19 (Extended Data Fig. 1a–c and Supplementary Tables 1–3). Comorbidities such as hypertension were more common in LC individuals (6/27 for LC and 1/16 for R), who also had higher body mass index (BMI; Extended Data Fig. 1d,e). A CyTOF panel designed to interrogate the differentiation and/or activation states, effector functions and homing properties of T cells (Extended Data Fig. 1f and Supplementary Table 4) was applied to cryopreserved blood at baseline (post-thaw) or following stimulation with SARS-CoV-2 spike and T-scan peptides (Methods) to identify SARS-CoV-2-specific T cells through intracellular cytokine staining.

Both baseline and poststimulation datasets were gated on $CD3^+$ events to identify T cells (Extended Data Fig. 1g,h), which were assessed for the expression of a panel of effector molecules, consisting of the cytokines interferon-γ (IFN-γ), tumor necrosis factor (TNF), interleukin (IL)-2, IL-4, IL-6, IL-17 and CCL4, and the cytolytic markers granzyme B and perforin (Extended Data Fig. 2a,b). Based on criteria comparing stimulated versus baseline samples (Methods), IFN-γ, TNF and/or IL-2 positivity identified SARS-CoV-2-specific $CD4^+$ T cells, whereas IFN-γ, TNF and/or CCL4 positivity identified SARS-CoV-2-specific $CD8^+$ T cells (Fig. 1b,c and Extended Data Fig. 2a,b). Using Boolean gating, we did not find significant differences between the frequencies of total SARS-CoV-2-specific $CD4^+$ or $CD8^+$ T cells (Fig. 1d), or those producing individual effector cytokines IFN-γ, TNF, IL-2 or CCL4 (Extended Data Fig. 2c,d) between LC and R individuals. Furthermore, the distribution of polyfunctional (producing at least two cytokines) SARS-CoV-2-specific $CD4^+$ and $CD8^+$ T cells was similar between LC and R individuals (Fig. 1e, f). However, SARS-CoV-2-specific IFN-γ$^+$TNF$^+$IL-2$^+$CD4$^+$ T cells and SARS-CoV-2-specific IFN-γ$^+$TNF$^+$CCL4$^+$CD8$^+$ T cells were more abundant, without reaching statistical significance, in R individuals (Fig. 1e,f). IL-6 expression in $CD4^+$ T cells was induced exclusively in those with LC, albeit only in a small subset (14%; Extended Data Fig. 2e,f).

$CD45RA^+CD45RO^-CCR7^+CD95^-$ naïve T ($T_N$) cells, $CD45RA^+CD45RO^-CCR7^+CD95^+$ stem cell memory T ($T_{SCM}$) cells, $CD45RA^-CD45RO^+CCR7^+CD27^+$ central memory T ($T_{CM}$) cells, $CD45RA^-CD45RO^+CCR7^-CD27^-$ effector memory T ($T_{EM}$) cells, $CD45RA^-CD45RO^+CCR7^-CD27^+$ transitional memory T ($T_{TM}$) cells and $CD45RA^+CD45RO^-CCR7^-$ effector memory RA T ($T_{EMRA}$) cells were identified in both $CD4^+$ and $CD8^+$ T cell compartments through manual gating (Extended Data Fig. 1i,j). In addition, $CD45RA^-CD45RO^+CD127^-CD25^+$ T regulatory ($T_{reg}$) cells and $CD45RA^-CD45RO^+PD1^+CXCR5^+$ peripheral T follicular helper ($pT_{FH}$) cells were identified in the $CD4^+$ T cell compartment, and we additionally established a more stringent $CD45RA-CD45RO^+PD1^{hi}CXCR5^{hi}$ $T_{FH}$ cell gate (Extended Data Fig. 1i). Total $CD4^+$ $T_{CM}$, $pT_{FH}$, $T_{FH}$ and $T_{reg}$ cell subsets were more frequent in LC compared to R individuals with no difference between LC and R in the other total $CD4^+$ T cell subsets analyzed (Fig. 1g), while none of these subsets were significantly different between LC and R when examining SARS-CoV-2-specific $CD4^+$ T cells (Fig. 1g,h). All analyzed subsets of total or SARS-CoV-2-specific $CD8^+$ T cells were statistically similar between LC and R individuals (Extended Data Fig. 3).

Analysis of expression levels of all CyTOF markers in total or SARS-CoV-2-specific $CD4^+$ or $CD8^+$ T cells found that no markers were significantly differentially expressed between LC and R individuals (Extended Data Figs. 4 and 5). We found no significant differences in the percentages of $CD4^+$ or $CD8^+$ T cells expressing the acute activation markers CD38, HLA-DR and/or Ki67 in LC compared to R individuals (Extended Data Fig. 6). Clustering analyses (Methods) revealed $CD4^+$ T cells fell into six clusters (A1–A6) and $CD8^+$ T cells into five clusters (B1–B5) clusters that did not differ significantly between LC and R individuals (Extended Data Fig. 7a,e). However, cluster A1 was significantly underrepresented in LC compared to R females, but not in males, while cluster A4 was significantly underrepresented in LC

compared to R males, but not in females (Extended Data Fig. 7b). Cluster A1 was composed of $CD45RO^{lo}CD45RA^{hi}CD4^+$ $T_N$ cells and expressed low levels of activation markers (HLA-DR and Ox40) and inflammatory tissue-homing receptors (CD29 and CXCR4), as well as high levels of lymph node homing receptors (CD62L and CCR7; Extended Data Fig. 7c). Cluster A4 was composed of terminally differentiated $CD45RO^{hi}CD27^{lo}CD57^{hi}CD4^+$ $T_{EM}$ cells and expressed high levels of receptors associated with homing to inflamed tissues (CD29, CXCR4 and CCR5) but not to lymph nodes (CD62L and CCR7). They also had high expression of cytolytic markers perforin and granzyme B (Extended Data Fig. 7d). Among $CD8^+$ T cells, cluster B1 was significantly underrepresented in LC females, while cluster B2 was significantly overrepresented in LC females, compared to their R female counterparts, with no differences observed in males (Extended Data Fig. 7f). Cluster B1 comprised $CD8^+$ T cells expressing markers of cluster A1 ($CD45RO^{lo}CD45RA^{hi}HLA-DR^{lo}Ox40^{lo}CD29^{lo}CXCR4^{lo}CD62L^{hi}CCR7^{hi}$), whereas cluster B2 comprised $CD8^+$ T cells expressing markers of cluster A4 ($CD27^{lo}CD57^{hi}CD29^{hi}CXCR4^{hi}CCR5^{hi}CD62L^{lo}CCR7^{lo}$). These observations suggested that females with LC had relatively low frequencies of resting $CD4^+$ and $CD8^+$ $T_N$ cells, which expressed low levels of inflammatory tissue-homing receptors, and high frequencies of terminally differentiated $CD4^+$ and $CD8^+$ $T_{EM}$ cells, which expressed inflammatory tissue-homing receptors and cytolytic markers.

The t-distributed stochastic neighbor embedding (t-SNE) visualization of SARS-CoV-2-specific $CD4^+$ T cells indicated that those from LC and R individuals tended to concentrate in different areas (Fig. 2a). The tissue-homing receptors CXCR4, CXCR5 and CCR6 were expressed higher on SARS-CoV-2-specific $CD4^+$ T cells from LC as compared to R individuals (Fig. 2b). Manual gating showed that the percentages of SARS-CoV-2-specific $CXCR4^+CXCR5^+CD4^+$ T cells and $CXCR5^+CCR6^+CD4^+$ T cells were significantly increased, and $CXCR4^+CCR6^+CD4^+$ T cells showed a trend toward higher percentages, in LC compared to R individuals (Fig. 2c). Higher percentages of total $CXCR4^+CXCR5^+CD4^+$ T cells and $CXCR5^+CCR6^+CD4^+$ T cells were found in LC compared to R as well (Fig. 2d). Flow cytometric analysis of the same LC and R specimens found statistically significant elevated frequencies of $CXCR4^+CXCR5^+CD4^+$, $CXCR5^+CCR6^+CD4^+$ and $CXCR4^+CCR6^+CD4^+$ T cells in LC compared to R (Extended Data Fig. 8a–c). Expression of CXCR5 is common among the $CXCR4^+CXCR5^+CD4^+$ T cell, $CXCR5^+CCR6^+CD4^+$ T cell and $pT_{FH}$ cell subsets, and we observed significant positive associations between the percentages of $pT_{FH}$ cells and other $CXCR5^+CD4^+$ T cells, particularly in the LC group (Fig. 2e,f).

SARS-CoV-2-specific $CD8^+$ T cells were also globally different between LC and R (Fig. 3a), and those from the individuals with LC preferentially expressed the checkpoint markers PD1 and CTLA4, but not TIGIT (Fig. 3b). Consistently, SARS-CoV-2-specific $PD1^+CTLA4^+CD8^+$ T cells were significantly elevated in LC compared to R individuals, while SARS-CoV-2-specific $TIGIT^+CTLA4^+CD8^+$ or $PD1^+TIGIT^+CD8^+$ T cells were not (Fig. 3c). However, the frequencies of total $PD1^+CTLA4^+CD8^+$ T cells were similar in the LC and R groups (Fig. 3d and Extended Data Fig. 8d).

Serological analysis indicated significantly higher (2.3×) total receptor binding domain (RBD)-specific antibody titers in LC as compared to R individuals (Fig. 4a). LC individuals with the highest frequencies of SARS-CoV-2-specific $PD1^+CTLA4^+CD8^+$ T cells had near undetectable antibody levels (Fig. 4b). LC individuals with the highest frequencies of SARS-CoV-2-specific $PD1^+CTLA4^+CD8^+$ T cells had the lowest frequencies of SARS-CoV-2-specific $CD4^+$ $T_{reg}$ cells, and the frequencies of these two subsets of cells negatively correlated in LC, but not R individuals (Fig. 4b). A significant positive correlation between RBD-specific titers and total SARS-CoV-2-specific total $CD4^+$ and $CD8^+$ T cell frequencies was detected in R but not LC individuals (Fig. 4c). The frequencies of SARS-CoV-2-specific $pT_{FH}$ cells also correlated positively with RBD-specific antibody titers in R but not LC individuals (Fig. 4c), suggesting a mis-coordinated humoral and cell-mediated

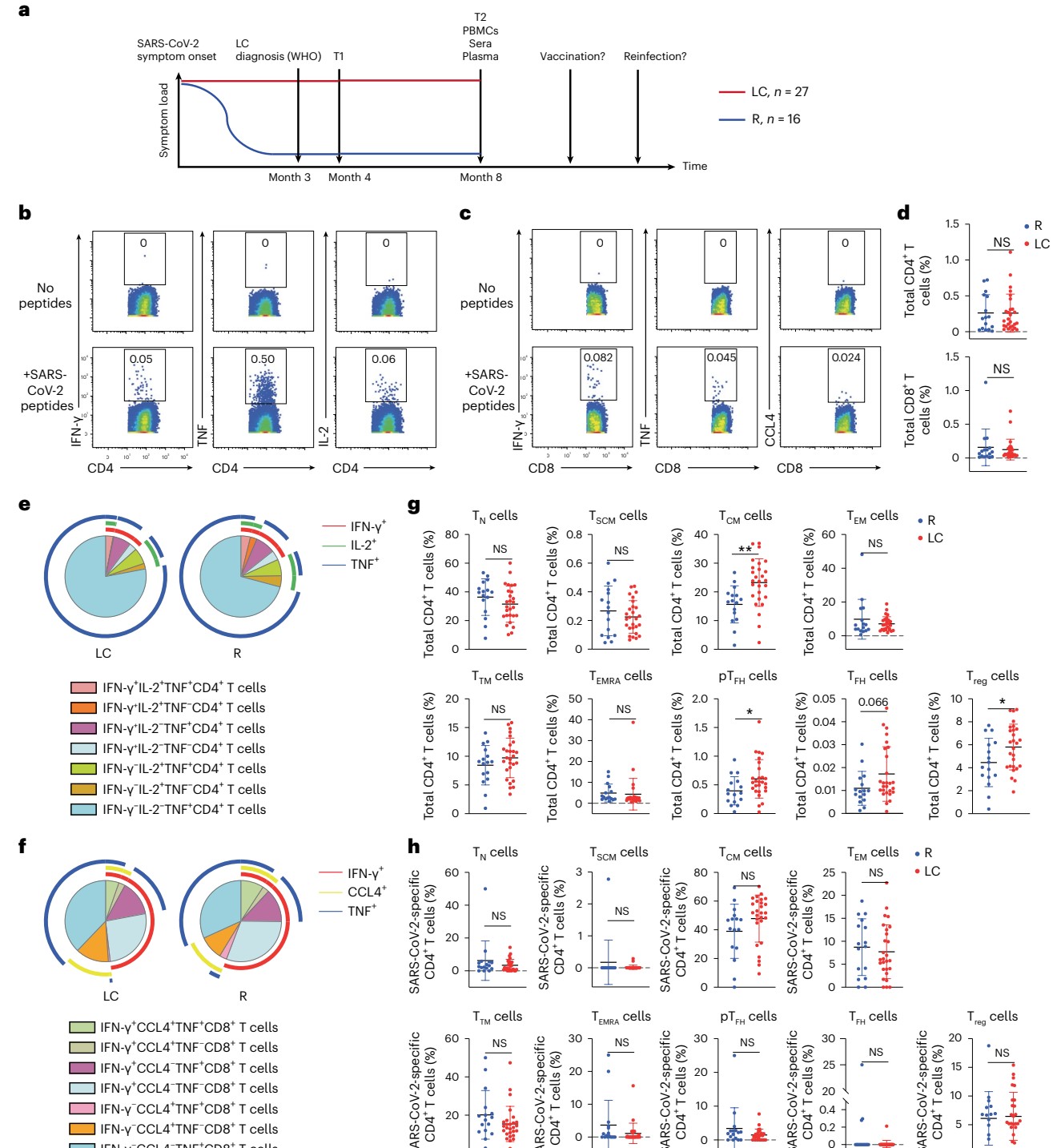

**Fig. 1 | CD4⁺ T cell phenotypes are perturbed in individuals with LC.**
**a**, Strategy of biospecimen selection in individuals who resolved symptoms (R, $n = 16$) or who continuously experienced symptoms at month 4 (T1) and month 8 (T2) postinitial SARS-CoV-2 infection (LC, $n = 27$). The WHO definition for LC is persistent symptoms for 3 months or more after infection[14]. All analyzed PBMCs, sera and plasma were from 8 months postinfection, a timepoint when none of the participants had been vaccinated nor re-infected. **b,c**, Expression of IFN-γ, TNF or IL-2 in CD4⁺ T cells (**b**) or IFN-γ, TNF or CCL4 in CD8⁺ T cells (**c**) stimulated (bottom) or not (top) with SARS-CoV-2 spike and T-scan peptides (Methods). **d**, Frequency of SARS-CoV-2-specific CD4⁺ or SARS-CoV-2-specific CD8⁺ T cells in LC and R individuals (two-sided Student's $t$ tests). **e,f**, Frequency of monofunctional or polyfunctional SARS-CoV-2-specific CD4⁺ (**e**) or SARS-CoV-2-specific CD8⁺ (**f**) T cells in LC versus R individuals. Polyfunctional

cells co-express at least two of the cytokines IFN-γ, IL-2 and TNF (**e**) or IFN-γ, IL-2 and CCL4 (**f**). **g**, Frequencies of CD45RA⁺CD45RO⁻CCR7⁺CD95⁻ $T_N$ cells, CD45RA⁺CD45RO⁻CCR7⁺CD95⁺ $T_{SCM}$ cells, CD45RA⁻CD45RO⁺CCR7⁺CD27⁺ $T_{CM}$ cells, CD45RA⁻CD45RO⁺CCR7⁻CD27⁻ $T_{EM}$ cells, CD45RA⁻CD45RO⁺CCR7⁻CD27⁺ $T_{TM}$ cells, CD45RA⁺CD45RO⁻CCR7⁻ $T_{EMRA}$ cells, CD45RA⁻CD45RO⁺PD1⁺CXCR5⁺ peripheral p$T_{FH}$ cells, CD45RA⁻CD45RO⁺PD1^high^CXCR5^high^ $T_{FH}$ cells and CD45RA⁻CD45RO⁺CD127⁻CD25⁺ $T_{reg}$ cells among total CD4⁺ T cells from LC and R individuals. **$P < 0.01, *$P < 0.05$ (two-sided Student's $t$ test). **h**, Frequencies of $T_N$ cells, $T_{SCM}$ cells, $T_{CM}$ cells, $T_{EM}$ cells, $T_{TM}$ cells, $T_{EMRA}$ cells, p$T_{FH}$ cells, $T_{FH}$ cells and $T_{reg}$ cells among SARS-CoV-2-specific CD4⁺ T cells from LC and R individuals. Horizontal bars indicate mean, error bars indicate s.d., and dots represent individuals, with $n = 27$ LC and $n = 16$ R (**d**, **g** and **h**). NS, not significant; WHO, World Health Organization.

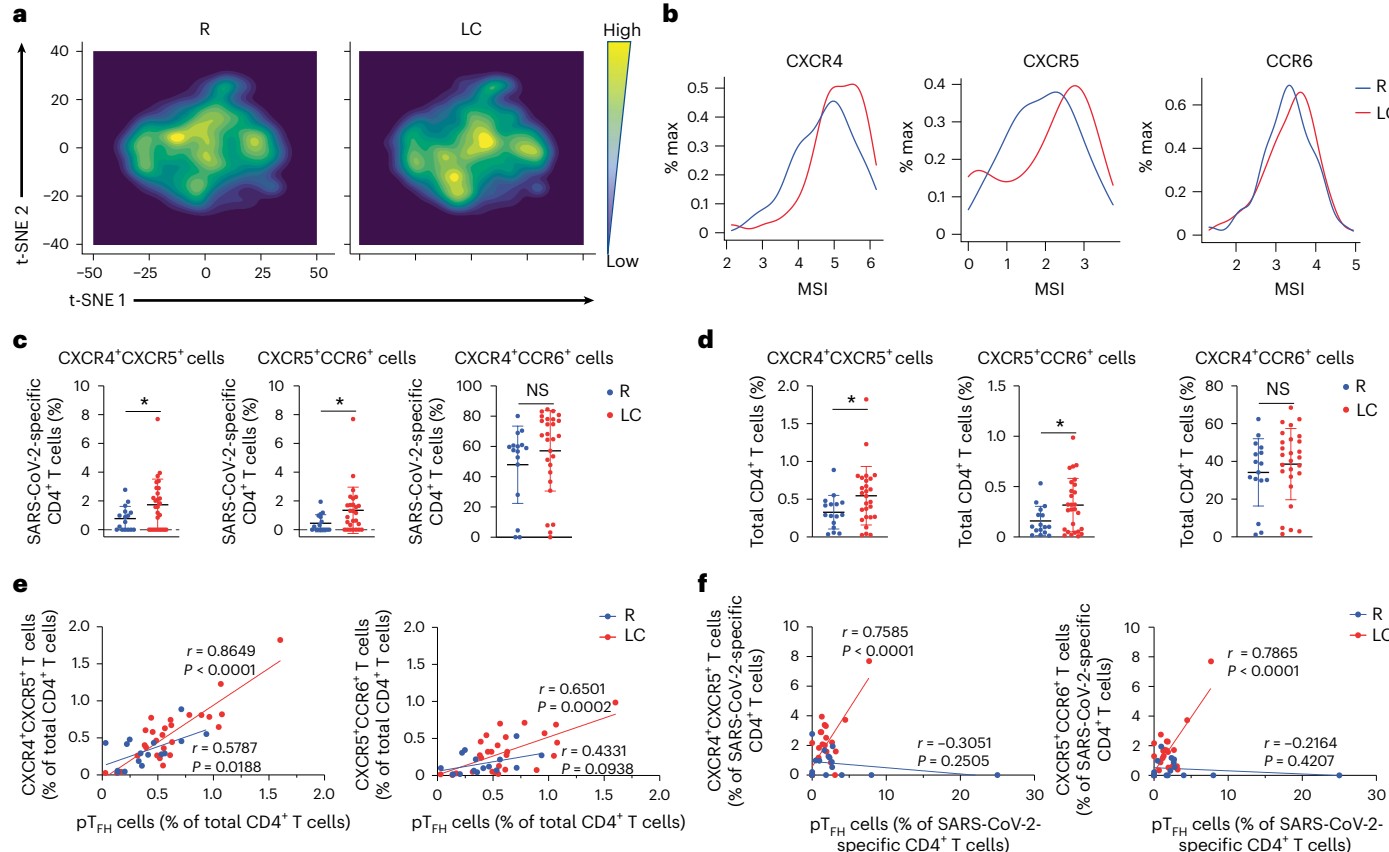

**Fig. 2 | SARS-CoV-2-specific CD4⁺ T cells from individuals with LC preferentially express homing receptors associated with migration to inflamed tissues. a**, t-SNE contour depiction of SARS-CoV-2-specific CD4⁺ T cells from LC and R individuals. **b**, Expression of CXCR4, CXCR5 and CCR6 in SARS-CoV-2-specific CD4⁺ T cells from LC and R individuals. MSI corresponds to the mean signal intensity of the indicated markers' expression level, reported as arcsinh-transformed CyTOF data. **c,d**, Percentages of CXCR4⁺CXCR5⁺CD4⁺,

CXCR5⁺CCR6⁺CD4⁺ and CXCR4⁺CCR6⁺CD4⁺ SARS-CoV-2-specific (**c**) and total (**d**) CD4⁺ T cells in LC and R individuals. *$P < 0.05$ (two-sided Student's *t* test). **e,f**, Associations of percentages of total (**e**) or SARS-CoV-2-specific (**f**) CXCR4⁺CXCR5⁺CD4⁺ and CXCR5⁺CCR6⁺CD4⁺ T cells with percentages of pT$_{FH}$ cells in LC and R individuals. Data were analyzed by Pearson correlation coefficient and two-tailed unpaired *t* tests. Horizontal bars indicate mean, error bars indicate s.d. and dots represent individuals, with $n = 27$ LC and $n = 16$ R (**c,d**).

response, previously implicated in severe COVID-19 (ref. 8), may also be a hallmark of LC.

Bulk RNA-seq identified only two genes, *OR7D2* and *ALAS2*, that were significantly differentially expressed between LC and R. *OR7D2* encodes a G-protein-coupled receptor that is activated by odorant molecules, whereas *ALAS2* encodes an enzyme that catalyzes the first step in heme synthesis to generate δ-aminolevulinic acid from succinyl-CoA and glycine. Both *OR7D2* and *ALAS2* were overexpressed in LC individuals although not necessarily together, as the four individuals with the highest *OR7D2* expression in peripheral blood mononuclear cells (PBMCs) did not have the highest *ALAS2* expression (Fig. 5a). Supervised clustering found upregulation of a module of genes that regulate heme synthesis and carbon dioxide transport (*ALAS2, HBB, CA1, HBA1, SLC4A1, HBD* and *HBA2*) and the downregulation of a module consisting of immunoglobin kappa, lambda and heavy chain genes in LC compared to R individuals (Fig. 5b,c), suggesting the involvement of heme biosynthesis and immune dysregulation in LC.

To gain a more granular view of the transcriptome, we selected a subset of the specimens analyzed by bulk RNA-seq for repeat analysis by scRNA-seq. We limited these studies to females because individuals with high levels of *OR7D2* or *ALAS2* were mostly female (the top five *OR7D2* expressors were female, as were five of the top six *ALAS2* expressors). For comparison, we included four randomly selected females from the R specimens. Integration of data from all 12 samples identified 11 clusters of cells and revealed that the granulocyte cluster was significantly less

abundant ($P = 0.006$) and the platelet cluster more abundant ($P = 0.01$) in LC compared to R individuals, while the other clusters (CD4⁺ T cells, CD8⁺ T cells, CTLs, B cells, monocytes and NKT/NK/MAIT/γδ T cells) did not differ between the groups (Fig. 5d). Visualization based on LC versus R status, or based on *OR7D2*ʰⁱ LC versus *ALAS2*ʰⁱ LC, did not reveal profound differences (Extended Data Fig. 9a,b). Among all cells, *OR7D2* expression was highest in cells of the *OR7D2*ʰⁱ LC group and *ALAS2* was highest in cells of the *ALAS2*ʰⁱ LC group, and all clusters except granulocytes and platelets expressed *OR7D2* and *ALAS2* (Extended Data Fig. 9c–e).

Interrogation of cluster-specific gene expression identified three additional genes (*THEMIS, NUDT2* and *PPIE*) that were differentially expressed ($P < 0.05$) in LC individuals, two within CD8⁺ T cell cluster 1 and one within monocyte cluster 3 (Fig. 5e). Using a less stringent cutoff ($P < 0.1$), we found 16 differentially expressed genes (DEGs) within CD8⁺ T cell cluster 1 (for example, *THEMIS, HMGB2* and *TNFRSF18*), monocyte cluster 3 (*PPIE*) and CD4⁺ T cell cluster 7 (for example, *CAST* and *APBA2*; Fig. 5f and Supplementary Table 5). Gene Ontology (GO) pathway analysis found significant ($P < 0.05$) differences between LC and R individuals within monocyte cluster 3, in pathways associated with transcriptional regulation and splicing, protein regulation and neutrophil degranulation (Supplementary Table 6). Trends ($P < 0.1$) were observed for pathways associated with apoptosis and metabolism and/or oxidative stress in CD8⁺ T cell cluster 1 (Supplementary Table 7). CXCR4, CXCR5 and CCR6 were upregulated in CD4⁺ T cell clusters 0 and 7 from LC compared to their counterpart clusters in R (Extended

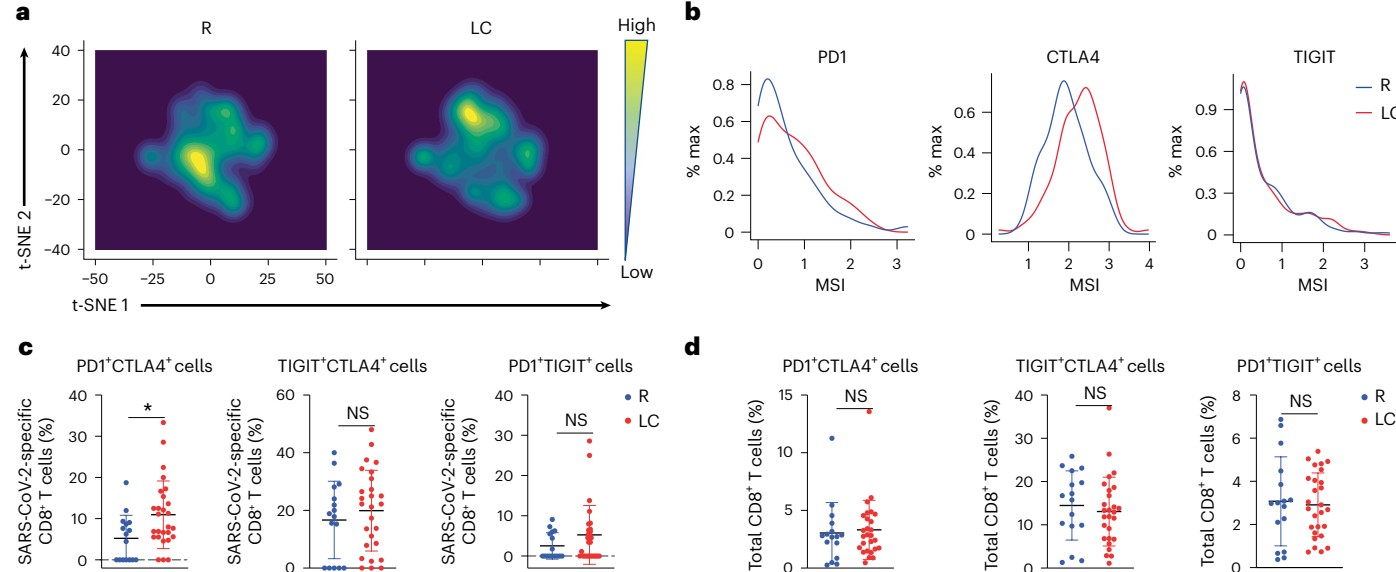

**Fig. 3 | SARS-CoV-2-specific CD8⁺ T cells from individuals with LC preferentially express the exhaustion markers PD1 and CTLA4. a**, t-SNE contour depiction of SARS-CoV-2-specific CD8⁺ T cells from LC and R individuals. **b**, Expression of PD1, CTLA4 and TIGIT on SARS-CoV-2-specific CD8⁺ T cells from LC and R individuals.

**c,d**, Percentages of PD1⁺CTLA4⁺CD8⁺, TIGIT⁺CTLA4⁺CD8⁺ and PD1⁺TIGIT⁺CD8⁺ SARS-CoV-2-specific (**c**) and total (**d**) CD8⁺ T cells in LC and R individuals. *$P < 0.05$ (two-sided Student's $t$ test). Horizontal bars indicate mean, error bars indicate s.d. and dots represent individuals, with $n = 27$ LC and $n = 16$ R (**c,d**).

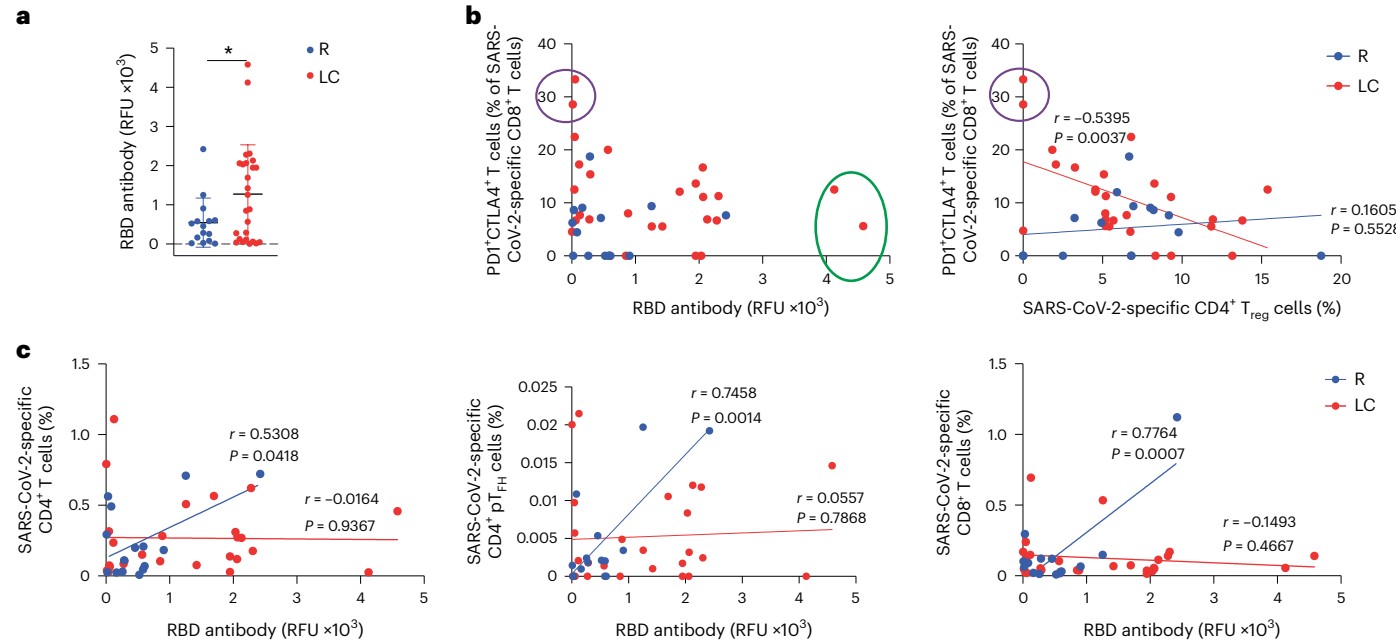

**Fig. 4 | Humoral and cellular immunity are discoordinated in individuals with LC. a**, Total SARS-CoV-2 RBD-specific antibody levels in LC and R individuals. *$P < 0.05$ (two-sided Student's $t$ test). Horizontal bars indicate mean, error bars indicate s.d. and dots represent individuals. LC ($n = 26$), R ($n = 15$). **b**, Plot depicting the percentage of PD1⁺CTLA4⁺ cells among SARS-CoV-2-specific CD8⁺ T cells and RBD antibody levels in LC and R individuals. Individuals with the highest humoral response are circled in green, and those with the highest percentages of PD1⁺CTLA4⁺ SARS-CoV-2-specific CD8⁺ T cells are circled in purple (left). **c**, Plot depicting the association between RBD antibody levels and the percentages of SARS-CoV-2-specific CD4⁺ T cells, SARS-CoV-2-specific CD4⁺ pT_FH cells (middle) and SARS-CoV-2-specific CD8⁺ T cells (right) in LC and R individuals. Data were analyzed by Pearson correlation coefficient and two-tailed unpaired $t$ tests.

Data Fig. 9f). Comparison of *OR7D2*^hi LC versus R revealed 35 DEGs in the *OR7D2*^hi LC group (Extended Data Fig. 9g and Supplementary Table 8) including upregulation of the histone family genes *HIST1H2AM*, *HIST2H2AC* and *HIST1H1E*, while comparison of *ALAS2*^hi LC versus R revealed 14 DEGs including upregulation of *THEMIS* and downregulation of *BACH2* ($P < 0.05$; Extended Data Fig. 9h and Supplementary

Table 9). GO pathways associated with the *OR7D2*^hi LC DEGs included lipid transport and stress responses in CD4⁺ T cell cluster 7, RNA splicing in CD8⁺ T cell cluster 5 and immunoglobulin (Ig) production in B cell cluster 8 (Supplementary Table 10), while those associated with the *ALAS2*^hi LC DEGs included apoptosis and oxidative stress responses in CD8⁺ T cell cluster 1 (Supplementary Table 11).

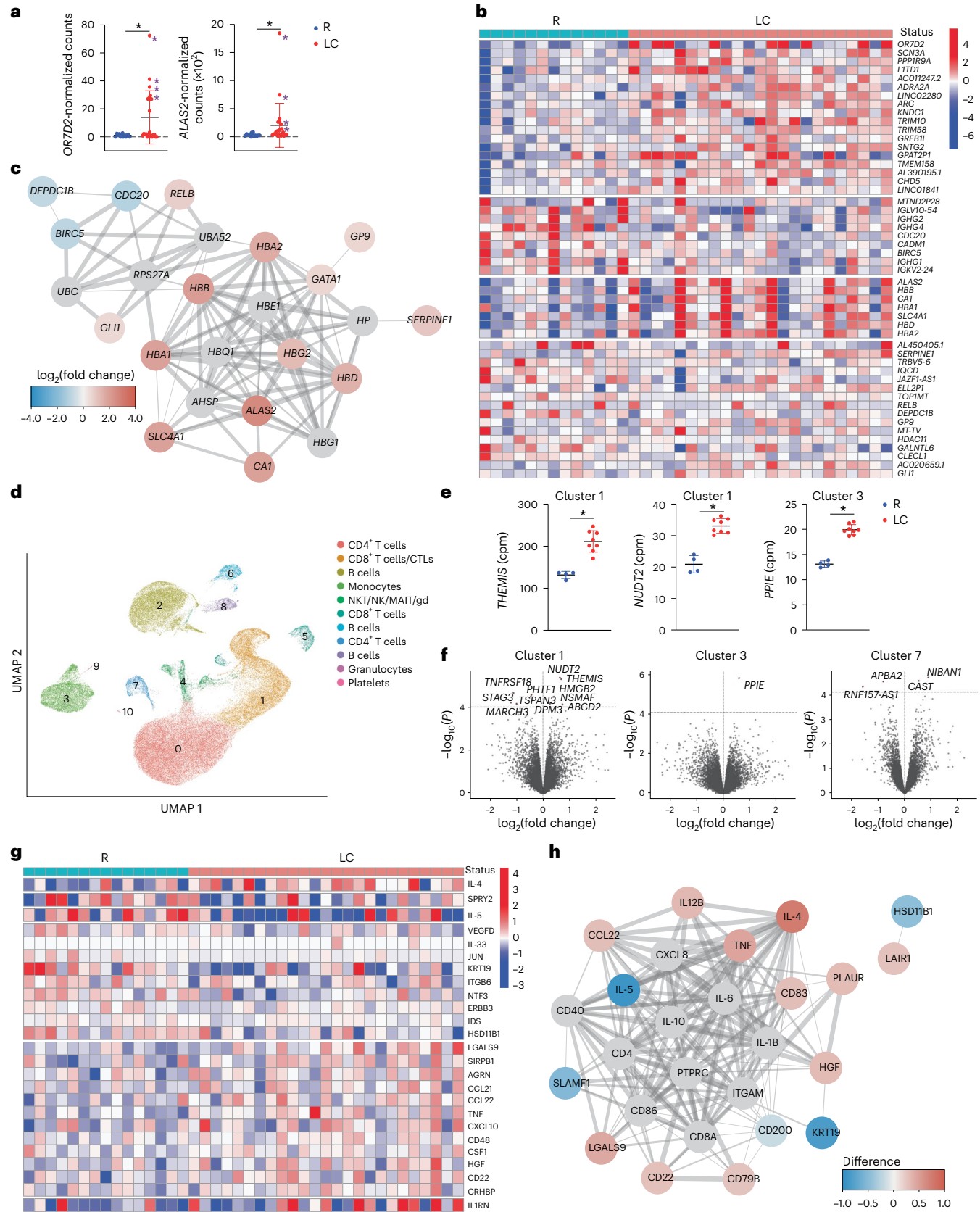

Olink proteomics indicated elevated expression of proteins associated with inflammation (LGALS9, CCL21, CCL22, TNF, CXCL10 and CD48) and immune regulation (IL1RN and CD22) in LC compared to R individuals (Fig. 5g). LC individuals had elevated expression of IL-4 and decreased expression of IL-5 compared to R individuals (Fig. 5g,h), although both cytokines are associated with T helper 2 (T_H2) cell responses. CCL22, a ligand for the T_H2 cell marker CCR4, was expressed at elevated levels in LC compared to R individuals (Fig. 5h).

**Fig. 5 | Global changes in gene and gene product expression in the blood of individuals with LC. a**, Relative expression of *OR7D2* and *ALAS2* as determined by bulk RNA-seq analysis of whole blood from LC versus R individuals. *$P < 0.05$ (two-sided Wald test, Benjamini–Hochberg correction). Purple asterisks identify the female donors selected for scRNA-seq analyses. Horizontal bars indicate mean, error bars indicate s.d. and dots represent individuals. LC ($n = 23$) and R ($n = 13$). **b**, Heatmap of the top 50 DEGs in LC versus R individuals based on clustering analysis of bulk RNA-seq data. Genes are grouped into k clusters based on similarity. **c**, Network mapping of DEGs from bulk RNA-seq analysis. Each node corresponds to a gene; colors of nodes indicate the extent of change; red indicates upregulation and blue indicates downregulation in LC compared to R. Edges depict the functional relevance between pairs of genes, where thickness corresponds to confidence of evidence. **d**, UMAP of clusters of all LC and R PBMCs analyzed by scRNA-seq. LC ($n = 8$) and R ($n = 4$). **e**, Relative expression of *THEMIS* and *NUDT2* in CD8+ T cell cluster 1 and *PPIE* in monocyte cluster 3 in LC versus R individuals as determined by scRNA-seq analysis. *$P < 0.05$ (two-sided empirical Bayes quasi-likelihood $F$ tests, with Benjamini–Hochberg correction). Horizontal bars indicate mean, error bars indicate s.d. and dots represent individuals. LC ($n = 8$) and R ($n = 4$). **f**, Volcano plots depicting DEGs in LC versus R individuals in scRNA-seq-defined clusters. DEGs with $P < 0.1$ (two-sided empirical Bayes quasi-likelihood $F$ tests, Benjamini–Hochberg correction) are labeled. The *x* axes represent the $\log_2$(fold change) of the mean expression of each gene between the comparison groups, and the *y* axes represent the raw $-\log_{10}$(*P* values). Dashed horizontal lines delineate thresholds corresponding to Benjamini–Hochberg adjusted *P* values of <0.1. **g**, Clustered heatmap of the top 25 differentially expressed proteins from Olink analysis performed on plasma of LC and R individuals with markers grouped into *k*-means clusters based on similarity. LC ($n = 25$) and R ($n = 15$). **h**, Network mapping of related differentially expressed proteins as detected by Olink. Graph representations as in **c**.

---

IL-4, but not IL-5 or CCL22, significantly positively associated with the percentages of total CXCR4+CXCR5+CD4+ and CXCR5+CCR6+CD4+ T cells in LC individuals (Extended Data Fig. 8e), suggesting an elevated, yet mis-coordinated, $T_H2$ cell response during LC.

In summary, using multiple 'omics' analytical approaches, we found that LC individuals exhibited phenotypic perturbations in both total and SARS-CoV-2-specific CD4+ and CD8+ T cells and changes in gene expression among CD4+ T cells, CD8+ T cells, monocytes and B cells. We found higher proportions of CD4+ $T_{CM}$ cells, $T_{FH}$ cells and $T_{reg}$ cells in LC compared to R individuals. SARS-CoV-2-specific CD8+ T cells, but not total CD8+ T cells, more frequently expressed the exhaustion markers PD1 and CTLA4, consistent with ongoing stimulation by viral antigens. Further supporting a potential persistent reservoir was our observation of higher SARS-CoV-2 antibody levels in LC individuals, consistent with reports of higher spike-specific IgG in LC compared to R individuals[9]. CyTOF, flow cytometry and scRNA-seq indicated that CD4+ T cells from LC individuals preferentially expressed CXCR4, CXCR5 and CCR6. CXCR4 expression is elevated on bystander CD4+ and CD8+ T cells in fatal COVID-19 (ref. 4) and on pulmonary CD4+ T cells, B cells, macrophages and granulocytes in the context of LC following SARS-CoV-2 infection of mice[10]. Although fully recovered individuals exhibited coordinated humoral and cellular immune responses to SARS-CoV-2, this coordination was lost in LC individuals, consistent with observations that about half of individuals with LC with no detectable SARS-CoV-2 antibodies have detectable SARS-CoV-2-specific T cell responses[11]. How the humoral response becomes divorced from the cellular response is unclear, but could involve a misalignment between IL-4 and IL-5 production by $T_H2$ cells, as indicated by our Olink analysis.

Our study has limitations. First, the cohort analyzed included only 43 participants; however, the rigor with which participants were characterized mitigates the limitations of the small sample size. Some findings were driven by small subsets of LC individuals, which is consistent with the notion of LC being a heterogeneous disease, and will require validation in larger cohorts. Second, due to limited channels available for CyTOF, we did not examine additional markers that would have been of interest such as the exhaustion marker thymocyte selection-associated high mobility group protein (TOX)[12], the activation marker CD40L and the proliferation marker 5-Iodo-2'-deoxyuridine (IdU)[13]. Third, the changes we saw in the blood subsets could reflect migration to tissues. Finally, our study was for the most part descriptive. However, for new and poorly understood diseases, in-depth 'omics'-based characterization of a well-annotated cohort is the critical first step for better understanding the condition's etiology and mechanistic underpinnings.

## Online content

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

[1]Gladstone Institutes, University of California, San Francisco, San Francisco, CA, USA. [2]Department of Urology, University of California, San Francisco, San Francisco, CA, USA. [3]Division of HIV, Infectious Diseases, and Global Medicine, University of California, San Francisco, San Francisco, CA, USA. [4]Division of Experimental Medicine, University of California, San Francisco, San Francisco, CA, USA. [5]Core Facility Functional Peptidomics, Ulm University Medical Center, Ulm, Germany. [6]Department of Epidemiology and Biostatistics, University of California, San Francisco, San Francisco, CA, USA. [7]Zuckerberg San Francisco General Hospital and the University of California, San Francisco, San Francisco, CA, USA. [8]Division of Laboratory Medicine, University of California, San Francisco, San Francisco, CA, USA. [9]These authors contributed equally: Kailin Yin, Michael J. Peluso. ✉e-mail: timothy.henrich@ucsf.edu; nadia.roan@gladstone.ucsf.edu

## Methods

### Study participants

Participants were enrolled in LIINC (www.liincstudy.org; NCT04362150)[7], a prospective observational study enrolling individuals with prior nucleic acid-confirmed SARS-CoV-2 infection, regardless of the presence or absence of postacute symptoms. At each study visit, participants underwent an interviewer-administered assessment of 32 physical symptoms that were newly developed or had worsened since the COVID-19 diagnosis. Detailed data regarding medical history, COVID-19 history, SARS-CoV-2 vaccination and SARS-CoV-2 reinfection were collected. Two participants had biospecimens collected via the COVID-19 Host Immune Response Pathogenesis (CHIRP) study[5]. For the present study, we selected participants who consistently met a case definition for LC based on the presence or absence of at least one symptom attributable to COVID-19 for the 8 months following SARS-CoV-2 infection (Fig. 1a). The LC group ($n = 27$) had a median age of 46 years, and was comprised of 63% females and 26% of whom were previously hospitalized for COVID-19. The R group ($n = 16$) had a median age of 45.5 years, and was comprised of 44% females and 12.5% of whom were previously hospitalized for COVID-19 (Supplementary Table 1). Participants were deliberately not matched by age and sex, but we ensured that there was overlap in the groups. Blood samples were collected between September 16, 2020 and April 6, 2021. All participants provided a post-COVID blood sample before a SARS-CoV-2 vaccination to exclude the potential effects of SARS-CoV-2 vaccination on our study. Specimens were collected 8 months postinfection from individuals. All assays were performed from the same parent set of $n = 27$ LC and $n = 16$ specimens. All participants provided written informed consent.

### Biospecimen collection

Whole blood was collected in EDTA tubes followed by isolation of PBMCs and plasma as described in ref. 15. Serum was obtained concomitantly from serum-separator tubes.

### Serology

Antibody responses against SARS-CoV-2 spike RBD were measured on sera using the Pylon COVID-19 total antibody assay (ET Health) and reported as relative fluorescence units (RFUs).

### SARS-CoV-2 peptides

Peptides used for T cell stimulation comprised a mix of overlapping 15-mers spanning the entire SARS-CoV-2 spike protein (PM-WCPV-S-1, purchased from JPT), and peptides corresponding to CD8+ T cell epitopes identified by T-scan[16] synthesized in-house (Supplementary Table 12). Final peptide concentrations were 300 nM for the 15-mers and 450 nM for the T-scan peptides.

### CyTOF

Sample preparation was performed similar to methods described[2-5]. Upon revival of cryopreserved PBMCs, cells were rested overnight to allow for antigen recovery[17] and then divided equally into two aliquots. To the first aliquot, we added 3 µg ml⁻¹ brefeldin A (BFA; to enable intracellular cytokine detection), the costimulation agonists anti-CD28 (2 µg ml⁻¹; BD Biosciences) and anti-CD49d (1 µg ml⁻¹; BD Biosciences), and the SARS-CoV-2 peptide pool prepared as described above. To the second aliquot, we added 1% DMSO (Sigma-Aldrich) and 3 µg ml⁻¹ BFA. Cells from both treatments were incubated at 37°C for 6 h. Cells were treated with cisplatin (Sigma-Aldrich) as a live/dead distinguisher and fixed in paraformaldehyde (Electron Microscopy Sciences) as described[2-5]. CyTOF antibody conjugation was performed using the Maxpar X8 Antibody Labeling Kit (Standard BioTools) according to the manufacturer's instructions. CyTOF staining was performed as described[2-5], but using the CyTOF panel created for this study (Supplementary Table 4). Stained samples were washed with CAS buffer (Standard BioTools), spiked with 10% (vol/vol) EQ Four Element Calibration Beads (Standard BioTools) and run on a Helios CyTOF instrument (UCSF Parnassus Flow Core).

### CyTOF data analyses

**Data preprocessing.** EQ bead-normalized CyTOF datasets were concatenated, de-barcoded and normalized using Standard BioTools Software version 6.7. Following arcsinh transformation of the data[18], cells were analyzed by FlowJo (version 10.8.1, BD Biosciences). Intact (Ir191+Ir193+), live (Pt195−), singlet events were identified, followed by gating on CD3+ T cells, and sub-gating on CD4+ T cells and CD8+ T cells (Extended Data Fig. 1g,h).

**CyTOF antibody validation.** CyTOF antibodies in our panel (Supplementary Table 4) were validated using methods previously described, including the use of human lymphoid aggregate cultures generated from tonsils[2-5,18,19]. The observed expression patterns among tonsillar T and B cells (Extended Data Fig. 10a) were similar to those previously observed[18]. To validate the detection of cytokines and other effectors, we stimulated PBMCs with 16 nM phorbol 12-myristate 13-acetate (PMA) (Sigma-Aldrich) and 1 µM ionomycin (Sigma-Aldrich), or 1 µg ml⁻¹ lipopolysaccharides (LPS; eBioscience), for 4 h in the presence of 3 µg ml⁻¹ BFA solution (eBioscience), combined the cells and prepared them for CyTOF as described above. We observed the expected induction of cytokines or cytolytic markers (Extended Data Fig. 10b)[2-5] and preferential expression of $T_{reg}$ lineage marker Foxp3 among CD3+CD4+CD45RO+CD45RA−CD127−CD25+ $T_{reg}$ cells (Extended Data Fig. 10c). We also observed preferential expression of CD30 and Ki67 in CD4+ $T_M$ as compared to CD4+ $T_N$ cells (Extended Data Fig. 10d). Examples of $pT_{FH}$ and $T_{FH}$ gates are depicted in Extended Data Fig. 10e.

**Identification of SARS-CoV-2-specific T cells.** For identification of SARS-CoV-2-specific T cells, we compared unstimulated specimens to their peptide-stimulated counterparts. Effector cytokines (IFN-γ, TNF, IL-2, IL-4, IL-6, IL-17 and CCL4) and cytolytic effectors (granzyme B and perforin) were assessed for the ability to identify antigen-specific T cells at the single-cell level. The following criteria were established to identify effector molecules appropriate for identifying SARS-CoV-2-specific T cells: (1) counts of positive cells in unstimulated sample (not receiving peptide) was less than 5 events, or the frequency of positive cells was lower than 0.1%; (2) counts of positive cells in the peptide-stimulated sample was not less than 5, or the frequency was higher than 0.1%; (3) differences in frequencies of positive cells between unstimulated and peptide-stimulated samples cells was not less than 0.01%; (4) fold change in frequencies of positive cells between unstimulated and peptide-stimulated samples cells was greater than 10 and (5) the aforementioned four criteria could identify SARS-CoV-2-specific T cells among >50% of participants. Effectors that fulfilled all five criteria were IFN-γ, TNF and IL-2 for CD4+ T cells and IFN-γ, TNF and CCL4 for CD8+ T cells. For a sub-analysis to identify responding cells that may only exist in a small subset of individuals, we removed criterion 5 and reduced the positive cell counts to number 3 within criteria 1 and 2. This approach allowed us to determine that SARS-CoV-2-specific CD4+ T cells producing IL-6 were exclusively detected from LC (Extended Data Fig. 2f). SARS-CoV-2-specific T cells were detected at a median of 163 cells (134 for CD4+ T cells and 29 for CD8+ T cells) and a mean of 221.7 cells (185.2 for CD4+ T cells and 36.4 for CD8+ T cells), per participant. SARS-CoV-2-specific T cells, once identified, were analyzed by Boolean gating[20] and exported for further analyses.

**SPICE.** SPICE analyses were performed using version 6.1 software[21]. CD4+ and CD8+ T cells were subjected to manual gating based on the expression of cytokines used to define SARS-CoV-2-specific T cells (IFN-γ, TNF, IL-2 and CCL4, see above) using operations of Boolean logic. The parameters for running the dataset were as follows: iterations for permutation test = 10,000 and highlight values = 0.05. The parameters

for the query structure were set as follows: values = frequency of single cytokine positive cells in total CD4⁺/CD8⁺ T cells; category = IFN-γ, TNF, IL-2 and CCL4; overlay = patient type (LC versus non-LC); group = all other variables in the data matrix.

**T cell subsetting.** Manual gating was performed using R (version 4.1.3). Arcsinh-transformed data corresponding to total or SARS-CoV-2-specific CD4⁺ or CD8⁺ T cells were plotted as 2D plots using the CytoExploreR package. Visualization of datasets by t-SNE was performed using methods similar to those described[2–5]. CytoExploreR and tidyr packages were used to load the data, and t-SNE was performed using Rtsne and RColorBrewer packages on arcsinh-transformed markers. Total CD4⁺/CD8⁺ T cells were downsampled to $n = 8,000$ (maximal cell number for individual samples) before t-SNE analysis. The parameters for t-SNE were set as iteration = 1,000, perplexity = 30 and $\theta = 0.5$.

**T cell clustering analysis.** Flow cytometry standard (FCS) files corresponding to total and SARS-CoV-2-specific CD4⁺ and CD8⁺ T cells were imported in R for data transformation. Packages of flowcore, expss, class and openxlsx were loaded in R. Arcsinh-transformed data were then exported as CSV files for clustering analyses. Biological (LC status, biological sex and hospitalization status) and technical (batch/run of processing) variables were visualized using the DimPlot function of Seurat[22]. Batch correction was performed by RunHarmony[23]. Optimal clustering resolution parameters were determined using Random Forests[24] and a silhouette score-based assessment of clustering validity and subject-wise cross-validation, as detailed in ref. [25]. A generalized linear mixed model (GLMM, implemented in the lme4 (ref. [26]) package in R with family argument set to the binomial probability distribution) was used to estimate the association between cluster membership and LC status and the sex of the participant, with the participant modeled as a random effect. For each individual, cluster membership of cells was encoded as a pair of numbers representing the number of cells in the cluster and the number of cells not in the cluster. Clusters having fewer than three cells were discarded. The sex-specific log odds ratio of cluster membership association with LC status was estimated using the emmeans[27] R package using the GLMM model fit. The estimated log odds ratio represented the change (due to LC status) in the average over all participants of a given sex in the log odds of cluster membership. The two-sided $P$ values corresponding to the null hypothesis of an odds ratio value of 1 were computed based on a $z$ statistic in the GLMM model fit. These $P$ values were adjusted for multiple testing using the Benjamini–Hochberg method.

**Flow cytometry**

Flow cytometry was performed on PBMCs from 25 LC and 15 R individuals from our cohort, obtained from aliquots of specimens analyzed by CyTOF. Cells were stained with the panel shown in Supplementary Table 13, using Zombie UV or Zombie NIR (BioLegend) as viability indicators. All cells were analyzed on a Fortessa X-20 (BD Biosciences). FCS files were exported into FlowJo (BD, version 10.9.0) for further analysis. Flow cytometric data were arcsinh-scaled before analyses. In flow cytometric experiments, SARS-CoV-2-specific CD8⁺ T cells were defined as those specifically inducing IFN-γ and/or TNF in response to SARS-CoV-2 peptide stimulation, as the CCL4 antibody exhibited background staining in flow cytometry and could not be used to define SARS-CoV-2-specific T cells.

**RNA-seq**

RNA-seq was performed on PBMCs from 23 LC and 13 R individuals from our cohort, obtained from aliquots of specimens analyzed by CyTOF. Samples were prepared using the AllPrep kit (Qiagen) per the manufacturer's instructions. RNA libraries, next-generation Illumina sequencing, quality control analysis, trimming and alignment were performed by Genewiz (Azenta). Briefly, following oligo dT enrichment,

fragmentation and random priming, cDNA syntheses were completed. End repair, 5′ phosphorylation and dA-tailing were performed, followed by adaptor ligation, PCR enrichment and sequencing on an Illumina HiSeq platform using PE150 (paired-end sequencing, 150 bp for reads 1 and 2). Raw reads (480 Gb in total) were trimmed using Trimmomatic (version 0.36) to remove adapter sequences and poor-quality reads. Trimmed reads were mapped to *Homo sapiens* GRCh37 using star aligner (version 2.5.2b)[28]. $\log_2$ fold changes were calculated between LC versus R individuals. Two-sided $P$ values corresponding to a null hypothesis of fold change of 1 were calculated using DESeq2's (ref. [29]) Wald test and were adjusted for multiple testing using false discovery rates. Genes with an adjusted $P$ value < 0.05 and absolute $\log_2$(fold change) > 1 were considered significant DEGs. Clustered heatmaps of DEGs were constructed with groups of genes (rows) defined using the $k$-means algorithm to cluster genes into $k$ clusters based on their similarity. $K = 4$ was determined using the Hierarchical Ordered Partitioning and Collapsing Hybrid (HOPACH) algorithm[30], which recursively partitions a hierarchical tree while ordering and collapsing clusters at each level to identify the level of the tree with maximally homogeneous clusters.

**scRNA-seq**

scRNA-seq was performed on PBMCs from 8 LC and 4 R individuals from our cohort, obtained from aliquots of specimens analyzed by CyTOF. Library preparation was performed using the Chromium Next GEM Single-Cell 5′ Reagent Kits v2 (10x Genomics) and sequenced on the Illumina NovaSeq 6000 S4 300 platform. Samples were sequenced at a mean of >50k reads per cell (minimum 51k, maximum 120k and median 83k). A median of 7,888 cells was analyzed per donor (minimum 4,189 and maximum 9,511). Demultiplexed fastq files were aligned to human reference genome GRCh38 using the 10x Genomics Cell Ranger v7.1.0 count pipeline[31]. The include-introns flag for the count pipeline was set to true to count reads mapping to intronic regions. The filtered count matrices generated by the Cell Ranger count pipeline were processed using Seurat[22]. Each sample was preprocessed as a Seurat object, and the top 1% of cells per sample with the highest numbers of unique genes, cells with ≤200 unique genes and cells ≥10% mitochondrial genes were filtered out for each sample. The samples were then merged into a single Seurat object, and normalization and variance stabilization were performed using sctransform86 with the 'glmGamPoi' method[32] for initial parameter estimation.

Graph-based clustering was performed using the Seurat[22] functions FindNeighbors and FindClusters. First, the cells were embedded in a $k$-nearest neighbor graph (with $k = 20$) based on the Euclidean distance in the principal component analysis (PCA) space. The edge weights between the two cells were further modified using Jaccard similarity. Next, clustering was performed using the Louvain algorithm[33] implementation in the FindClusters Seurat function. Clustering with 15 principal components (PCs, determined based on the location of the elbow in the plot of variance explained by each of the top 25 PCs) and 0.1 resolution (determined using the resolution optimization method described above for CyTOF data clustering) resulted in 11 distinct biologically relevant clusters (clusters 0–11), which were used for further analyses. Marker genes for each cluster were identified using the FindAllMarkers Seurat function. Marker genes were filtered to keep only expressed genes detected in at least 25% of the cells, with at least 0.5 $\log_2$ fold change. Cluster annotation was performed according to subset definitions previously established[34–36]. Classification markers included *CD19*, *MS4A1* and *CD79A* for B cells; *CD3D*, *CD3E*, *CD5* and *IL7R* for CD4⁺ T cells; *CD3D*, *CD3E*, *CD8A*, *CD8B* and *GZMK* (CTL subset) for CD8⁺ T cells; *CD14*, *CD68*, *CYBB*, *S100A8*, *S100A9*, *S100A12* and *LYZ* for monocytes; *CSF2RA*, *LYZ*, *CXCL8* and *CD63* for granulocytes and *PF4*, *CAVIN2*, *PPBP*, *GNG11* and *CLU* for platelets.

The counts-per-million reads for *ALAS2* and *OR7D2* were assessed using edgeR[37], and associations with group status were made using

# Letter

the two-sample Welch $t$ test, followed by multiple correction testing using the Holm[38] procedure. For establishing associations between clusters and group status, GLMM implemented in the lme4 R package was used. The model was performed with the family argument set to the binomial probability distribution and with the 'nAGQ' parameter set to 10 corresponding to the number of points per axis for evaluating the adaptive Gauss–Hermite approximation for the log-likelihood estimation. Cluster membership was modeled as a response variable by a two-dimensional vector representing the number of cells from a given sample belonging or not to the cluster under consideration. The corresponding sample from which the cell was derived was the random effect variable, and the group (R, LC, $OR7D2$high LC, or $ALAS2$high LC) was considered the fixed variable. The log odds ratio for all pairwise comparisons was estimated using the model fits provided to the emmeans function in the emmeans R package[27]. The resulting $P$ values for the estimated log odds ratio and clusters were adjusted for multiple testing using the Benjamini–Hochberg method[39]. For associations of gene expression with group status, raw gene counts per cell were loaded as a SingleCellExperiment object. Cells from clusters 9 and 10 were not included in this analysis as the median number of cells across samples was less than 20 per cluster. The aggregateData function in the muscat bioconductor package[40] was used to pseudo-bulk the gene read counts across cells for each cluster group. Genes with raw counts less than ten in more than eight samples were removed from the analyses. The pbDS function implementing the statistical methods in the edgeR package[37] was used to assess associations of gene expression with group identity. Results from the cluster-specific pseudo-bulked gene expression association analyses were visualized as volcano plots using EnhancedVolcano[41,42]. Select genes of interest or genes that passed a multiple testing-adjusted $P$ value threshold of 0.05 or 0.1 as indicated were indicated in the volcano plots. For gene set enrichment analyses, the raw $P$ values for each gene derived from hypothesis tests for associations of interest were combined with a list of genes annotated with each of the gene sets in the biological processes domain of GO[43] and analyzed via the simultaneous enrichment analysis method[44] using the rSEA R package[45]. The family-wise error rate-adjusted $P$ values for cluster-specific associations of interest with each of the annotated gene sets were used to identify significant associations.

## Olink

The Olink EXPLORE 384 inflammation protein extension assay was performed per manufacturer's protocol as published in ref. [46].

## Data visualization

HOPACH[30] was used to find the best cluster number. Gene expression values were log-transformed and centered using the average expression value. Clustering was performed by running the $k$-means algorithm using the best cluster number $k$ found, and the results were plotted using the pheatmap package[47]. For gene network analyses, the STRING interaction database was used to reconstruct gene networks using stringApp[48] for Cytoscape[49]. For the network, the top 50 genes or 25 proteins with the lowest $P$ values were selected from the RNA-seq data and Olink data, respectively. They were then subjected to stringApp with an interaction score cutoff = 0.5 and the number of maximum additional indirect interactors cutoff = 10.

## Statistical tests

Unless otherwise indicated, permutation tests, two-tailed unpaired Student's $t$ tests and Welch's $t$ test were used for statistical analyses. *$P < 0.05$, **$P < 0.01$, ***$P < 0.001$, ****$P < 0.0001$ and NS. Error bars corresponded to s.d. Graphs were plotted by GraphPad Prism (version 9.4.1). All measurements were taken from distinct samples, no samples were measured repeatedly to generate data. Where appropriate, $P$ values were corrected for multiple testing (across three pairwise comparisons) using the Holm procedure[38]. Tests involving cluster

membership differences assumed a binomial probability distribution, and those involving RNA expression differences assumed a negative binomial probability distribution, but these were not formally tested. All other tests were based on the normality assumption but this was not formally tested.

## Statistics and reproducibility

No statistical method was used to predetermine the sample size. Samples were chosen based on the availability of specimens meeting our LC criteria. No samples were excluded from the analyses. Randomization was not implemented as the study compared LC to R individuals. Data collection and analysis were not performed blind to the conditions of the experiments.

## Reporting summary

Further information on research design is available in the Nature Portfolio Reporting Summary linked to this article.

## Data availability

The raw CyTOF datasets for this study corresponding to total and SARS-CoV-2-specific CD4+ and CD8+ T cells are publicly accessible through the following link: https://datadryad.org/stash/dataset/doi:10.7272/Q6WD3XTB. The raw Olink data are also downloadable through this link. The raw bulk RNA-seq and scRNA-seq data from this study are deposited in the Gene Expression Omnibus database—GSE224615 (for bulk RNA-seq) and GSE235050 (for scRNA-seq).

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

## Acknowledgements

This work was supported by the Van Auken Private Foundation, D. Henke, P. Taft and E. Taft; philanthropic funds donated to Gladstone Institutes by the Roddenberry Foundation and individual donors devoted to COVID-19 research; the Program for Breakthrough Biomedical Research, which is partly funded by the Sandler Foundation and awards 2164 and 2208 from Fast Grants, a part of Emergent Ventures at the Mercatus Center, George Mason University (to N.R.R.). We acknowledge the National Institutes of Health (NIH) DRC Center Grant P30 DK063720 and the S10 1S10OD018040-01 for use of the CyTOF instrument and the NIH S10 RR028962 and the James B. Pendleton Charitable Trust for use of the Fortessa X-20. This study was also funded by the Ministerium für Wissenschaft, Forschung und Kunst, Baden Württemberg, Germany (KNKC.031) and the Deutsche Forschungsgemeinschaft (DFG; German Research Foundation)—Projektnummer 316249678—SFB 1279 (to J.M.). Funding from the PolyBio Research Foundation supported both the experiments reported herein and the parent cohort (LIINC); specimen and clinical data collection were also supported by NIH 3R01AI141003-03S1 and NIH R01AI158013 (to M.J.P. and T.J.H.). The funders had no role in study design, data collection and analysis, decision to publish or preparation of the manuscript.

We thank S. Tamaki, V. Nguyen, P. Sanchez and C. Bispo for CyTOF assistance at the Parnassus Flow Core, J. Srivastava and V. Saware for technical assistance in flow cytometry, M. Karacan and N. Preising for technical assistance in peptide synthesis, E. Ghosn for guidance on annotation of cell clusters identified by scRNA-seq, J. Carroll for assistance on graphics, F. Chanut for editorial assistance and R. Givens for administrative assistance. We are grateful to the study participants and their medical providers. We acknowledge current and former LIINC clinical study team members T. Abualhsan, A. Alvarez, M. Arreguin, M. Buitrago, M. Deswal, N. DelCastillo, E. Fehrman, H. Grebe, H. Hartig, Y. Hernandez, M. Kerbleski, R. Kirtikar, J. Lombardo, M. Luna, L. Ngo, E. Ortiz, A. Rodriguez, J. Romero, D. Ryder, R. Sanchez, M. So, C. Song, V. Tai, A. Tang, C. Thanh, F. Ticas, L. Torres, B. Tran, D. Varma and M. Williams. We also acknowledge LIINC laboratory team members A. Buck, J. Donatelli, J. Hakim, N. Iyer, O. Janson, B. LaFranchi, C. Nixon, I. Thomas and K. Turcios. We thank J. Chen, A. Donovan and C. Forman for assistance with data entry and review. We thank the UCSF AIDS Specimen Bank for processing specimens and maintaining the LIINC biospecimen repository. We are grateful to E. Eilkhani and M. Deswal for regulatory support. We are also grateful for the contributions of additional current and former LIINC leadership team members—M. Durstenfeld, P. Hsue, B. Greenhouse, I. Rodriguez-Barraquer and R. Rutishauser.

## Author contributions

K.Y. designed the experiments, performed CyTOF, flow cytometry and scRNA-seq experiments, conducted analyses, and prepared figures and tables. M.J.P. designed the LIINC cohort, oversaw LIINC cohort procedures and interpreted data. X.L. developed pipelines for data analyses and performed scRNA-seq. R.T. performed clustering and scRNA-seq analyses. M.S. performed RNA-seq and Olink analyses. J.N. prepared peptides and helped with experiments. A.A. and K.C.Y. prepared and analyzed CyTOF specimens. T.M. designed protocols for CyTOF analyses. R.H., K.A. and B.H. managed the LIINC cohort, recruited participants, collected clinical data and collected biospecimens. U.A., M.L., D.V., K.A., T.D. and S.E.M. recruited LIINC participants, collected clinical data and collected biospecimens. T.D. and S.E.M. processed specimens. L.S. and J.M. synthesized peptides. R.I. entered, cleaned and performed quality control on LIINC data. S.L. and S.A.G. managed LIINC data and selected biospecimens. K.L.L. performed antibody assays. S.A.L. designed the CHIRP cohort

and oversaw CHIRP cohort procedures. J.D.K. and J.N.M. designed the LIINC cohort and interpreted LIINC clinical data. S.G.D. designed the LIINC cohort, oversaw cohort procedures and interpreted LIINC clinical data. T.J.H. designed the LIINC cohort, oversaw cohort procedures, performed the RNA-seq and Olink studies, interpreted data and prepared figures. N.R.R. conceived the study, performed supervision, conducted data analyses and prepared figures and tables. K.Y., M.J.P., T.J.H. and N.R.R. wrote the manuscript. All authors have read and approved this manuscript.

## Competing interests

M.J.P. reports consulting fees from Gilead Sciences and AstraZeneca, outside the submitted work. S.G.D. reports grants and/or personal fees from Gilead Sciences, Merck & Co., Viiv, AbbVie, Eli Lilly, ByroLogyx and Enochian Biosciences, outside the submitted work. T.J.H. receives grant support from Merck and consults for Roche. All other authors report no conflicts of interest.

## Additional information

**Extended data** is available for this paper at https://doi.org/10.1038/s41590-023-01724-6.

**Correspondence and requests for materials** should be addressed to Timothy J. Henrich or Nadia R. Roan.

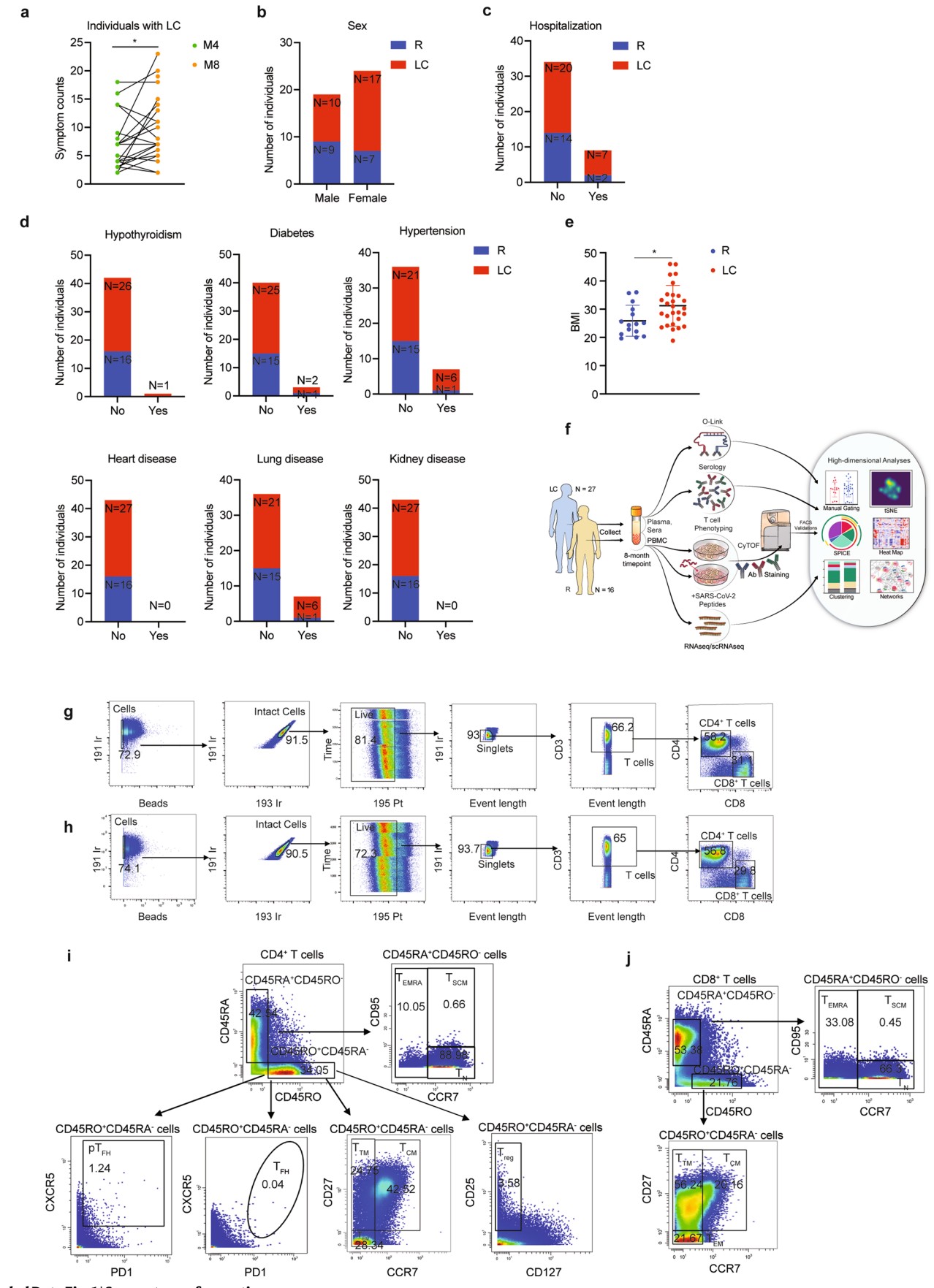

**Extended Data Fig. 1 | See next page for caption.**

**Extended Data Fig. 1 | Cohort characteristics, study design, and subset identification. a–c**, Number of sequelae symptoms at 4 (M4) and 8 (M8) months post-infection (n = 27 LC, n = 16 R) (**a**), and the numbers of individuals that were male or female (**b**) and that were hospitalized at the time of acute COVID-19 infection (**c**), in LC and R study participants. *p < 0.05 (two-sided paired sample t-test). **d**, The numbers of indicated co-morbidities in LC vs R study participants. **e**, BMI in LC vs R study participants. *p < 0.05 (two-sided student's t-test). Horizontal bars indicate mean, error bars indicate SD, and dots represent individuals, with n = 27 LC and n = 16R. **f**. Schematic of experimental design and data analyses. Blood specimens from 27 LC and 16 R individuals were subjected to Olink, serology, CyTOF, and RNA-seq and scRNA-seq analysis. The indicated tools on the right were then used for analyses of the resulting high-dimensional datasets. **g,h**, Gating strategy to identify T cell populations. Intact, live, singlet cells from baseline (**g**) or SARS-CoV-2 peptide-treated (**h**) samples were gated for CD3+ T cells followed by sub-gating on CD4+ and CD8+ T cells as indicated. **i,j**, Gating strategy to define classical CD4+ (**i**) and CD8+ (**j**) T cell subsets.

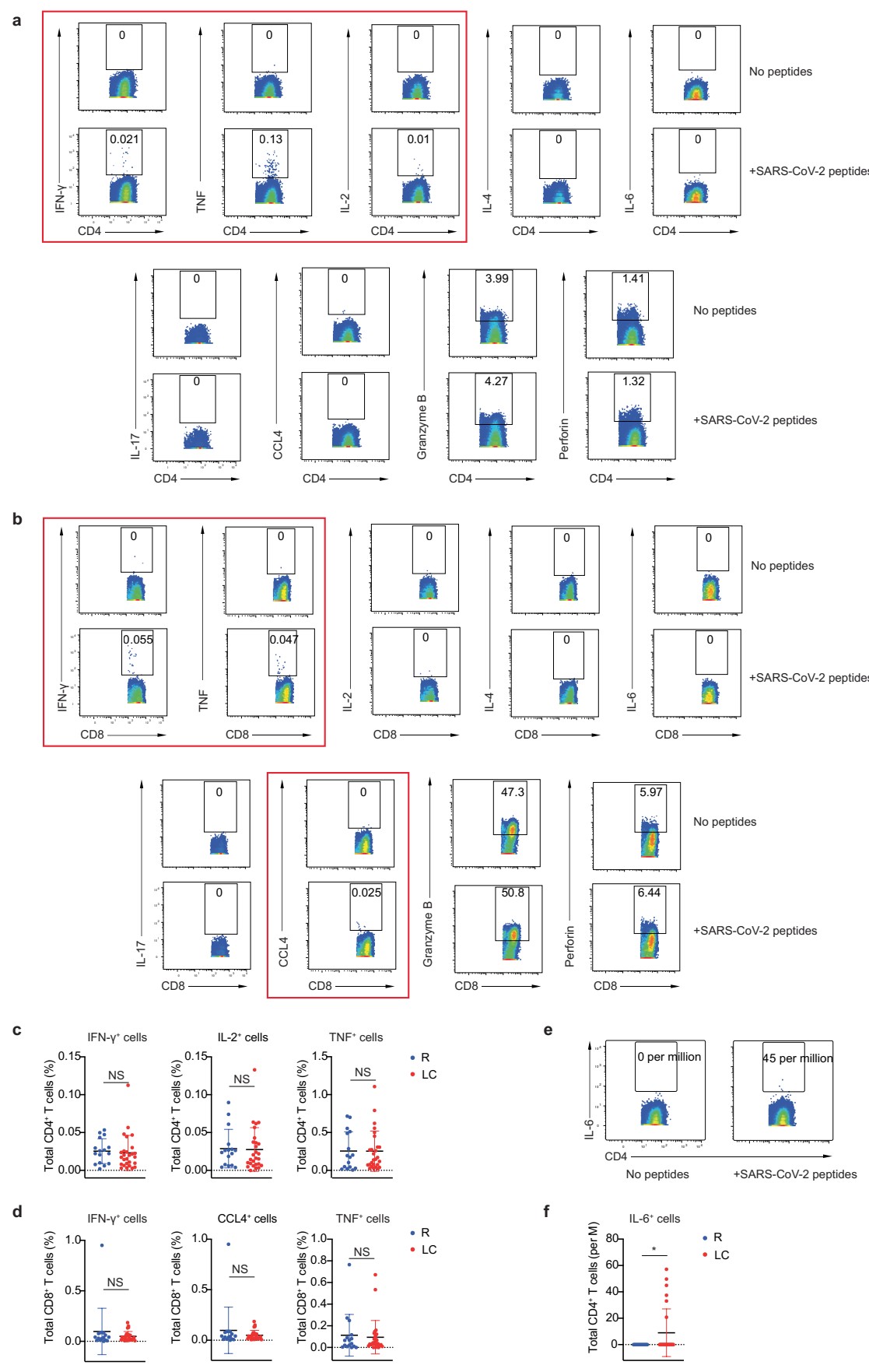

**Extended Data Fig. 2 | See next page for caption.**

**Extended Data Fig. 2 | Cytokine and effector molecule expression in SARS-CoV-2-specific T cells. a,b** CD4[+] (**a**) or CD8[+] (**b**) T cells from representative donor, stimulated (bottom) or not (top) with SARS-CoV-2 spike and T-scan peptides (Methods). Red boxes highlight the cytokines used to define the SARS-CoV-2-specific T cells. **c,d** The percentages of SARS-CoV-2-specific CD4[+] (**c**) and CD8[+] (**d**) T cells as defined by induction of IFN-γ, IL-2, CCL4, or TNF in response to SARS-CoV-2 peptide stimulations (two-sided student's t-test). **e,f,** IL-6[+] CD4[+] T cells are observed in LC individuals. **e**, CD4[+] T cells from representative donor, stimulated (right) or not (left) with SARS-CoV-2 spike and T-scan peptides (Methods). **f**, The percentages of SARS-CoV-2-specific CD4[+] T cells inducing IL-6 in response to SARS-CoV-2 peptide stimulations. *p < 0.05 (two-sided Welch's t-test). Horizontal bars indicate mean, error bars indicate SD, and dots represent individuals, with n = 27 LC and n = 16 R (**c**, **d**, **f**).

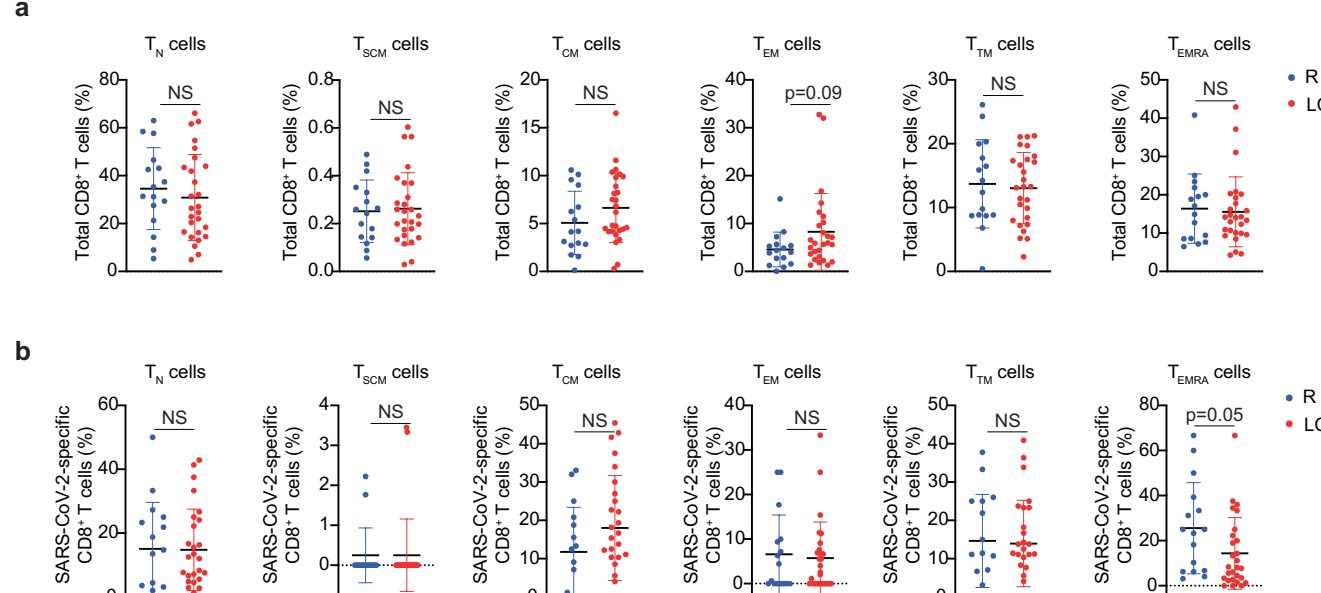

**Extended Data Fig. 3 | Subset distribution of total and SARS-CoV-2-specific CD8+ T cells among LC and R individuals. a**, Frequencies of $T_N$ cells, $T_{SCM}$ cells, $T_{CM}$ cells, $T_{EM}$ cells, $T_{TM}$ cells, and $T_{EMRA}$ cells among total CD8+ T cells from LC and R individuals (two-sided student's t-test). **b**, Frequencies of $T_N$ cells, $T_{SCM}$ cells, $T_{CM}$ cells, $T_{EM}$ cells, $T_{TM}$ cells, and $T_{EMRA}$ cells among SARS-CoV-2-specific CD8+ T cells from LC and R individuals (two-sided student's t-test).

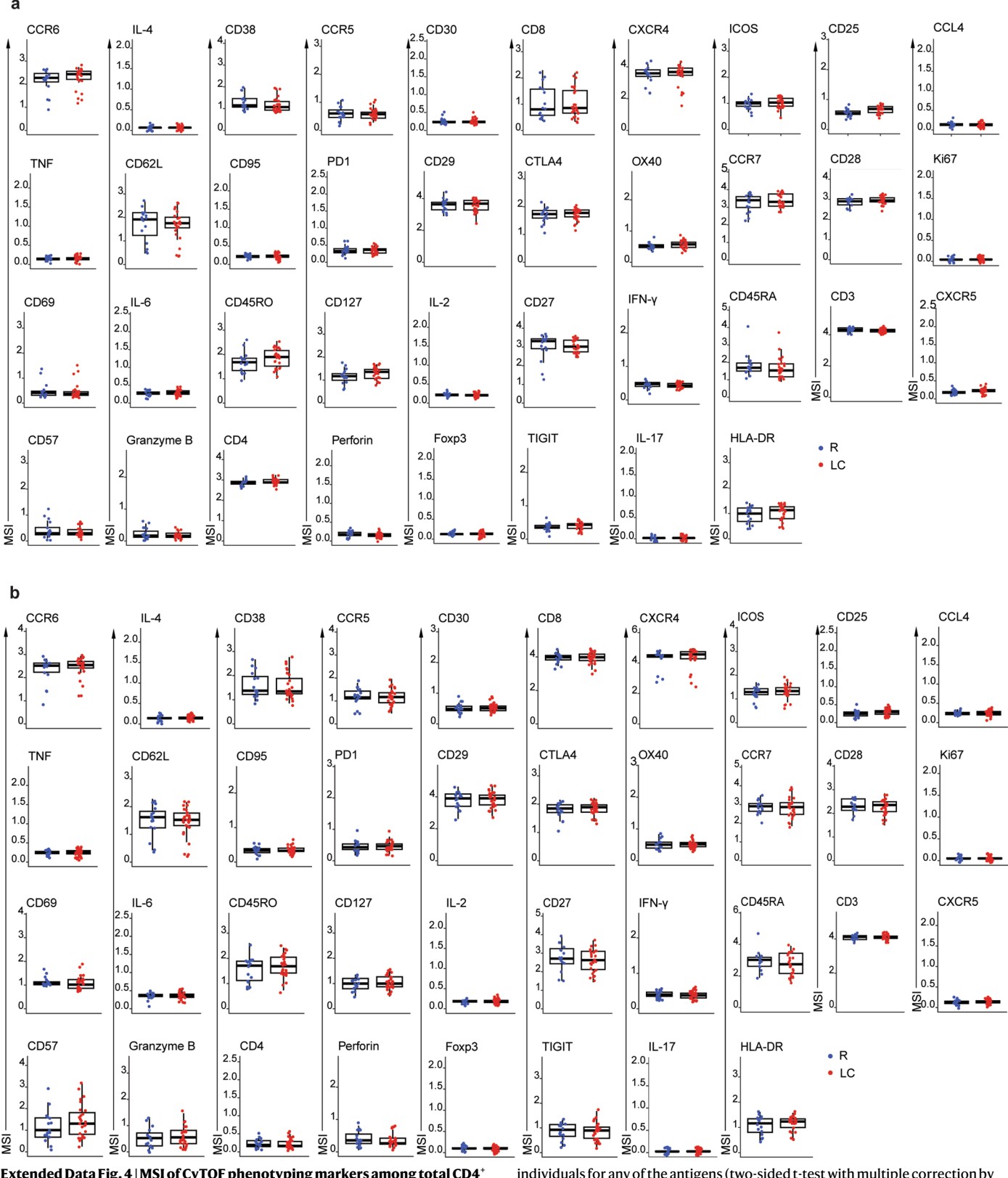

**Extended Data Fig. 4 | MSI of CyTOF phenotyping markers among total CD4+ and CD8+ T cells from LC and R individuals.** Antigens are shown in the order listed in Supplementary Table 4. Results are gated on live, singlet CD4+ (**a**) or CD8+ (**b**) T cells. No significant differences were observed between LC and R individuals for any of the antigens (two-sided t-test with multiple correction by Sidak adjustment). Box plots represent the median (middle bar), 75% quartile (upper hinge) and 25% (lower hinge) with whiskers extending 1.5× interquartile range, dots represent individuals with n = 27 LC and n = 16 R.

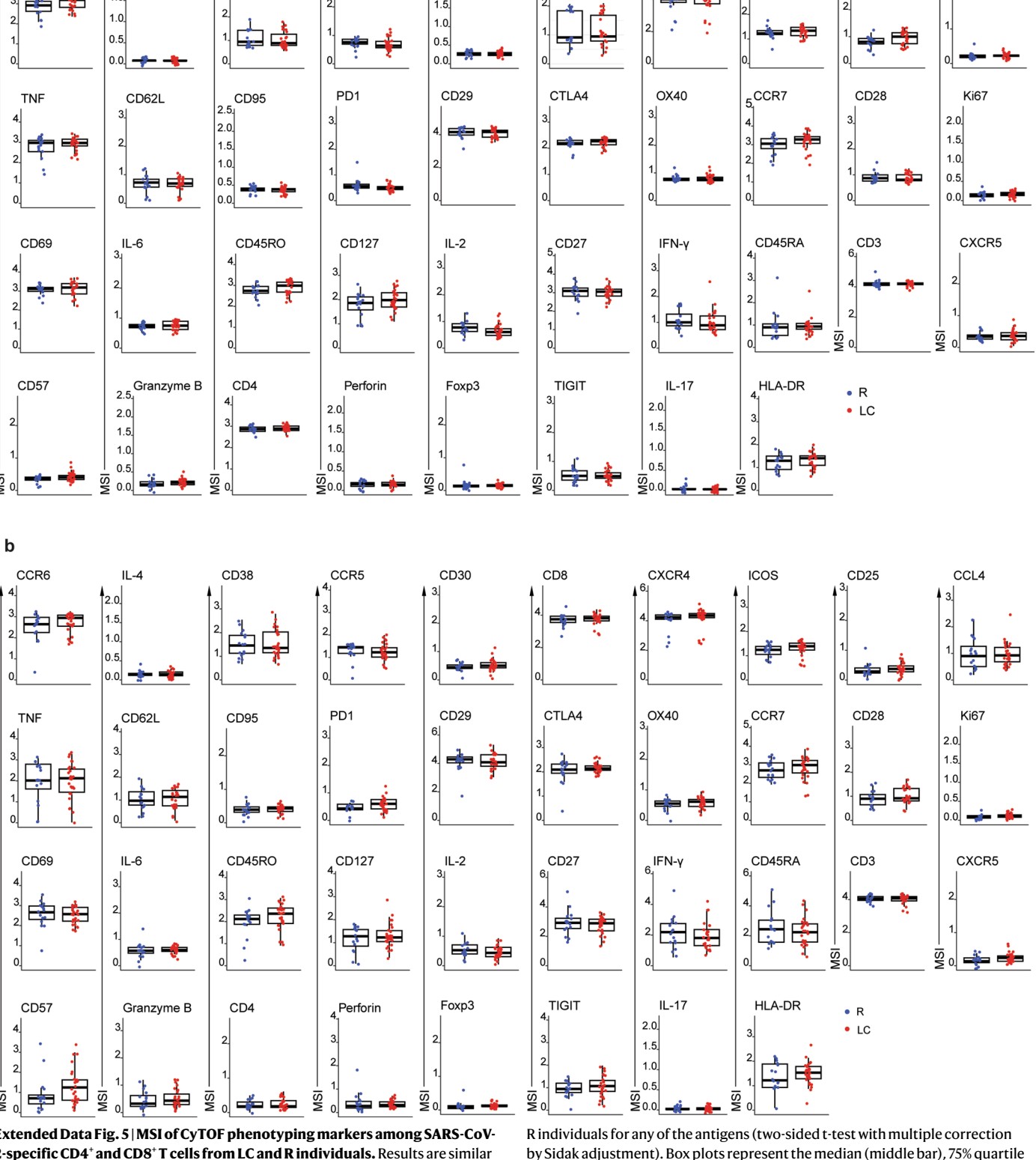

**Extended Data Fig. 5 | MSI of CyTOF phenotyping markers among SARS-CoV-2-specific CD4⁺ and CD8⁺ T cells from LC and R individuals.** Results are similar to that shown in Extended Data Fig. 4, but gated on SARS-CoV-2-specific CD4⁺ (**a**) or CD8⁺ (**b**) T cells. No significant differences were observed between LC and R individuals for any of the antigens (two-sided t-test with multiple correction by Sidak adjustment). Box plots represent the median (middle bar), 75% quartile (upper hinge) and 25% (lower hinge) with whiskers extending 1.5× interquartile range, dots represent individuals with n = 27 LC and n = 16 R.

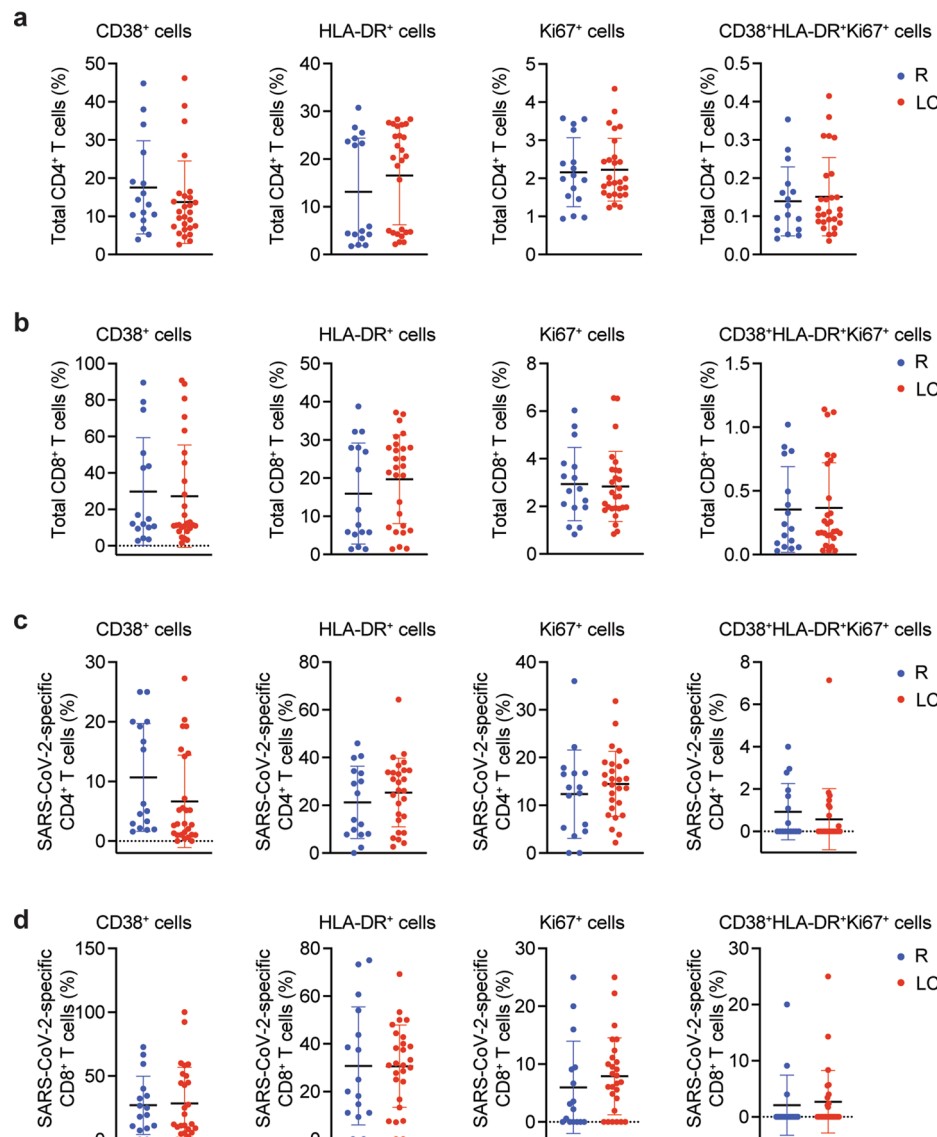

**Extended Data Fig. 6 | Activated T cells are not more abundant in individuals with LC.** The percentages of total CD4+ T cells (**a**), total CD8+ T cells (**b**), SARS-CoV-2-specific CD4+ T cells (**c**), and SARS-CoV-2-specific CD8+ T cells (**d**) expressing acute activation markers CD38, HLA-DR, and/or Ki67 in LC and R individuals (two-sided student's t-tests). Horizontal bars indicate mean, error bars indicate SD, and dots represent individuals, with n = 27 LC and n = 16 R.

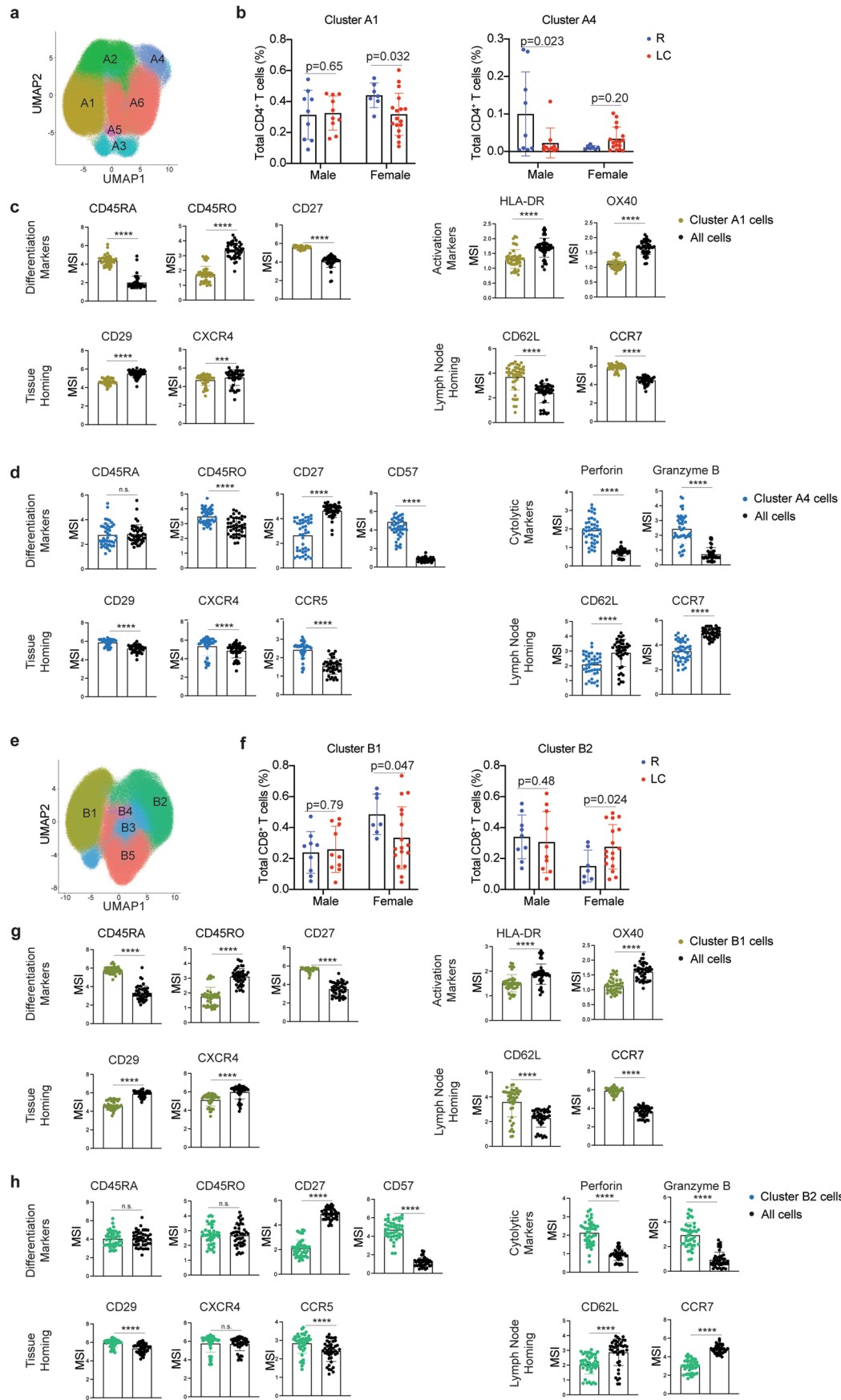

**Extended Data Fig. 7 | See next page for caption.**

**Extended Data Fig. 7 | Sex-dimorphic T cell cluster distribution in individuals with LC. a**, Cluster distribution among total CD4⁺ T cells as depicted by UMAP. **b**, The distributions of CD4⁺ T cell clusters A1 and A4 in male and female individuals, with or without LC. Two-sided p-values were derived from a GLMM fit (see Methods). Individual points represent individuals, with n = 10 LC and n = 9 R in the male group and n = 17 LC and n = 7 R in the female group, and where the value corresponds to % of cells belonging to clusters A1 or A4. **c**, Expression levels of differentiation markers (CD45RA, CD45RO, CD27), activation markers (HLA-DR, OX40), tissue homing receptors (CD29, CXCR4), and lymph node homing receptors (CD62L, CCR7) on CD4⁺ T cell cluster A1 compared to total baseline CD4⁺ T cells. **d**, Expression levels of differentiation markers (CD45RA, CD45RO, CD27, CD57), cytolytic effectors (perforin, granzyme B), tissue homing receptors (CD29, CXCR4, CCR5), and lymph node homing receptors (CD62L, CCR7) on CD4⁺ T cell cluster A4 compared to total baseline CD4⁺ T cells. **e**, Cluster distribution among total CD8⁺ T cells as depicted by UMAP. **f**, The distributions of CD8⁺ T

cell clusters B1 and B2 in male and female individuals, with or without LC. Two-sided p-values were derived from a GLMM fit (see Methods). Individual points represent individuals, with n = 10 LC and n = 9 R in the male group and n = 17 LC and n = 7 R in the female group, and where the value corresponds to % of cells belonging to clusters B1 or B2. **g**, Expression levels of differentiation markers (CD45RA, CD45RO, CD27), activation markers (HLA-DR, OX40), tissue homing receptors (CD29, CXCR4), and lymph node homing receptors (CD62L, CCR7) on CD8⁺ T cell cluster B1 compared to total baseline CD8⁺ T cells. **h**, Expression levels of differentiation markers (CD45RA, CD45RO, CD27, CD57), cytolytic effectors (perforin, granzyme B), tissue homing receptors (CD29, CXCR4, CCR5), and lymph node homing receptors (CD62L, CCR7) on CD8⁺ T cell cluster B2 compared to total baseline CD8⁺ T cells. ****p < 0.0001 (two-sided paired t-test, **c**,**d**,**g**,**h**). Horizontal bars indicate mean, error bars indicate SD, and dots represent individuals, with n = 27 LC and n = 16 R (**b**–**d**,**f**–**h**).

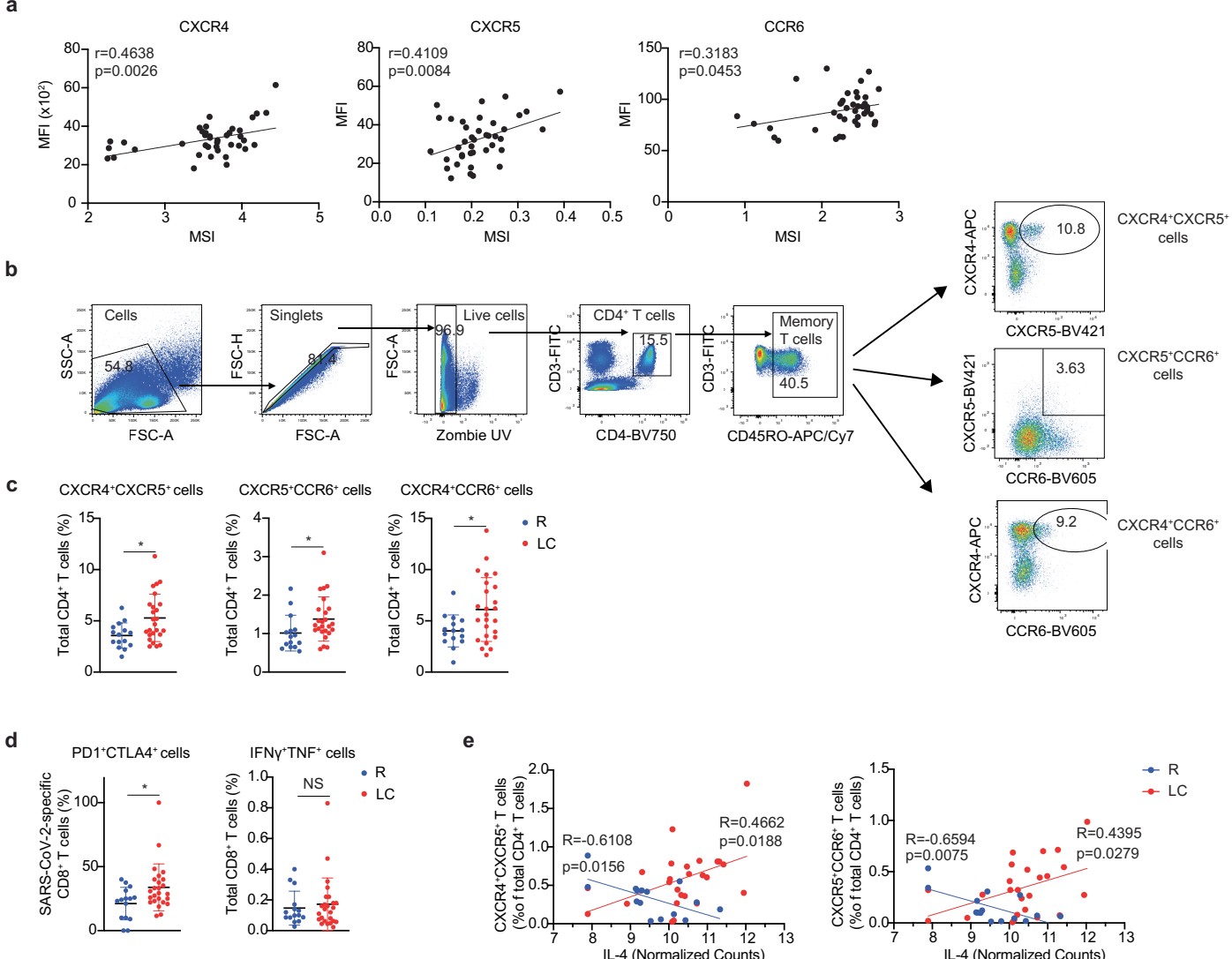

**Extended Data Fig. 8 | Flow cytometric validation and association analyses. a**, Association of flow cytometric (mean fluorescence intensity, MFI) vs CyTOF (MSI) expression levels of CXCR4, CXCR5, and CCR6. Data were analyzed by Pearson correlation coefficient and two-tailed unpaired t-tests. **b**, Flow cytometric gating strategy to identify memory CD4[+] T cells expressing various combinations of CXCR4, CXCR5, and CCR6. **c**, The percentages of CXCR4[+]CXCR5[+]CD4[+], CXCR5[+]CCR6[+]CD4[+], and CXCR4[+]CCR6[+]CD4[+] T cells in LC vs R individuals as determined by flow cytometry. *p < 0.05 (two-sided student's t-test). **d**, The percentages of cells dually expressing PD1 and CTLA4 among SARS-CoV-2-specific CD8[+] (left) or cells dually expressing IFN-γ and TNF among total CD8[+] T cells (right), as determined by flow cytometry. *p < 0.05 (two-sided student's t-test). Horizontal bars indicate mean, error bars indicate SD, and dots represent individuals, with n = 25 LC and n = 15 R (**c**,**d**). **e**, Associations of percentages of CXC4[+]CXCR5[+]CD4[+] T cells or CXCR5[+]CCR6[+]CD4[+] T cells with IL-4 levels in LC vs R individuals. Data were analyzed by Pearson correlation coefficient and two-tailed unpaired t-tests.

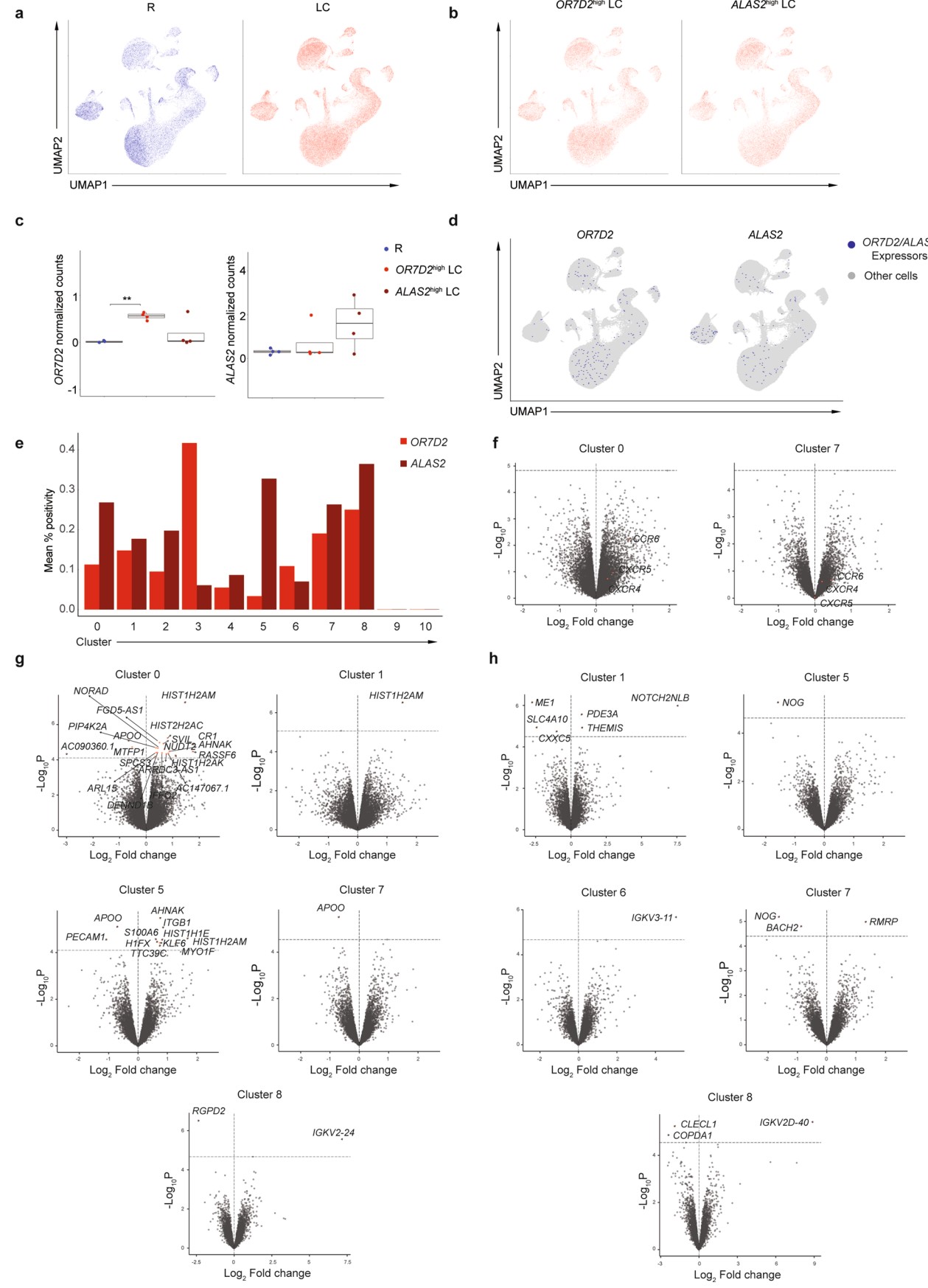

**Extended Data Fig. 9 | See next page for caption.**

**Extended Data Fig. 9 | scRNAseq analysis reveals *OR7D2* and *ALAS2* expression in multiple subsets, validates tissue-homing chemokine receptor expression among LC CD4⁺ T cells, and identifies DEGs among subsets in LC individuals. a**,**b**, UMAP of cells analyzed by scRNA-seq among LC (n = 8) vs R (n = 4) individuals (**a**), and among the LC individuals classified as *OR7D2*^high (n = 4) vs. *ALAS2*^high (n = 4) (**b**). **c**, *OR7D*2 and *ALAS2* expression in the *OR7D2*^high LC, *ALAS2*^high LC, and R individuals. **p < 0.01 (two-sided Welch two-sample t-test). Box plots represent the median (middle bar), 75% quartile (upper hinge) and 25% (lower hinge) with whiskers extending 1.5× interquartile range, dots represent individuals with n = 8 LC and n = 4R. **d**, UMAP depictions of cells expressing (blue) or not expressing (grey) *OR7D2* or *ALAS2* in individuals with LC. **e**, *OR7D2* and *ALAS2* expression in scRNA-seq-identified clusters labeled in Fig. 5d in individuals with LC, depicted as mean % of cells that were positive for *OR7D2* or *ALAS2* reads. **f**, Volcano plots showing LC vs R individuals for scRNA-seq-identified CD4⁺ T cell clusters 0 and 7, depicting *CXCR4*, *CXCR5*, and *CCR6*. **g**,**h**, Volcano plots depicting scRNA-seq-defined clusters 0, 1, 5, 7, and 8 for *OR7D2*^high vs. R (**g**), or clusters 1, 5, 6, 7, and 8 for *ALAS2*^high vs. R (**h**) individuals. DEGs with p < 0.05 (as determined empirical Bayes quasi-likelihood F-tests, with Benjamini-Hochberg correction) are labeled. Genes preferentially expressed in LC individuals are depicted on the right, and those preferentially expressed in R individuals on the left. The x-axes represent the log₂(fold-change) of the mean expression of each gene between the comparison groups, and the y-axes represent the raw −log₁₀(p-values). Dashed horizontal lines delineate the thresholds corresponding to Benjamini-Hochberg adjusted two-tailed p-values of <0.05 (Methods).

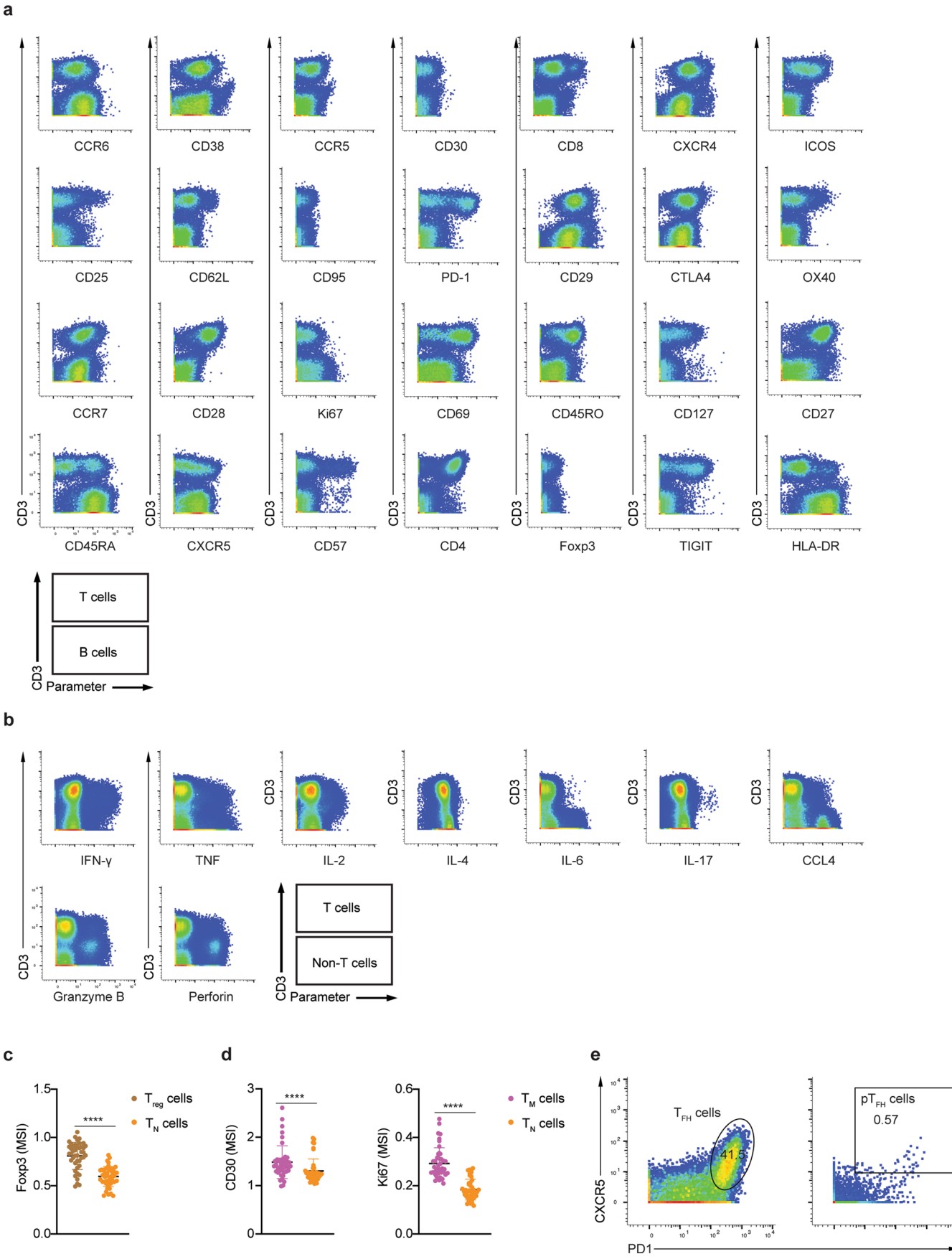

**Extended Data Fig. 10 | See next page for caption.**

**Extended Data Fig. 10 | Validation of CyTOF antibodies. a**, CyTOF analysis of human lymphoid aggregate cultures generated from tonsils depicting CD3$^+$ T cells on the top and CD3$^-$ B cells on the bottom as indicated, analogous to methods previously described[18]. **b**, CyTOF analysis of PMA/ionomycin- or LPS-stimulated PBMCs, depicting CD3$^+$ T cells on the top and CD3$^-$ cells on the bottom, similar to prior studies[2–5]. **c**, Expression of Foxp3 among CD4$^+$ T$_{reg}$ cells and CD4$^+$ T$_N$ cells, as assessed by CyTOF. **d**, Expression of CD30 and Ki67 among CD3$^+$ CD45RO$^+$CD45RA$^-$CD4$^+$ T memory (T$_M$) cells and CD4$^+$ T$_N$ cells, as assessed by CyTOF. ****p < 0.0001 (two-sided paired t-test). **e**, Illustration of pT$_{FH}$ gate implemented on PBMC samples, and T$_{FH}$ gate implemented on tonsil samples. Cells were pre-gated on CD4$^+$ T$_M$ cells.

# Reporting Summary

## Statistics

For all statistical analyses, confirm that the following items are present in the figure legend, table legend, main text, or Methods section.

| n/a | Confirmed | |
|---|---|---|
| ☐ | ☒ | The exact sample size (*n*) for each experimental group/condition, given as a discrete number and unit of measurement |
| ☐ | ☒ | A statement on whether measurements were taken from distinct samples or whether the same sample was measured repeatedly |
| ☐ | ☒ | The statistical test(s) used AND whether they are one- or two-sided<br>*Only common tests should be described solely by name; describe more complex techniques in the Methods section.* |
| ☒ | ☐ | A description of all covariates tested |
| ☐ | ☒ | A description of any assumptions or corrections, such as tests of normality and adjustment for multiple comparisons |
| ☐ | ☒ | A full description of the statistical parameters including central tendency (e.g. means) or other basic estimates (e.g. regression coefficient) AND variation (e.g. standard deviation) or associated estimates of uncertainty (e.g. confidence intervals) |
| ☐ | ☒ | For null hypothesis testing, the test statistic (e.g. *F*, *t*, *r*) with confidence intervals, effect sizes, degrees of freedom and *P* value noted<br>*Give P values as exact values whenever suitable.* |
| ☒ | ☐ | For Bayesian analysis, information on the choice of priors and Markov chain Monte Carlo settings |
| ☐ | ☒ | For hierarchical and complex designs, identification of the appropriate level for tests and full reporting of outcomes |
| ☐ | ☒ | Estimates of effect sizes (e.g. Cohen's *d*, Pearson's *r*), indicating how they were calculated |

*Our web collection on statistics for biologists contains articles on many of the points above.*

## Software and code

Policy information about availability of computer code

| Data collection | No software was used for data collection. |
|---|---|
| Data analysis | The Seurat package version 4.3.0 was used for scRNAseq analysis. FlowJo (version 10.8.1) was used for CyTOF data analyses. |

For manuscripts utilizing custom algorithms or software that are central to the research but not yet described in published literature, software must be made available to editors and reviewers. We strongly encourage code deposition in a community repository (e.g. GitHub). See the Nature Portfolio guidelines for submitting code & software for further information.

## Data

Policy information about availability of data

All manuscripts must include a data availability statement. This statement should provide the following information, where applicable:
- Accession codes, unique identifiers, or web links for publicly available datasets
- A description of any restrictions on data availability
- For clinical datasets or third party data, please ensure that the statement adheres to our policy

The raw CyTOF datasets for this study corresponding to total and SARS-CoV-2-specific CD4+ and CD8+ T cells, as well as the raw Olink data, are publicly accessible through the following link: https://datadryad.org/stash/share/TE_QuY0JX23V2n2CIMO2PgsR6afIp6GGusdQ5nXVGnk. The human reference genome (GRCh38) was used for alignment. The raw bulk RNAseq and scRNAseq data from this study are deposited in the GEO (Gene Expression Omnibus) database: GSE224615 (for bulk RNAseq) and GSE235050 (for scRNAseq).

# Human research participants

Policy information about <u>studies involving human research participants and Sex and Gender in Research.</u>

| | |
|---|---|
| Reporting on sex and gender | Sex as a biological variable were considered in this study. Data were analyzed by biological sex at birth, as determined by self-reporting. Sex-disaggregated data are presented in the manuscript. |
| Population characteristics | Our research home is at San Francisco General Hospital, a large, public safety-net hospital that serves individuals with limited access to healthcare services. Our research center is located in the heart of the Mission, a working-class Latino community, and adjacent to Bayview, a working-class Black community. All the investigators are committed to promoting inclusivity and equity. The SARS-CoV-2 pandemic in San Francisco has disproportionately affected communities of color. Our team has a long history of engaging affected communities in research. Our clinical research home at San Francisco General Hospital (SFGH) has a social justice mission and serves as the safety net public hospital for the Bay Area. LIINC is highly diverse including participants who identify as cisgender women (45%), transgender men and women (3% and 2%, respectively), Latino (38%) including monolingual Spanish speakers (14%), Black (5%), Asian (15%), Pacific Islander (5%) and Native American (3%). The median age of our participants was 46 (46 among LC group, 45.5 among non-LC group). A subset of our participants experience physical disabilities, as well as limitations related to ME/CFS, dysautonomia, and post-exertional malaise. Our team is highly diverse and includes individuals from a variety of backgrounds including immigrants to the U.S., people of color, and LGBTQ+ individuals. We have a diverse and inclusive Community Advisory Board, representing the full spectrum of community, sociodemographic, and sexual identity. We believe that our Long COVID work gives agency and voice to individuals experiencing this medically unexplained condition. All recruitment materials, our study website, and study instruments are available in both English and Spanish, and we are currently developing Spanish-language materials to disseminate preliminary study findings to participants. Retention in the cohort is high; over 85% of participants remain in the study after the first 14 months. |
| Recruitment | Details of cohort recruitment, enrollment, and measurement procedures have been described in detail previously (PMID 35106317). Briefly, LIINC is a prospective observational study enrolling individuals with prior nucleic acid-confirmed SARS-CoV-2 infection in the San Francisco Bay Area, regardless of the presence or absence of post-acute symptoms. At each study visit, participants underwent an interviewer-administered assessment of 32 physical symptoms that were newly developed or had worsened since COVID-19 diagnosis, as well as assessment of mental health and quality of life. Pre-existing and unchanged symptoms were not considered to be attributable to COVID-19. In addition, detailed data regarding medical history, COVID-19 history, SARS-CoV-2 vaccination, and SARS-CoV-2 reinfection were collected. Two participants enrolled in LIINC had biospecimens collected previously via the UCSF COVID-19 Host Immune Response Pathogenesis (CHIRP) study, which utilizes identical procedures for ascertainment of clinical history as the LIINC study (PMID 34636722). |
| Ethics oversight | The study protocol was approved by the UCSF Institutional Review Board (IRB). |

Note that full information on the approval of the study protocol must also be provided in the manuscript.

# Field-specific reporting

Please select the one below that is the best fit for your research. If you are not sure, read the appropriate sections before making your selection.

☒ Life sciences    ☐ Behavioural & social sciences    ☐ Ecological, evolutionary & environmental sciences

For a reference copy of the document with all sections, see <u>nature.com/documents/nr-reporting-summary-flat.pdf</u>

# Life sciences study design

All studies must disclose on these points even when the disclosure is negative.

| | |
|---|---|
| Sample size | No sample-size calculation was performed, as this is the first study to perform this kind of analysis, and well-annotated specimens from well-characterized and individuals with a clear post-acute sequelae of SARS-CoV-2 infection diagnosis are limited. |
| Data exclusions | No data were excluded from the analyses. |
| Replication | Reproducibility of experimental findings was established as detailed in the manuscript. For each assay, each donor was only measured once; hence all replicates corresponded to biological replicates, and not technical replicates. These biological replicates ranged from n=4 to n=27, as detailed within the manuscript. Of note, only statistically significant findings were used to draw conclusions. |
| Randomization | This is not pertinent to our study, because our study was not a clinical trial. In addition, we could not randomize since we were comparing two patient groups: LC vs. non-LC. |
| Blinding | Blinding was not appropriate for our study, as the CyTOF data needed to be generated in multiple batches, and equal distribution of study groups between batches was established so as to minimize the effect of batch on data outcome. This required knowing which samples belonged to which patient group. The RNAseq and scRNAseq data were run in single batches each. |

# Reporting for specific materials, systems and methods

We require information from authors about some types of materials, experimental systems and methods used in many studies. Here, indicate whether each material, system or method listed is relevant to your study. If you are not sure if a list item applies to your research, read the appropriate section before selecting a response.

## Materials & experimental systems

| n/a | Involved in the study |
|-----|-----------------------|
| ☐ | ☒ Antibodies |
| ☒ | ☐ Eukaryotic cell lines |
| ☒ | ☐ Palaeontology and archaeology |
| ☒ | ☐ Animals and other organisms |
| ☒ | ☐ Clinical data |
| ☒ | ☐ Dual use research of concern |

## Methods

| n/a | Involved in the study |
|-----|-----------------------|
| ☒ | ☐ ChIP-seq |
| ☐ | ☒ Flow cytometry |
| ☒ | ☐ MRI-based neuroimaging |

## Antibodies

| | |
|---|---|
| Antibodies used | Detailed information about antibodies used, including antibody dilutions and vendors are provided in the Tables and Methods sections. |
| Validation | All antibodies were validated by CyTOF on multiple cell types, prior to their application on the test samples. In particular, antibodies were tested on single-cell suspensions generated from fresh human tonsils, where expression of each antigen was compared between B cells and T cells, or different subset of T cells. Only batches of antibodies staining as expected based on known expression on cellular subsets were used. The validation studies are depicted in Extended Figure 10. |

## Flow Cytometry

### Plots

Confirm that:

☒ The axis labels state the marker and fluorochrome used (e.g. CD4-FITC).

☒ The axis scales are clearly visible. Include numbers along axes only for bottom left plot of group (a 'group' is an analysis of identical markers).

☒ All plots are contour plots with outliers or pseudocolor plots.

☒ A numerical value for number of cells or percentage (with statistics) is provided.

### Methodology

| | |
|---|---|
| Sample preparation | *Describe the sample preparation, detailing the biological source of the cells and any tissue processing steps used.* |
| Instrument | *Identify the instrument used for data collection, specifying make and model number.* |
| Software | *Describe the software used to collect and analyze the flow cytometry data. For custom code that has been deposited into a community repository, provide accession details.* |
| Cell population abundance | *Describe the abundance of the relevant cell populations within post-sort fractions, providing details on the purity of the samples and how it was determined.* |
| Gating strategy | *Describe the gating strategy used for all relevant experiments, specifying the preliminary FSC/SSC gates of the starting cell population, indicating where boundaries between "positive" and "negative" staining cell populations are defined.* |

☐ Tick this box to confirm that a figure exemplifying the gating strategy is provided in the Supplementary Information.

