## [Peer Review File · Nature Immunology]

Peer Review Information

Journal: Nature Immunology

Manuscript Title: Long COVID manifests with T cell dysregulation, inflammation, and an uncoordinated adaptive immune response to SARS-CoV-2

Corresponding author name(s): Dr Nadia Roan, Dr Timothy Henrich

Reviewer Comments & Decisions:

Decision Letter, initial version:
--

27th Mar 2023

Dear Dr. Roan,

Thank you for your response to the referees comments on your manuscript entitled "Long COVID manifests with T cell dysregulation, inflammation, and an uncoordinated adaptive immune response to SARS-CoV-2", reference number NI-A35392. The reviewers have raised substantial concerns and as such, we cannot accept the current manuscript for publication.

Should you find yourself able to thoroughly address the referees' concerns, please let me know. We believe it is important to present the entirety of the analysis, including the markers that did not show differences, to highlight the scope of the study. In addition, standard flow cytometry or other assays should be used to validate or investigate observations of interest, as requested by the referees (including on cytokine receptors). While we understand the limitations of a human study and heterogeneous disease, the study should not be driven by the choice of markers in the CyTOF panel. It would have been desirable to include the autoantibody observations in the current paper, but alternatively, please refer to it and reference the preprint study. Finally, it would be desirable, although we understand the limitations, to provide both bulk RNA and scRNA transcriptomic analysis on these valuable samples. If the novelty of the paper has not been compromised in the interim, we would invite you to submit a new paper, although we hope you understand that we cannot promise that it will be sent back for peer review until we've read the revised manuscript in its entirety.

At resubmission, please include a "Response to referees" detailing, point-by-point, how you addressed each referee comment. This response will be sent back to the referees along with the revised manuscript. In addition, please include a revised version of any required reporting checklist. It will be available to referees (and, potentially, statisticians) to aid in their evaluation if the manuscript goes back for peer review. A revised checklist is essential for re-review of the paper.

The Reporting Summary can be found here:

<https://www.nature.com/authors/policies/ReportingSummary.pdf>.

The Editorial Policy Checklist can be found here: <https://www.nature.com/authors/policies/Policy.pdf>.

In addition, please pay close attention to our [href="https://www.nature.com/nature-research/editorial-policies/image-integrity">Digital Image Integrity Guidelines.](https://www.nature.com/nature-research/editorial-policies/image-integrity) and to the following points below:

Nature Immunology is committed to improving transparency in authorship. As part of our efforts in this direction, we are now requesting that all authors identified as 'corresponding author' on published papers create and link their Open Researcher and Contributor Identifier (ORCID) with their account on the Manuscript Tracking System (MTS), prior to acceptance. This applies to primary research papers only. ORCID helps the scientific community achieve unambiguous attribution of all scholarly contributions. You can create and link your ORCID from the home page of the MTS by clicking on 'Modify my Springer Nature account'. For more information please visit [please visit www.springernature.com/orcid](http://www.springernature.com/orcid).

Please do not hesitate to contact me if you have any questions. Thank you for the opportunity to review your work.

Sincerely,

Ioana Visan, Ph.D.
Senior Editor
Nature Immunology

Tel: 212-726-9207
Fax: 212-696-9752
www.nature.com/ni

Reviewers' comments:

Reviewer #1 (Remarks to the Author):

In their manuscript, Yin et al. seek to define immunological PASC by deploying deep immune profiling of T cells, anti-viral serology, RNAseq of PBMC and plasma profiling for inflammation-associated soluble factors (Olink) by comparing cohorts of individuals with and without “long-COVID” (LC and non-LC) eight months after acute SARS-CoV-2 infection and prior to SARS-CoV-2 vaccination. The cohort selection, despite its small size, constitutes a particular strength of the work as does the focus on SARS-CoV-2 specific T cell immunity as well as a balanced discussion of their specific findings in the context of relevant prior literature. Nevertheless, and although the authors generously emphasize that LC is associated “with T cell dysregulation, inflammation and an uncoordinated adaptive immune response to SARS-CoV-2”, the reported differences, although intriguing, would appear to be somewhat modest; furthermore, it is not clear if they sufficiently leveraged the granularity of their data sets and the potential for integrated data analyses.

Much of the analytical effort is devoted to the characterization of T cell properties in LC and non-LC individuals (4 out of 6 main data figures, 6 out of 7 supplementary data figures) and the use of short-term peptide stimulation in conjunction with inducible T cell effector functions allows for better capture and characterization of specific T cell populations as compared to the frequently used detection of AIMs following longer stimulation cultures. The combinatorial analyses of specific CD4 and CD8 T cells (via induced IFN γ , TNF α , IL-2 or IFN γ , TNF α , MIP1b, respectively) and designation of suitable denominators appears convincing (Figure 2) though additional readouts (eg, CD40L for CD4 T cells) would have been helpful. The LAMP1 analyses, however, are confusing. To assess degranulation of specific T cells, LAMP1 antibodies need to be included in the stimulation culture; in contrast, the value of intracellular LAMP1 staining after stimulation as performed here remains unclear.

The deep immune profiling of SARS-CoV-2 specific T cells in LC and non-LC individuals as afforded by the authors’ CyTOF analyses (Figures 3B, 4 and 5) offers a unique opportunity to define and elucidate potential differences yet there remain several concerns:

1. Unclear if the relatively low absolute numbers of specific T cells captured is sufficient to permit robust phenotypic analyses.
2. Some of the staining patterns and associated gating strategies are not always convincing; this includes perforin (Figure S2) as well as CD95, delineation of Tfh by CXCR5/PD1 coexpression and Treg gating (Figure S4; also, what about FoxP3 stains for CD25+CD127- gated Treg vs. other CD4 T cells?). What are the respective positive/negative controls and exemplary staining patterns for all of the other phenotypic markers?
3. Provided that specific cell numbers and robust staining patterns permit downstream analyses, have the authors explored other relevant parameters such as, for example, acute T cell activation (HLADR, CD38, Ki67)? Given the properties of the staining panel, how is T cell exhaustion convincingly distinguished from activation (PD-1 and CTLA-4 are also expressed by activated T cells)? What about other “exhaustion markers” or associated transcription factors (e.g., TOX)? Do expression patterns of other important markers (CD28, CD57 etc.) offer relevant clues? How valuable are the CD62L stains, i.e. is recovery after thaw and rest sufficient to permit CD62L re-expression? The elevated frequencies of specific and total CD4 T cells expressing CXCR5+ in conjunction with CXCR4 or CCR6 in LC patients are intriguing; how does this relate to Tfh cells as defined by the authors (CXCR5+PD-1+)? And for that matter, is there a correlation between Tfh and sika RBD antibody (cf Figure 6)? In general, reliable detection of chemokine receptor expression may benefit from adjusted staining strategies Painter et al., PMID: 34453880); were these considered?
4. The apparent IL-6 production by specific CD4 T cells in some LC individual is rather curious (Figure S3C/D); a little more literature context about this “unusual” functional T cell capacity would be helpful.

5. MSI values in histograms figures 4, 5, S6 and S7: what is the unit of the x-axis values?
6. Also, please list metal conjugates for all CyTOF antibodies in Table S2 and specify conjugation methodology where applicable in Methods.

Both the PBMC RNAseq and plasma Olink analyses are appreciated but the discussion remains somewhat cursory and in the absence of further follow-up and/or correlation analyses with T cell and/or serology data their precise value is difficult to assess ...

The speculation about prolonged viral persistence in LC patients is sensible, so consider inclusion of the important reference Stein et al. PMID: 36517603 in discussion.

Reviewer #2 (Remarks to the Author):

Summary of the key results:

In this study, Roan and colleagues tackle an interesting research question; are there specific immunological parameters associated with long COVID (LC), especially for T cells?

For this, the authors analyzed blood samples from two groups of patients; a group of individuals who consistently reported at least one of 19 COVID19-attributed symptoms that developed or became worse at 8 months post their confirmed COVID19 infection (long COVID, LC group) n=27, compared to a non-LC, n=16, who reported none of the symptoms. The samples were collected at 8 months post-COVID.

The authors mainly focused on 4 parameters; 1- phenotyping T cells by CyTOF (total, and SARS-CoV-2-specific as identified by cytokine production in response to a pool of 28 SARS-CoV-2 peptides and an overlapping peptide pool spanning the entire spike protein), 2- bulk RNAseq for total PBMCs, 3- Serology of RBD-specific antibodies, and 4- OLINK analysis of serum proteins.

1- The authors results show that there were a few significant differences in the phenotype of SARS-CoV-2-specific T cells (specifically, CD4 T cells that are CXCR5+CCR6+ and CXCR5+CXCR4+, and CD8 T cells that are PD1+CTLA4+). Significant differences in the frequency of Tcm, Tfh, and Treg were observed, but only for total CD4 T cells.

2- Bulk RNAseq performed on total PBMCs showed significant differences in two genes; OR7D2 and ALAS2.

3- Serology revealed higher levels of SARS-CoV-2 RBD antibody levels in the LC group.

4- OLINK analysis of serum proteins showed elevated levels of some proteins in the LC group, e.g. proteins associated with inflammation, as well as a mis-coordinated Th2 response reflected by the elevated levels of IL4 and CCL22 while IL5 was diminished.

Originality and significance:

Several studies, cited by the authors, have tackled the same research question with more granular dissection of specific aspects of the immune response.

Data & methodology:

1- It is not clear to me why the authors chose to perform bulk RNAseq on total PBMCs which is much less informative than other techniques such as single-cell RNAseq. Performing scRNAseq even on a few patients from each group would have been more informative than bulk RNAseq on total PBMCs from 36 patients.

2- It is not discussed whether there were any specific comorbidities that correlated with LC.

3- Similarly, the authors did not examine or discuss the possibility of presence of autoantibodies in LC patients and their correlation with any of the characteristics they observed.

4- In their explanation of the criteria they used to define SARS-CoV-2-specific T cells, the authors mentioned that the differences between Unstimulated and Peptide-stimulated was not less than 1% (line 632). However, in almost all their representative dotplots, the differences between the Unstimulated and Peptide-stimulated is below this.

5- It is unclear to me whether the data for total CD4 in Figure 2A are from the unstim, stim, or calculated in another way.

- The tSNE in Fig5A is very vague, as the authors do not provide any explanation of what these clusters that are enriched in either groups correspond to. It would be clearer for potential readers if there is an additional tSNE showing the clusters occupying the different parts of the tSNE to appreciate the identity of the ones that are enriched in either group of patients.

- The Y-axis in all graphs describing phenotype of T cells is "% of CD4/CD8 T cells", whether it is for total or SARS-CoV-2-specific T cells.

- The table with patient data could be simplified to make it clearer for potential readers the distribution of the different characteristics among the two study groups. An example of information that I was unable to deduce from the table was the number of hospitalized in each group. I think having a Table 1 with just the last 2 columns having the distribution of all characteristics among the two study groups would make it simpler. The authors can then have the first 5 columns as a Table 2 if they think it is relevant to show.

- Despite staining for Foxp3, it does not seem that it was used for the gating strategy used to identify Tregs in Supplementary Figure S4

Appropriate use of statistics:

In several instances, the significant differences observed seem to be mainly driven by a few patients (e.g. Fig7A/OR7D2 and Supp. FigS3 D), or even one/two (e.g. Fig4C, Fig6A, and Fig7A/ALAS2). It would also be interesting to observe whether it is the same patient that is driving the significance in Fig4C for both graphs. This point is not discussed by the authors.

Conclusions robustness, validity, reliability:

The conclusions proposed by the authors are not fully supported by their data, since the majority of significant differences are driven by a few patient samples, sometimes 1-2 samples. Except for the OLINK results, the majority of data are not robust enough to support the conclusions.

Reviewer #3 (Remarks to the Author):

In their manuscript Kailin Lin et al. investigate changes in immunological parameters in Long Covid patients by using CyTOF, RNAseq and Olink. A small cohort of patients has been investigated and compared to non-Long covid patients. The patient cohort did not receive any vaccination prior to onset of Long Covid and was investigated at a late time point (8 months). The authors conclude that several immunological parameters are different on the Long Covid patients as compared to the control group. In particular, they note some changes in migration surface marker expression, exhaustion phenotype, antibody levels. This should indicate a breakdown on proper crosstalk of the humoral and cellular arms

of adaptive immunity. Overall, the study is very descriptive. The conclusions are extrapolated from analysis of marker expressions levels but lack functional and mechanistic validation. The patient cohorts are very small given that the authors inform on 10% Long Covid incidence following a SARS-Cov2 infection. Although Long Covid clearly is a very important and highly prevalent clinical manifestation and although there is a tremendous interest in the scientific community and general population to understand this pathology and to arrive at therapeutic solutions, the findings outlined in this manuscript remain vague and it they do not reveal robust novel diagnostic markers or disease activity parameters, nor do they provide specific information on how to therapeutically address this disease.

Several issues remain to be clarified.

Fig.S1B. The authors claim that individuals with LC were more likely to have been hospitalized during the acute phase of COVID-19. This conclusion is based on 7 patients having been hospitalized as compared to 20 in the LC group, versus 2 hospitalized compared to 14 in the non-LC group. These patient numbers are too low to arrive at this conclusion.

Fig.S1B. The authors claim that the number of LC symptoms was increasing between month 4 and month 8. The data shows, however, a decrease in symptom counts in many individuals and a few (3) strong outliers showing an increase in symptom counts. There is no detailed information on the exact statistical test available (matched pairs?).

Fig.1. The figure illustrates clearly the experimental approach. Additional information on timepoints could be added to highlight the late analysis timepoint. It is unusual to commit a whole figure for a graphical outline for the study design. This figure could be added to the subsequent data figure or to the supplementary figures.

Fig. 2. A gating strategy could be placed in the supplementary figures since it does not demonstrate any results. The "FACS" (Cytof) plots lack a scale (axes). It would be helpful to get information on whether percentages or MFI were provided. It is surprising that so few protein markers have been assessed/evaluated given the technical potential of CyTOF to measure many more parameters reported to be of relevance in COVID-19 or immune perturbations in general.

Fig.S3D. The authors claim that SARS.CO2 specific CD4+ T cells producing IL-6 may be specifically associated with a subset of individuals with LC. Given that IL-6 is not a T helper cell cytokine it would be recommended to assess IL-6 production also by ELISA and to ensure the purity of the CD4+ T cells (since also monocytes and other human APCs express CD4). The extreme low frequency of IL-6+ cells (low MFI) could also represent background staining. The statement that this finding could be specific for LC should therefore be deemphasized.

Fig. S6-7. The clustering analysis did not reveal any difference between the LC and control group. However, when the aspect of sex difference was considered, a few clusters revealed differences. This sex stratification is of general interest and an important aspect to be considered in clinical studies. Overall, the total patient numbers, however, seem too low to arrive at such a strong conclusion. It is also not clear to which extent the effects were driven by individual patients since too little information is provided on data analysis (concatenation?). The authors claim that profound changes in classical subset distribution have been found. However, looking at the statistical differences and the wide

interindividual distribution of the tested parameters, the changes are quite subtle and would require retesting in large cohorts.

Fig.5. It is not so clear why different analysis methods were used for different parameters (Fig. S6-7 vs. Fig. 4).

In T cells, CXCR4 is known to be constitutively expressed. The authors should comment on why they chose to assess the differential expression of this marker in T cells.

It is also recommended to explain the choice of markers for analysis. The chemokine receptors, which were chosen are not classic tissue-homing receptors, as claimed. To serve the purpose to investigate tissue-homing, chemokine receptors such as CCR9 (gut), CCR8 (lung) or the marker CLA for skin and others are recommended.

The markers PD-1 and CTLA-4 are known to be upregulated upon T cell activation as well as exhaustion. It is therefore recommended to phrase the conclusion that SARS-CoV2 specific CD8+ T cells in LC are exhausted with more caution. Additionally, the whole study is limited to the investigation of blood immune cells. A higher frequency of certain T cell subsets in the blood could indicate a lower frequency in the tissues (mobilization out of tissue), where the actual pathology takes place.

Fig. 7. It would be helpful to provide a heatmap that groups the LC and control patients separately to have easier visual access to the differences between both groups. This would still allow to see the interindividual differences.

The authors also discuss that the detection of SARS-Cov2 specific T cells 8 months after infection could be due to persistence of the virus. It would be very interesting to find out whether LC is indeed a smoldering ongoing infection. However, also uninfected healthy individuals have SARS-COV2 specific T cells as part of their natural TCR repertoire (found previously in PBMC obtained before the pandemic started). It would have also been helpful for the interpretation of the data if the authors would have used a non-SARS-Cov2 control peptide in addition to no peptide.

Author Rebuttal to Initial comments

Dear Dr. Visan,

Thank you for the opportunity to resubmit. As requested, we have prepared below a point-by-point response. This incorporates all the points had listed in your summary of the components requested for the resubmission, including showing the entirety of the analysis (inclusive of all the markers that did not show differences), doing FACS-based validations, and performing scRNAseq studies. Please note that line numbers below refer to that in the “clean” (non-tracked) version of the revised manuscript.

Reviewer #1

(Remarks to the Author)

In their manuscript, Yin et al. seek to define immunological PASC by deploying deep immune profiling of T cells, anti-viral serology, RNAseq of PBMC and plasma profiling for inflammation-associated soluble factors (Olink) by comparing cohorts of individuals with and without “long-

COVID” (LC and non-LC) eight months after acute SARS-CoV-2 infection and prior to SARS-CoV-2 vaccination. The cohort selection, despite its small size, constitutes a particular strength of the work as does the focus on SARS-CoV-2 specific T cell immunity as well as a balanced discussion of their specific findings in the context of relevant prior literature. Nevertheless, and although the authors generously emphasize that LC is associated “with T cell dysregulation, inflammation and an uncoordinated adaptive immune response to SARS-CoV-2”, the reported differences, although intriguing, would appear to be somewhat modest; furthermore, it is not clear if they sufficiently leveraged the granularity of their data sets and the potential for integrated data analyses.

We thank the reviewer for recognizing the strengths of our study. As we have detailed further below, we had analyzed all markers in our panel using multiple integrative analytical approaches, and reported multiple biologically-relevant important, statistically significant results. In our revision, we have better highlighted how we had leveraged the granularity of our datasets by including all our analyses, including negative findings, as requested by the editor. New figures showing these additional analyses of our original datasets are the new Figures S6, S7, S8, S9, and S10. We also note that in response to a request from Reviewer #2 to also generate scRNAseq data, we have added these additional granular data to our study.

Much of the analytical effort is devoted to the characterization of T cell properties in LC and non-LC individuals (4 out of 6 main data figures, 6 out of 7 supplementary data figures) and the use of short-term peptide stimulation in conjunction with inducible T cell effector functions allows for better capture and characterization of specific T cell populations as compared to the frequently used detection of AIMs following longer stimulation cultures. The combinatorial analyses of specific CD4 and CD8 T cells (via induced IFN γ , TNF α , IL-2 or IFN γ , TNF α , MIP1b, respectively) and designation of suitable denominators appears convincing (Figure 2) though additional readouts (eg, CD40L for CD4 T cells) would have been helpful.

We have previously found that activation induced markers (AIM, e.g., CD40L, Ox40, CD69, CD38) do not effectively identify antigen-specific T cells after a 6-hour stimulation with peptide (longer stimulations, e.g., 24 hours, are required). An example of a COVID-19 study where we formally demonstrated this was PMID 34260965. Because a main goal of this study was immunophenotyping, we used the 6-hour instead of 24-hour stimulation to better preserve the original phenotypes of the cells. Furthermore, these were the best conditions enabling us to assess effector function (e.g., cytokine secretion). The notion that AIM could not be used to specifically identify SARS-CoV-2-specific T cells in our system, and the citation demonstrating this to be the case, has now been added to the revision (Lines 794-796).

The LAMP1 analyses, however, are confusing. To assess degranulation of specific T cells, LAMP1 antibodies need to be included in the stimulation culture; in contrast, the value of intracellular LAMP1 staining after stimulation as performed here remains unclear.

LAMP1 analysis by FACS indeed typically involves addition of LAMP1 antibodies during the stimulation. This is not an assay established for CyTOF, where metal-conjugated antibodies are used. We were concerned that metal-conjugated antibodies may interfere with the stimulation

process, and hence only added the antibody during intracellular staining (as a side note however, as a methods study, we are currently developing an approach to add LAMP1 antibodies during stimulation in the context of CyTOF). We acknowledge that our use of LAMP1 in our panel is not the way it is typically used by FACS, but also note that it was not used in our system to define antigen-specific T cells. In our case, we used LAMP1 staining simply to assess the expression levels of this antigen on our cells of interest, rather than as a degranulation marker (and accordingly, we ensure that we never state that LAMP1 was being used as a degranulation marker in our study). We have now clarified in our manuscript that LAMP1 was not used as a degranulation marker, but rather just to monitor intracellular expression levels of this protein (Lines 776-778). If the reviewer feels strongly we should instead remove all LAMP1 data from our study, we would be happy to do so; it does not make a difference in any of our conclusions.

The deep immune profiling of SARS-CoV-2 specific T cells in LC and non-LC individuals as afforded by the authors' CyTOF analyses (Figures 3B, 4 and 5) offers a unique opportunity to define and elucidate potential differences yet there remain several concerns:

1. Unclear if the relatively low absolute numbers of specific T cells captured is sufficient to permit robust phenotypic analyses.

Total SARS-CoV-2-specific T cells were detected at a median of 163 cells (134 for CD4+ T cells, 29 for CD8+ T cells) per donor, with the mean being 221.7 cells (185.2 for CD4+ T cells, 36.5 for CD8+ T cells) per donor. This is sufficient to permit robust phenotypic analyses, and in fact is markedly higher numbers than our prior COVID-19 CyTOF studies where lower numbers of SARS-CoV-2-specific T cells were analyzed (PMIDs 35584773, 34636722, 34389625, 34260965, 32839763). We have now provided this additional information in the revised manuscript (Lines 796-798).

2. Some of the staining patterns and associated gating strategies are not always convincing;

In our original submission, we did not show all our validation plots as these have previously been shown in our prior CyTOF publications. In the revision, we now not only cite these publications (Lines 755-756), but also have added a new figure (Fig. S18) showing specific validations of antibody lots used in the current study. This includes our typical validation of differential antigen expression among T vs. other immune subsets in a tonsil sample (since many of our antigens are differentially between tonsillar T and B cells), as well as use of PMA/ionomycin-stimulated and LPS-stimulated PBMCs to validate staining for effector cytokines and cytolytic molecules. We address the specific concerns of individual antigens below:

this includes perforin (Figure S2)

As shown in the new Fig. S18 and depicted below, we saw clear populations of both T cells (CD3+) and non-T cells staining positive for perforin in our PMA/ionomycin-stimulated PBMCs.

Of note, the perforin staining in the original Fig. S2 did not look robust because SARS-CoV-2-specific T cells did not specifically induce large amounts of perforin in response to cognate peptide stimulation. We now show in the revised manuscript the validation data that clearly shows that our perforin staining was robust (Fig. S18B).

as well as CD95,

The reviewer may be bringing up a concern for CD95 because it does not stain as strongly as is observed in the more commonly used FACS assay. In fact, the staining of CD95 is similar to our prior COVID-19 CyTOF studies, where this antibody has been validated (PMIDs 34636722, 34389625, 34260965, 32839763). We now clarify our validation methods and cite these prior studies (Lines 755-759).

delineation of Tfh by CXCR5/PD1 coexpression

Staining for CXCR5 and PD1 were defined through approaches we have previously implemented in our prior COVID-19 CyTOF-based studies wherein we characterized peripheral Tfh (PMIDs 34636722, 34389625, 34260965, 32839763). For establishing the Tfh gate, we use an aliquot of a “bridge” sample (generated from multiple donors, and where we can identify a PD1^{high}CXCR5^{high} population) that is run within every CyTOF run. This bridge sample is also used for batch normalization between different CyTOF runs. As part of the new antibody validation figure (Fig. S18), we now show how we had established the Tfh gate using this sample:

We also note that we established this Tfh gate to be similar to our prior COVID-19 publications (PMIDs 34636722, 34389625, 34260965, 32839763) in order that frequencies of Tfh cells can be more directly compared between these different studies.

and Treg gating (Figure S4; also, what about FoxP3 stains for CD25+CD127- gated Treg vs. other CD4 T cells?).

Foxp3 staining by CyTOF is typically weak, so we do not use it to actually identify Tregs. Instead, we use the more easily gated population of memory CD4+ T cells that are CD25+CD127- to identify these cells. Importantly, we have demonstrated that this method of gating markedly enriches for cells expressing higher levels of Foxp3 (relative to control naïve CD4+ T cells), validating this method of Treg identification (Fig. S18C, and copied below):

What are the respective positive/negative controls and exemplary staining patterns for all of the other phenotypic markers?

Markers are validated in CyTOF studies by demonstrating that expression patterns are as expected among immune subsets known to differentially express markers in the panel. This was detailed in the first CyTOF paper we had published back in 2017 (PMID 28746881), where we validated antigens from the panel by comparing expression levels on different subsets of T cells

(e.g., naïve T cells and memory T cells), and between B cells and T cells (because many markers in our panel are differentially expressed between these two adaptive immune subsets). Additional markers incorporated into our panel for the current study were used and validated in subsequent publications. We have an extensive list of publications using the CyTOF antibodies implemented in this study, which have been validated: PMIDs 28882052, 32452381, 32839763, 32990219, 33115867, 33910003, 34235864, 34260965, 34389625, 34636722, 34899645, 35154118, 35296537, 35784348, 35787792, 36073812. We have now clarified this in the Methods section (Lines 755-772), cited a subset of these prior publications, and included the new supplementary figure Fig. S18 showing validations.

3. Provided that specific cell numbers and robust staining patterns permit downstream analyses, have the authors explored other relevant parameters such as, for example, acute T cell activation (HLADR, CD38, Ki67)?

We had examined all parameters (including the above-listed acute activation markers) from our CyTOF datasets for differences between LC and control patients. We had not observed significant differences between HLADR, CD38, Ki67, and hence had not reported these data. Per the reviewer's suggestion, we now formally report that the expression levels of HLADR, CD38, and Ki67 are not significantly different between the LC and non-LC individuals. This was true among CD4+ T cells, CD8+ T cells, SARS-CoV-2-specific CD4+ T cells, and SARS-CoV-2-specific CD8+ T cells from LC vs. non-LC individuals, which are shown within the new Fig. S6-S9 (wherein we show mean signal intensity of not just these 3 antigens but also all the CyTOF phenotyping markers from our panel), and summarized below:

To further hit home the point that acute T cell activation levels are not different between the individuals with vs. without LC, we additionally performed manual gating for T cells expressing HLADR, CD38, or Ki67, or all 3 of these antigens. Consistent with the mean expression data, we also found no significant differences between the Non-LC and LC groups, data which are now presented as the new Fig. S10 and copied below:

Given the properties of the staining panel, how is T cell exhaustion convincingly distinguished from activation (PD-1 and CTLA-4 are also expressed by activated T cells)?

As detailed in the point immediately above and now shown in the new Fig. S10D, we found the percentages of activated T cells (based on expression of CD38, HLADR, and/or Ki67) among

SARS-CoV-2-specific CD8+ T cells to not be significantly different between those with vs. without LC, in contrast to SARS-CoV-2-specific T cells co-expressing PD1 and CTLA4 which were significantly higher in those with LC (Fig. 5C). Therefore, the difference we observed indeed appears to be more reminiscent of exhaustion rather than overall activation.

What about other “exhaustion markers” or associated transcription factors (e.g., TOX)?

TOX was not on our CyTOF panel. We had included three exhaustion markers (PD1, CTLA4, and TIGIT) in our panel, which when we initiated our study we felt was sufficient to examine exhaustion. Inclusion of TOX in future studies would be interesting in future studies given the exhaustion phenotype that emerged from our current study.

Do expression patterns of other important markers (CD28, CD57 etc.) offer relevant clues?

CD28 and CD57 did not reveal significant differences in expression levels between Non-LC and LC participants:

These data are presented in the new Fig. S6-S9, wherein we show mean expression intensities of not just these 2 antigens but also all the CyTOF phenotyping markers from our panel.

How valuable are the CD62L stains, i.e. is recovery after thaw and rest sufficient to permit CD62L re-expression?

It is standard to culture cryopreserved cells overnight to allow for recovery of CD62L expression (classic methods paper describing this is PMID 12957403). Of note, in our prior HIV and COVID-19 CyTOF studies (see extensive list of PMIDs listed earlier), we have stained for CD62L including on cryopreserved cells and saw good signal.

Validation of the CD62L antibody lot used in this particular study are shown below, where a distinct population of CD62L+ cells (red arrow) can be observed among T cells (CD3+):

We now show these validation data in the new Fig. S18, and additionally cite the methods paper discussing CD62L recovery following overnight culture (Line 694).

The elevated frequencies of specific and total CD4 T cells expressing CXCR5+ in conjunction with CXCR4 or CCR6 in LC patients are intriguing; how does this relate to Tfh cells as defined by the authors (CXCR5+PD-1+)?

We thank the reviewer for this suggested analysis. When we performed a correlation analysis between SARS-CoV-2-specific CXCR4+CXCR5+ or CXCR5+CCR6+ CD4+ T cells vs. SARS-CoV-2-specific Tfh cells, we observed a highly statistically significant positive association only in the individuals with LC:

When we performed a correlation analysis between *total* CXCR4+CXCR5+ or CXCR5+CCR6+ CD4+ T cells vs. *total* Tfh cells, we observed a similar phenomenon, although in the CXCR4+CXCR5+ vs. Tfh non-LC group, the association was significant, albeit barely ($p=0.02$):

Together, these results suggest that there are unique correlations between CD4+ T cell subsets – particularly among SARS-CoV-2-specific CD4+ T cells – in the LC group that are not observed in non-LC individuals. These new analyses have now been integrated into Fig. 4 (the new panels E and F).

And for that matter, is there a correlation between Tfh and sike RBD antibody (cf Figure 6)?

We also thank the Reviewer for this excellent suggestion to look for a correlation between Tfh and RBD antibody levels. When looking at *total* Tfh vs. RBD antibody levels, we did not observe a significant correlation (not shown). Very interestingly, however, we *did* observe a significant

correlation between SARS-CoV-2-specific Tfh and RBD antibody levels in the control group only, and not the LC group:

These results further support our conclusion of a dis-coordination between the cellular and humoral arms of adaptive immunity in LC, and have been added to Fig. 6E.

In general, reliable detection of chemokine receptor expression may benefit from adjusted staining strategies Painter et al., PMID: 34453880); were these considered?

Chemokine receptors have been a part of our CyTOF panels ever since our first published CyTOF study (PMID 28746881), and we have always used a CyTOF staining protocol adapted from Eugene Butcher (expert in chemokine receptor research) at Stanford University, in which special care was made to ensure that the protocol would not interfere with detection of these sometimes difficult-to-stain receptors. This included avoiding EDTA during the CyTOF staining process, as well as other precautions. We note that to date we have published 18 CyTOF papers (PMIDs 28746881, 28882052, 32452381, 32839763, 32990219, 33115867, 33910003, 34235864, 34260965, 34389625, 34636722, 34899645, 35154118, 35296537, 35784348, 35787792, 36073812, 35584773), and all of them have incorporated staining for multiple chemokine receptors. We have now added in some of these details with regards to the precautions we took to ensure proper staining of chemokine receptors, and have cited a subset of these prior papers (Lines 711-713).

To further establish that we are reliably detecting chemokine receptor expression, for the revision we have taken one additional aliquot of all specimens for which aliquots remained (25 out of 27 for LC, 15 out of 16 for non-LC), and performed FACS validation experiments, staining for CXCR4, CXCR5, and CCR6, the 3 chemokine receptors which we found to be elevated in expression on CD4+ T cells from individuals with LC (Fig. 4). For this, we established and validated a FACS panel to assess memory CD4+ T cells for expression levels of these 3 chemokine receptors (the new Fig. S13B):

We further confirmed that the expression levels of all 3 chemokine receptors as measured by FACS highly correlated with that as measured by CyTOF (the new Fig. S13A):

Finally, we demonstrated that the FACS data recapitulated our main CD4+ T cell finding from the CyTOF analyses, that CXCR4+CXCR5+ and CXCR5+CCR6+ CD4+ T cells are significantly elevated in individuals with LC (the new Fig. S13C):

Interestingly, while the population of CXCR4+CCR6+ CD4+ T cells only trended higher in individuals with LC as assessed by CyTOF (Fig. 4), this difference became statistically significant when assessed by FACS (the new Fig. S13C):

These FACS validation data have now all been incorporated into the results section of the revised manuscript (Lines 250-254).

In conclusion, our prior extensive experience with chemokine receptor staining, together with these new FACS experiments validating the CyTOF chemokine receptor data, make us very confident that what we are reporting is real and not artificial.

4. The apparent IL-6 production by specific CD4 T cells in some LC individual is rather curious (Figure S3C/D); a little more literature context about this “unusual” functional T cell capacity would be helpful.

We find it interesting that IL6 is one of the most commonly cited inflammatory markers in LC. We have recently shown in our LIINC cohort (the parent cohort of the current study) that plasma IL6 is elevated in individuals with LC compared to individuals with full recovery, and that these relationships are maintained or strengthened in individuals with cardiac and neurocognitive symptoms in comparison to those without those symptoms or who report full recovery (PMIDs 34677601, 35389890, 35701186). With regards to IL6 production by T cells, it is something we have previously observed in the context of severe COVID-19 (PMID 34260965). Interestingly, IL6 production from “atypical” cells has also recently been observed in the context of HIV, where IL6 was produced by B cells (PMID 29104574). In our revised manuscript, we have added these points and additional literature citations to the Discussion (Lines 484-489).

5. MSI values in histograms figures 4, 5, S6 and S7: what is the unit of the x-axis values?

We apologize for the ambiguity. The MSI values in the histograms corresponded to mean signal intensity of arcsinh-transformed data, with the transformation implemented as detailed in our prior publication (PMID 28746881). We have now indicated this in the methods and figure legends (Lines 744 and 1367) and cited this prior publication.

6. Also, please list metal conjugates for all CyTOF antibodies in Table S2 and specify conjugation methodology where applicable in Methods.

This additional metal conjugates information has now been incorporated into the updated Table S2, and a description of the conjugation methodology is now described is now included in the Methods (Lines 669-689).

Both the PBMC RNAseq and plasma Olink analyses are appreciated but the discussion remains somewhat cursory and in the absence of further follow-up and/or correlation analyses with T cell and/or serology data their precise value is difficult to assess ...

We believe that the high-dimensional datasets generated using these two “omics” tools were interesting in their own right, as they revealed important differences between LC and control patients as detailed in the paper. We do however agree with the reviewer that associating these data with the serological / T cell findings could reveal additional interesting findings. Therefore, we looked for associations between our most interesting T cell / serological findings (antibody levels, % SARS-CoV-2-specific CD4+ T cells, % SARS-CoV-2-specific CD8+ T cells, %CXCR4+CXCR5+ total CD4+ T cells, %CXCR5+CCR6+ total CD4+ T cells, %PD1+CTLA4+ SARS-CoV-2-specific CD8+ T cells) with our most interesting RNAseq (expression levels of OR7D2, ALAS2, BIRC5) and Olink (expression levels of IL4, IL5, and CCL22) findings. While most associations were not statistically significant, we found the % of CXCR4+CXCR5+ CD4+ T cells and CXCR5+CCR6+ CD4+ T cells (both of which are increased in LC individuals, Fig. 4) positively and significantly associated with levels of IL4 only in LC individuals, while in control participants we observed the opposite pattern of statistically significant negative associations:

These data were interesting in light of the discussion of dysregulated IL4-driven Th2 responses, and have now been incorporated into the manuscript as the new Fig. S17.

The speculation about prolonged viral persistence in LC patients is sensible, so consider inclusion of the important reference Stein et al. PMID: 36517603 in discussion.

We thank the reviewer for this suggestion and have now cited the study both in the Introduction and Discussion (Lines 57-59 and 433-436).

Reviewer #2

(Remarks to the Author)

Summary of the key results:

In this study, Roan and colleagues tackle an interesting research question; are there specific immunological parameters associated with long COVID (LC), especially for T cells?

For this, the authors analyzed blood samples from two groups of patients; a group of individuals who consistently reported at least one of 19 COVID19-attributed symptoms that developed or became worse at 8 months post their confirmed COVID19 infection (long COVID, LC group) n=27, compared to a non-LC, n=16, who reported none of the symptoms. The samples were collected at 8 months post-COVID.

The authors mainly focused on 4 parameters; 1- phenotyping T cells by CyTOF (total, and SARS-CoV-2-specific as identified by cytokine production in response to a pool of 28 SARS-CoV-2 peptides and an overlapping peptide pool spanning the entire spike protein), 2- bulk RNAseq for total PBMCs, 3- Serology of RBD-specific antibodies, and 4- OLINK analysis of serum proteins.

1- The authors results show that there were a few significant differences in the phenotype of SARS-CoV-2-specific T cells (specifically, CD4 T cells that are CXCR5+CCR6+ and CXCR5+CXCR4+, and CD8 T cells that are PD1+CTLA4+). Significant differences in the frequency of Tcm, Tfh, and Treg were observed, but only for total CD4 T cells.

2- Bulk RNAseq performed on total PBMCs showed significant differences in two genes; OR7D2 and ALAS2.

3- Serology revealed higher levels of SARS-CoV-2 RBD antibody levels in the LC group.

4- OLINK analysis of serum proteins showed elevated levels of some proteins in the LC group, e.g. proteins associated with inflammation, as well as a mis-coordinated Th2 response reflected by the elevated levels of IL4 and CCL22 while IL5 was diminished.

Originality and significance:

Several studies, cited by the authors, have tackled the same research question with more granular dissection of specific aspects of the immune response.

We note that the strengths of our study over prior ones are 1) our well-annotated cohort and inclusion of Long COVID (LC) and non-LC patients with clear and consistent clinical trajectories and who hadn't been vaccinated nor re-infected prior to the collection of the biological specimen, 2) use of the same specimens in 6 sets of assays (including the scRNAseq studies requested by this Reviewer; see details in Figure 1), and 3) a deep focus on the phenotypes, effector functions, and differentiation states of both total and SARS-CoV-2-specific T cells. We are not aware of a study that has gone into the level of granularity that we have in the context of T cell responses during LC. This had been detailed in Lines 81-91 of the original manuscript, and now where possible has been emphasized further in the revised manuscript.

Data & methodology:

1- It is not clear to me why the authors chose to perform bulk RNAseq on total PBMCs which is much less informative than other techniques such as single-cell RNAseq. Performing scRNAseq even on a few patients from each group would have been more informative than bulk RNAseq on total PBMCs from 36 patients.

We agree that scRNAseq advantages, the obvious being that it enables analysis at the single-cell level. At the same time, it also has some downsides relative to bulk RNAseq. Compared to bulk RNAseq, scRNAseq is substantially more expensive, analyzes fewer cells (typically <10,000 per sample even for droplet-based approaches) resulting in inability to detect differentially expressed genes in rare subsets of cells (e.g., those that wouldn't be captured among 10,000 cells), and is less optimal at detecting some rare transcripts due to the sparsity of the data. Importantly, we observed intriguing important differences (e.g., differences in odorant, heme regulation, immunoglobulin gene expression) with our bulk RNAseq datasets.

Although it is not feasible to run scRNAseq on all specimens (due to both financial and specimen availability issues), in response to the reviewer's comment we have now performed scRNAseq on a total of 12 of the patients from our cohort. This number was more than that suggested by the Reviewer, and was a number the Editor deemed reasonable given costs (reagents and sequencing alone for these samples amounted to ~\$40,000). For these scRNAseq studies, we used samples exactly matching the timepoints used in the bulk RNAseq, Olink, serology, and T cell datasets generated in the original manuscript. We chose for these scRNAseq studies to analyze the LC donors exhibiting highest OR7D2 expression and highest ALAS2 (the two genes most significantly upregulated in the LC as compared to non-LC group as assessed by bulk RNAseq, Fig. 7A), and compared them to 4 non-LC donors. Among the LC patients, the top OR7D2 expressors were distinct from the top ALAS2. Interestingly, the top four LC OR7D2 expressors were all female, and of the top 6 LC ALAS2 expressors 5 were female (for the latter, the 3rd top expressor out of the 6 was the male). This observation was interesting in its own right given the association of LC with female sex. But from a technical standpoint, to avoid sex-associated confounders in interpreting our scRNAseq data, we chose only female specimens from this analysis. This corresponded to the top four LC OR7D2 expressors (which were all female) and the 1st, 2nd, 4th, and 5th highest LC expressors of ALAS2. This was compared to four female non-LC controls randomly selected from our non-LC collection.

Integrative analysis of all 12 donors (n=4 non-LC, n=4 OR7D2^{high} LC, n=4 ALAS2^{high} LC) analyzed by scRNAseq identified 11 clusters of cells, including subsets of CD4+ T cells, CD8+ T cells, B cells, monocytes, and innate-like cells. In addition, small numbers of granulocytes and platelets were identified; their low numbers were expected as PBMCs and not whole blood were analyzed (the new Fig. 7D):

While the major clusters did not differ in frequency between the LC and non-LC groups, the granulocyte cluster was significantly less abundant ($p=0.006$) while the platelet cluster was significantly more abundant ($p=0.01$) in individuals with LC. Consistent with lack of major perturbations at the transcriptional level, visualization of the datasets by LC vs. non-LC status, and by the two groups of LC ($OR7D2^{high}$ and $ALAS2^{high}$) vs. non-LC status, did not reveal profound differences (the new Fig. S16A, B):

Consistent with the bulk RNAseq data, by scRNAseq, $OR7D2$ was expressed at the highest levels in the $OR7D2^{high}$ LC specimens and $ALAS2$ was expressed at the highest levels in the $ALAS2^{high}$ LC specimens (the new Fig. S16C):

When we assessed which clusters of cells expressed OR7D2 and ALAS2, we found that these transcripts were broadly expressed in all subsets except the granulocyte and platelet clusters, with the caveat that the low numbers of cells in the latter two clusters likely precluded ability to detect these transcripts (the new Fig. 7E and S16D):

Interestingly, among clusters expressing OR7D2 or ALAS2, only a small fraction of cells (<0.5%) expressed these genes, suggesting that the small numbers of cells harboring high levels of OR7D2/ALAS2 were responsible for the original identification of these DEGs from the bulk RNAseq analysis. We also found that clusters expressing OR7D2 generally expressed this gene at the highest levels in the OR7D2^{high} LC group, and that clusters expressing ALAS2

generally expressed this gene at the highest levels in the ALAS2^{high} LC group (the new Fig. S16E and F):

These scRNAseq-based analyses of subset distribution and OR7D2 and ALAS2 are now detailed in the revision (Lines 323-352). We additionally took advantage of these new scRNAseq results to interrogate cluster-specific gene expression patterns differing between our LC vs. non-LC samples, as well as between our two groups of LC samples (OR7D2^{high} and ALAS2^{high}) vs. non-LC samples. These data are presented in the Results section (Lines 352-365) and as the new Tables S5-S7, and revealed a number interesting DEGs among subsets of cells, including ones associated immune dysregulation. We have also made these new scRNAseq datasets publicly available for further data mining by the community, as detailed in the Methods section.

Together, these scRNAseq validate the bulk RNAseq data, and provide a more granular view of expression of OR7D2, ALAS2, and other transcripts associated with LC.

2- It is not discussed whether there were any specific comorbidities that correlated with LC.

For this study, we have used the term LC to refer specifically to unexplained symptoms following SARS-CoV-2 infection, that are new or worse since before someone had COVID-19 and cannot be attributed to another cause. In general, comorbidities like obesity and hypertension commonly associate with LC, and this is true in our larger LIINC cohort, information we have now added to the revised manuscript (Lines 629-633).

We have also analyzed comorbidity data from the selected participants used in the current study. As shown in the new Fig. S1D and below, most participants were generally healthy prior to COVID and few exhibited pre-COVID comorbidities. (Of note, for consistency, the original plots depicting the sex distribution and hospitalization rates of the participants – Fig. S1A and B in the original submission– were re-plotted in a different fashion – now Fig. S1B and C

respectively in the revision – to match the format and color scheme of these co-morbidity analyses.) The most common comorbidities were pre-existing hypertension (6/7 of those with prior hypertension were in the LC group) and lung disease (mainly asthma; 6/7 again in the LC group):

We also inquire at each visit regarding incident comorbid diagnoses. None of our participants reported such diagnoses during the follow-up period. Interestingly, unlike the comorbidities listed above, the individuals with LC had a significantly higher BMI, data which have been now incorporated as the new Fig. S1E, and copied below:

Overall, these results are consistent with what is known about risk factors for LC in the larger population – that it is more common in women than men, in those who had more severe acute illness (e.g., requiring hospitalization), and that certain comorbidities (especially obesity) are common risk factors. We believe that the general lack of other major pre-existing comorbidities in the population selected for this analysis is also a strength, as many prior reports of LC have studied populations with much poorer overall health and/or a large number of comorbidities, that is unlikely to be representative of the vast majority of individuals experiencing LC.

3- Similarly, the authors did not examine or discuss the possibility of presence of autoantibodies in LC patients and their correlation with any of the characteristics they observed.

We have separately performed a larger analysis using PHIP-seq (a method for proteome-wide autoantibody detection) in our cohort (just published as PMID 37288661), which asked complementary questions. While we found a clear autoreactivity signature associated with prior COVID (compared with pre-pandemic controls) we did not identify a unique autoreactivity signature comparing people with and without LC. This is consistent with another recent study of LC focused on unexplained symptoms (preprint PMID 35982667) which used REAP (a complementary technology). Therefore, while there appears to be in general an increased incidence of clinical autoimmune conditions following SARS-CoV-2 infection, we believe that the data on autoreactivity causing the unexplained symptoms of LC are more limited. In the revised manuscript, we have now cited and discussed this complementary study (Lines 68-70).

4- In their explanation of the criteria they used to define SARS-CoV-2-specific T cells, the authors mentioned that the differences between Unstimulated and Peptide-stimulated was not less than 1% (line 632). However, in almost all their representative dotplots, the differences between the Unstimulated and Peptide-stimulated is below this.

We apologize that we had made a typo in line 632 of our original manuscript, and had meant to state 0.01% instead of 1% (the mistake came about because we forgot to move the decimal two places over when converting between frequencies and percentages). This typo has now been corrected (Line 786 of revised manuscript).

5- It is unclear to me whether the data for total CD4 in Figure 2A are from the unstim, stim, or calculated in another way.

This was from the unstimulated condition (similar results were observed for the stimulated condition). This has now been clarified within the figure legend, which of note has now been moved to supplement (and is now Fig. S2A) at the request of Reviewer #3. For completeness, we have also added the corresponding gating strategy plots for a stimulated sample (the new Fig. S2B).

- The tSNE in Fig5A is very vague, as the authors do not provide any explanation of what these clusters that are enriched in either groups correspond to. It would be clearer for potential readers if there is an additional tSNE showing the clusters occupying the different parts of the tSNE to appreciate the identity of the ones that are enriched in either group of patients.

We apologize for the confusion. Fig. 5A (and the analogous Fig. 4A for SARS-CoV-2-specific CD4+ T cells) did not depict any clusters, but rather was highlighting the notion that there were global phenotypic differences between SARS-CoV-2-specific CD8+ T cells from LC vs. Non-LC (as demonstrated by localization of cells in different regions of the tSNE), which we subsequently detail in the rest of the figure. We have previously used these types of contour tSNEs to help visualize differences in SARS-CoV-2-specific T cells in individuals with vaccine-

only as compared to “hybrid” (infection+vaccine) immunity (PMID 34636722). Of note, the tSNE is simply a visualization mechanism, not a clustering analysis.

We also note that our clustering analysis of our CyTOF datasets did not reveal any significant differences among SARS-CoV-2-specific CD4+ or CD8+ T cells, and the cluster distributions of these cells looked similar between the two patient groups; this may be in part because only 3 clusters were found for each of these populations (C1, C2, and C3 for SARS-CoV-2-specific CD4+ T cells, and D1, D2, and D3 for SARS-CoV-2-specific CD8+ T cells):

It was because the contour visualizations suggested differential phenotypic features of the cells, that we decided to manually dig into the datasets for differentially expressed patterns, which led to the findings reported in Fig. 4 and Fig. 5. In the revised manuscript, we have indicated that cluster distributions were not significantly different between the individuals with vs. without LC (Lines 241-244), and followed that up with the data that led to our identification of the significant differences reported in these two figures.

- The Y-axis in all graphs describing phenotype of T cells is “% of CD4/CD8 T cells”, whether it is for total or SARS-CoV-2-specific T cells.

Although we had indicated in the figure legends whether the data were referring to total or SARS-CoV-2-specific T cells, we agree with the Reviewer that it would be clearer to also include this information on the y-axes of the plots. We have now added this information to all the relevant graphs.

- The table with patient data could be simplified to make it clearer for potential readers the distribution of the different characteristics among the two study groups. An example of information that I was unable to deduce from the table was the number of hospitalized in each group. I think having a Table 1 with just the last 2 columns having the distribution of all characteristics among the two study groups would make it simpler. The authors can then have the first 5 columns as a Table 2 if they think it is relevant to show.

We thank the reviewer for this suggestion, and have now split the patient data table into two separate tables (Tables 1 and 2) as the Reviewer suggests.

- Despite staining for Foxp3, it does not seem that it was used for the gating strategy used to

identify Tregs in Supplementary Figure S4

As also detailed in our response to Reviewer #1 above, due to the weakness of Foxp3 staining by CyTOF, we didn't use it to actually identify Tregs (since gating on a distinct population would be difficult). Instead, we use the more easily gated population of memory CD4+ T cells that are CD25+CD127- (a standard approach) to identify Tregs. Importantly, we show that this gated population of cells expresses high levels of Foxp3, validating their Treg identity:

In the above plot, Foxp3 expression is clearly expressed at significantly higher levels among Tregs as compared to control naive CD4+ T cells, as expected. We have now included these data in our new Fig. S18 that shows validation analyses of antibodies from our CyTOF panel.

Appropriate use of statistics:

In several instances, the significant differences observed seem to be mainly driven by a few patients (e.g. Fig7A/OR7D2 and Supp. FigS3 D),

We note that the Fig. 7A/OR7D2 was not driven by a few patients (9 patients had high OR7D2 expression levels) and these results was statistically significant after multiple correction in the RNAseq analysis:

Fig. S3D was also not driven by a few patients (6 patients in this case exhibited the phenotype of interest):

or even one/two (e.g. Fig4C, Fig6A, and Fig7A/ALAS2). It would also be interesting to observe whether it is the same patient that is driving the significance in Fig4C for both graphs. This point is not discussed by the authors.

In Fig. 4C, indeed there is one outlier participant (green arrows) in the LC group, but we note that the other datapoints in the LC group were also higher than in the non-LC group:

When we manually remove this single outlier participant (which in this case is the same participant in the two plots), we still see observe significant findings for the CXCR5+CCR6+ data, and a near-significant trend ($p=0.0589$) for the CXCR4+CXCR5+ data:

As in general we are not comfortable selectively removing datapoints from our analyses, but instead just relying on statistics to inform on whether the data are believable, we are opting to

keep our original data. However, if the editor and reviewer can suggest a good way to report on data where we selectively remove datapoints, we can do so.

As the Reviewer also mentioned Fig. 6A and 7A, we examine those figure panels closely below:

With regards to Fig. 6A, there were two outlier LC participants which had very high antibody levels, but we also note that there were many more datapoints in the intermediate range (region highlighted by the green line, between 1500-3000 units) in the LC as compared to the non-LC group:

With regards to the ALAS2 data for Fig. 7A, we would like to point out that 8 LC participants exhibited relatively high expression levels of ALAS2, versus zero in the non-LC group:

In conclusion, our reported results were not based on simply one, two, or several outlier points. This, but more importantly our having used proper statistical tests to make our conclusions, make us confident that we are reporting are real and not simply noise. On a broader note, we note that LC is a heterogeneous disease and that perhaps it is in fact expected that differences can be driven by small numbers of patients, and we believe that in itself is of important interest.

Conclusions robustness, validity, reliability:

The conclusions proposed by the authors are not fully supported by their data, since the majority of significant differences are driven by a few patient samples, sometimes 1-2 samples. Except for the OLINK results, the majority of data are not robust enough to support the conclusions.

As detailed above, we used legitimate statistical analyses in our study, and all conclusions we drew were based on statistically significant results. As detailed above, the data were not driven by just one or a few outliers. Furthermore, outliers were different in the different analyses. For example, we had demonstrated in Fig. 6B that the two LC participants with highest levels of SARS-CoV-2-specific CD8+ T cell exhaustion were (purple circle) were different from those with the highest antibody levels (green oval):

Intriguingly, this was also the case in our other datasets; for example, the four individuals with highest expression levels of OR7D2 were different from the four individuals with the highest levels of ALAS2, an aspect we had not stated in the original manuscript but which we have now added to the revision (Lines 309-311).

Finally, as mentioned earlier, we note that LC is a heterogeneous disease and that the notion that differences can be driven by small numbers of patients is not surprising, and is informative for our understanding of the disease. We have tried to hit home this point by stating in the Limitations section: “Some findings we report were driven by small subsets of LC patients, which is consistent with the notion of LC being a very heterogeneous disease” (Lines 535-537).

Reviewer #3

(Remarks to the Author)

In their manuscript Kailin Lin et al. investigate changes in immunological parameters in Long Covid patients by using CyTOF, RNAseq and Olink. A small cohort of patients has been investigated and compared to non-Long covid patients. The patient cohort did not receive any vaccination prior to onset of Long Covid and was investigated at a late time point (8 months). The authors conclude that several immunological parameters are different on the Long Covid

patients as compared to the control group. In particular, they note some changes in migration surface marker expression, exhaustion phenotype, antibody levels. This should indicate a breakdown on proper crosstalk of the humoral and cellular arms of adaptive immunity. Overall, the study is very descriptive. The conclusions are extrapolated from analysis of marker expressions levels but lack functional and mechanistic validation.

Due to its poorly understood etiology, the nature of Long COVID (LC) studies in humans is descriptive. Importantly, we feel that these types of comprehensive, multi-omics approaches on well-characterized LC patients and matched controls is the first, crucial step to increase our understanding of the disease and to develop potential treatment options. To date, such studies are rare, and none have focused on T cell responses in a way that we have in this paper. In the limitations section, we have now directly acknowledged the descriptive nature of our study while at the same time acknowledging the importance of such “omics”-based studies for advancing our understanding of Long COVID: “We note that our study is for the most part descriptive, but as for any such new and poorly-understood disease for which currently there is no good animal model, such in-depth “omics”-based characterization of a well-annotated cohort is the critical first step for better understanding the condition’s etiology and mechanistic underpinnings” (Lines 543-546). We also note that in the revised manuscript, we have added FACS and scRNAseq validation data, in response to comments by the other reviewers.

The patient cohorts are very small given that the authors inform on 10% Long Covid incidence following a SARS-Cov2 infection.

We have previously shown (PMID 35106317) that many individuals demonstrate LC symptoms that resolve over 4-8 months, and other people experience new symptoms that emerge late after SARS-CoV-2 infection and for whom it is unclear if these symptoms are directly attributable to SARS-CoV-2. Reinfections and vaccinations have made immunologic studies of LC even more complicated to conduct in 2023, without well-annotated archived specimens. All of these factors can lead to problematic confounding of the results. Because of these issues, we leveraged the fact that LIINC was one of the first prospective studies of LC in the world. The study began collecting clinical data and biospecimens prospectively starting in April 2020, before LC had even been identified as a clinical syndrome. The study also began before any reinfections were documented and before vaccines became available in January 2021.

In our analysis, we evaluated a highly selected cohort of people with and without LC. We studied people on whom we had available longitudinal clinical data (up to 8 months) after a single, first SARS-CoV-2 infection, and with biospecimens collected prior to receipt of any SARS-COV-2 vaccine. We further restricted the cohort to individuals who demonstrated consistency in meeting the case definition of LC or complete recovery over the course of the post-acute period, from 4 to 8 months post-COVID. The nature of this selection limited the size of the cohort, but in doing so eliminated most of the confounders that make LC immunology research so challenging in 2023 (variable reinfections, vaccinations, and treatments, as well as lack of prospectively collected longitudinal data). So although the total study sample is relatively small, we believe that the rigor with which the participants were characterized and their consistency in meeting the case definition longitudinally is a major strength of this analysis,

increasing the likelihood that our biological observations are clinically meaningful. It would be very difficult to conduct a study that adequately accounts for all of the potential confounders without such a cohort. In the revised manuscript, we have highlighted some of the points above justifying the size of the cohort (Lines 529-533). We have additionally created a new figure panel (Fig. S1A) to highlight the lengths we had gone through to stringently identify patients with vs. without LC. This new panel illustrates the heterogeneity of LC symptoms, and the types of samples that were chosen for our study:

Legend for above figure panel (Fig. S1A in revised manuscript): Strategy of biospecimen selection for study. Individuals demonstrate variable recovery following COVID-19, with many experiencing symptom resolution within 4 months (typically <30 days) in a manner that is sustained (solid blue line). However, some experience symptoms that persist (solid red line). Symptoms can also newly develop (dotted red line) or diminish (dotted blue line) over time. Per WHO definition, symptoms persisting beyond 3 months post-infection are defined as LC. In order to minimize the effects of between-individual heterogeneity, we selected individuals demonstrating a consistent post-acute phenotype (LC or Non-LC) at two timepoints timed 4 and 8 months following initial SARS-CoV-2 infection, and performed biologic measurements on samples from the 8-month timepoint. All samples were collected and cryopreserved prior to the individuals ever receiving a SARS-CoV-2 vaccine or experiencing a clinically detected reinfection.

Although Long Covid clearly is a very important and highly prevalent clinical manifestation and although there is a tremendous interest in the scientific community and general population to understand this pathology and to arrive at therapeutic solutions, the findings outlined in this manuscript remain vague and if they do not reveal robust novel diagnostic markers or disease activity parameters, nor do they provide specific information on how to therapeutically address this disease.

In the Discussion of our paper, we had discussed potential therapeutic opportunities based on our data. These included use of antivirals, potentially in conjunction with checkpoint inhibition as well as use of CXCR4 and other chemokine receptor inhibitors. Interestingly, an independent study published after our initial submission also suggested chemokine/chemokine receptor signaling as a therapeutic target for LC (PMID 36879067). In the revision, we have better highlighted the link between our data and therapeutic opportunities, and also cited the new chemokine/chemokine receptor signaling study that supports and complements with our results (Lines 450-452, 477-479).

Several issues remain to be clarified.

Fig.S1B. The authors claim that individuals with LC were more likely to have been hospitalized during the acute phase of COVID-19. This conclusion is based on 7 patients having been hospitalized as compared to 20 in the LC group, versus 2 hospitalized compared to 14 in the non-LC group. These patient numbers are too low to arrive at this conclusion.

Although it is generally accepted that severity of acute infection can affect the likelihood of developing LC, we have removed this statement since we agree that it is based on small patient numbers and this was not an epidemiologic assessment of risk factors of LC. Of note, this was a minor comment in our study and did not affect any of our overall conclusions.

Fig.S1B. The authors claim that the number of LC symptoms was increasing between month 4 and month 8. The data shows, however, a decrease in symptom counts in many individuals and a few (3) strong outliers showing an increase in symptom counts. There is no detailed information on the exact statistical test available (matched pairs?).

We believe the reviewer was referring to Fig. S1C not Fig. S1B in the original submission (and is now Fig. S1D in the revision). We had indeed used a paired sample t-test in this analysis, and this has now been clarified in the revised figure legend. With regards to our stating that there was an increase in symptom count, this was based on observation that there was a statistically significant increase in symptom count in the LC patients, from month 4 to month 8. We had chosen patients that had symptoms well-curated at multiple timepoints, and our main point here was that the symptoms were maintained (did not go down to zero at any timepoint). In response to the reviewer's comment, we have now instead stated that symptom counts in our long COVID participants did not decrease over time.

Fig.1. The figure illustrates clearly the experimental approach. Additional information on timepoints could be added to highlight the late analysis timepoint. It is unusual to commit a whole figure for a graphical outline for the study design. This figure could be added to the subsequent data figure or to the supplementary figures.

In the revision, we have added a component to our schematic figure illustrating that was had analyzed the late 8-month timepoint. We favor starting the paper with the schematic as a main figure, but would be fine moving it to supplement if the Editor/Reviewer feels that is important. Of note, we now also add to the figure the additional assays added to the revised manuscript

(scRNAseq and FACS validation studies).

Fig. 2. A gating strategy could be placed in the supplementary figures since it does not demonstrate any results. The “FACS” (Cytof) plots lack a scale (axes). It would be helpful to get information on whether percentages or MFI were provided. It is surprising that so few protein markers have been assessed/evaluated given the technical potential of CyTOF to measure many more parameters reported to be of relevance in COVID-19 or immune perturbations in general.

We have moved the gating strategy to the supplement (Fig. S2A and S2B), as requested. Also, we have now shown all axes scales, and provided MSI (in CyTOF, we measure MSI, not MFI which is for FACS) info (the new Fig. S6-S9), as requested. Importantly, we note that it was not the case that “so few markers have been assessed/evaluated given the technical potential of CyTOF...” We in fact assessed had every marker extensively, and when looking at each marker individually, we did not observe any significant differences between LC and control patients. We had opted to not show these negative data, but agree with the reviewer that it is important to show and hence now present them in their entirety in Figs. S6-S9. Of note, it is perhaps not surprising that individual markers’ MSI were not significantly different, given the complexity of LC. It was with the integrative analyses which led to the identification of significant differences which we focused on in our study.

Fig.S3D. The authors claim that SARS.CO2 specific CD4+ T cells producing IL-6 may be specifically associated with a subset of individuals with LC. Given that IL-6 is not a T helper cell cytokine it would be recommended to assess IL-6 production also by ELISA and to ensure the purity of the CD4+ T cells (since also monocytes and other human APCs express CD4).

Importantly, our IL6 data was generated from an intracellular cytokine staining assay, which reads out IL6 production at the single-cell level (i.e. we are detecting IL6 from the actual cells producing it). We had pre-gated on CD4+ T cells in our analyses, so there would be no monocytes or other human APCs contaminating our datasets. This is in fact one advantage of the intracellular cytokine staining assay over ELISA since no matter how pure a population of cells is in an ELISA, one could have a small population of contaminating cells producing the IL6. We have now clarified that the results in Fig. S3D were pre-gated on live, singlet CD4+ T cells (Lines 1471-1472) and that intracellular cytokine staining is a single-cell assay (Lines 145-147, 774-778). We have additionally cited prior studies demonstrating IL6 production by “atypical” cells, including T cells and B cells (Lines 484-486).

The extreme low frequency of IL-6+ cells (low MFI) could also represent background staining. The statement that this finding could be specific for LC should therefore be deemphasized.

We have de-emphasized this finding as requested, by deleting our original statement “These results suggest that, although rare, SARS-CoV-2-specific CD4+ T cells producing IL6 may be specifically associated with a subset of individuals with LC.” We would like to point out, however, we don’t believe what we observed to be background staining, since it was not observed in cells from our control group.

Fig. S6-7. The clustering analysis did not reveal any difference between the LC and control group. However, when the aspect of sex difference was considered, a few clusters revealed differences. This sex stratification is of general interest and an important aspect to be considered in clinical studies. Overall, the total patient numbers, however, seem too low to arrive at such a strong conclusion. It is also not clear to which extent the effects were driven by individual patients since too little information is provided on data analysis (concatenation?). The authors claim that profound changes in classical subset distribution have been found. However, looking at the statistical differences and the wide interindividual distribution of the tested parameters, the changes are quite subtle and would require retesting in large cohorts.

Our findings were not driven by individual patients, and we have now clarified this by adding a statement to our Methods section that our estimated log odds ratio data corresponded to the average over all participants, as summarized via the following statement: "The estimated log odds ratio represents the change (due to LC status) in the average over all participants of a given gender in the log odds of cluster membership". Furthermore, for the associations displayed in Figures S6 and S7, we have now reported multiple testing corrected FDR values in addition to the raw p-values. As in part reflected by the relatively high FDR values, the reviewer is correct in pointing out that the effects are subtle given the underlying variability of the cluster membership across the LC and the non-LC participants. Therefore, we fully agree that these findings will need to be validated in larger cohorts, and have now added this to the limitations section (Lines 535-537).

In addition, we have regenerated the violin plots (panels C and D in the original Figures S6 and S7, in revision now Figure S11 and S12, respectively) reflecting marker levels, to no longer reflect data concatenated from all donors, but rather reflecting the mean levels across cells for each participant via bar graphs. These plots that now compare the distribution of the highlighted markers at the participant level demonstrate the reproducibility (across participants) of the identified markers for the given clusters.

Fig.5. It is not so clear why different analysis methods were used for different parameters (Fig. S6-7 vs. Fig. 4).

The original Figures S6 and S7 (now Figures S11 and S12, respectively), reflect the result of clustering analyses, where we used the estimated marginal means applied to the Generalized Linear Mixed Effects Model (GLMM) to obtain p values. By contrast, Figures 4 and 5 used the more classical approach of manual gating to demonstrate significant differences in specific subsets of cells expressing the defined markers. As we now clarify in the Results section, the analogous GLMM clustering approach was not suitable for characterization of the SARS-CoV-2-specific T cells because significant differences were not observed in cluster distribution, likely due to more limited numbers of SARS-CoV-2-specific T cells relative to total T cells.

In T cells, CXCR4 is known to be constitutively expressed. The authors should comment on why they chose to assess the differential expression of this marker in T cells.

Although CXCR4 expression is quite ubiquitous among T cells, the absolute level of its expression varies between different subsets. This is further supported by our new FACS-based validation data (Fig. S13). CXCR4 was included in our panel design as it has previously been associated with severe/fatal COVID-19 (PMID 34260965). We assessed in depth differential expression of this marker on subsets of T cells because we had noticed that its expression levels were higher among SARS-CoV-2-specific CD4+ T cells from LC patients as compared to control patients (Fig. 4B).

It is also recommended to explain the choice of markers for analysis. The chemokine receptors, which were chosen are not classic tissue-homing receptors, as claimed. To serve the purpose to investigate tissue-homing, chemokine receptors such as CCR9 (gut), CCR8 (lung) or the marker CLA for skin and others are recommended.

CXCR4, CXCR5, CCR6, and CCR7 were the chemokine receptors incorporated into our CyTOF panel. CXCR4 was chosen because of its prior implication in severe COVID-19 (PMID 34260965), CXCR5 because of its ability to help identify T follicular helper cells, CCR6 because it is a canonical gut-homing receptor and is also commonly expressed on Th17 cells, and CCR7 because of its ability to help identify T central memory cells. Furthermore, these were antigens we had previously validated in prior CyTOF studies. CCR9, CCR8, and CLA are not antigens we have validated in our CyTOF protocol, although they would be of interest to examine in future studies given the results we obtained with the chemokine receptors in our study. In general, for CyTOF panel design, we need to be selective as we are limited to ~40 markers, and as this originally was not a study purely focused on chemokine receptor expression, we had chosen ones antigens that were validated and served multiple purposes. We have now clarified this, and added a section on marker selection (Lines 749-755).

The markers PD-1 and CTLA-4 are known to be upregulated upon T cell activation as well as exhaustion. It is therefore recommended to phrase the conclusion that SARS-CoV2 specific CD8+ T cells in LC are exhausted with more caution.

We have now made sure we never state that the SARS-CoV-2-specific CD8+ T cells are exhausted. Instead, we have stated that they “exhibit phenotypic features of exhaustion, as reflected by co-expression of PD1 and CTLA4” (Lines 281-282) and their “co-expressing the exhaustion markers PD1 and CTLA4” (Lines 442). We have further added additional data that canonical acute activation markers (CD38, HLADR, Ki67), unlike PD1 and CTLA4, are not expressed at significantly elevated levels on these cells (Lines 209-215, Fig. S10).

Additionally, the whole study is limited to the investigation of blood immune cells. A higher frequency of certain T cell subsets in the blood could indicate a lower frequency in the tissues (mobilization out of tissue), where the actual pathology takes place.

We had discussed in the limitations section that our study was one limited to blood specimens, and how future studies should perform similar studies in people (Lines 432-437 in original manuscript). Such studies will of course be much more logistically challenging and be limited in terms of the numbers of participants that can be analyzed, but nonetheless will be an important

area of future research. We appreciate the reviewer's point that changes in frequencies of T cell subsets could reflect tissue migration, and have incorporated this into our revised manuscript (Lines 537-540).

Fig. 7. It would be helpful to provide a heatmap that groups the LC and control patients separately to have easier visual access to the differences between both groups. This would still allow to see the interindividual differences.

We have prepared the requested heatmaps, for both the RNAseq and Olink data, and present these data as the new Fig. S15, shown shown below:

The authors also discuss that the detection of SARS-Cov2 specific T cells 8 months after infection could be due to persistence of the virus. It would be very interesting to find out whether LC is indeed a smoldering ongoing infection. However, also uninfected healthy individuals have SARS-COV2 specific T cells as part of their natural TCR repertoire (found previously in PBMC obtained before the pandemic started). It would have also been helpful for the interpretation of the data if the authors would have used a non-SARS-Cov2 control peptide in addition to no peptide.

It is standard to not add peptide as a negative control in such intracellular cytokine staining assays. Without addition of exogenous peptides, endogenous peptides occupy the MHC class I and class II molecules in the surface of the cells. We are not clear what "control peptide" is referring to, as separate peptides would be needed for MHC class I and class II, and different peptides would be needed depending on the HLA makeup of the individuals. (If we just pick a random peptide, it won't bind the pockets of the MHC molecules). In any case, if one were to add a known non-SARS-CoV-2 peptide that is recognized by T cells in a particular specimen, that would identify a distinct population of antigen-specific T cells. This was something we had done comparing the phenotypes of bystander CMV-specific T cells to SARS-CoV-2-specific ones in the context of convalescence from mild COVID-19 (PMID 32839763). The effects of LC on bystander T cell properties is well beyond the scope of the current study, although it is something we are interested in pursuing further (we are initiating a study to examine the features of bystander T cells against EBV, CMV, and influenza in the context of COVID-19 and convalescence). We also note that adding control peptide epitopes won't identify pre-existing

(pre-pandemic) SARS-CoV-2-specific T cells, which we agree do exist. However, these cells are at extremely low frequencies relative to those elicited after SARS-CoV-2 infection and during COVID-19 convalescence, something we had previously published on (PMID 34260965).

Decision Letter, first revision:

1st Aug 2023

Dear Dr. Roan,

Thank you for your response to the referees' comments on your Article, "Long COVID manifests with T cell dysregulation, inflammation, and an uncoordinated adaptive immune response to SARS-CoV-2". Although we are interested in the possibility of publishing your study in Nature Immunology, the issues raised by the referees need to be addressed.

Please revise the manuscript as specified in your letter. We consider the manuscript is better suited as a Letter, so please revise accordingly. Letters have 5 main figures, 10 Extended Data figures and about 2500 words (please see instructions below). I would suggest the following:

- Please delete figure 1. Per our style, 'no data' figures are not allowed; schematics are not allowed and non-essential graphic elements are not allowed. Please revise it as a timeline graph that contains all necessary information and only uses text, lines and arrows. Please include it as a panel in Fig. 1
- Combine Fig. 2 and Fig. 3 in one figure (Fig. 1, along with the timeline above).
- Please remove the bulk RNA-seq data, as it has limited value (or just use it as confirmatory data for the overexpression of ORD7 and ALAS2; in that case, please move it to ED Figures). Instead, please provide an informative and thorough analysis of the scRNA-seq, especially in the T cell clusters and integrate that, as much as possible, with the CyTOF data. Please avoid driving the scRNA-seq analysis from the perspective of ORD7 and ALAS2. Instead, analyze it as a discovery data set and describe it accordingly in the text.
- If necessary, please move the data in Fig. 7f,g to ED figures to allow space for the scRNA-seq results.
- As style suggestions, please avoid using "surprisingly, interestingly, intriguingly, remarkably" and so on in the narrative.
- Please revise to address all referee points, as specified in your letter.

At resubmission, please include a "Response to referees" detailing, point-by-point, how you addressed each referee comment. If no action was taken to address a point, you must provide a compelling argument. This response will be sent back to the referees along with the revised manuscript.

Please include a revised version of any required reporting checklist. It will be available to referees to aid in their evaluation.

Reporting summary:

When submitting the revised version of your manuscript, please pay close attention to our

[href="https://www.nature.com/nature-portfolio/editorial-policies/image-integrity">Digital Image Integrity Guidelines.](https://www.nature.com/nature-portfolio/editorial-policies/image-integrity) and to the following points below:

[REDACTED]

We hope to receive your revised manuscript within 4 weeks. If you cannot send it within this time, please let us know. We will be happy to consider your revision so long as nothing similar has been accepted for publication at Nature Immunology or published elsewhere.

Nature Immunology is committed to improving transparency in authorship. As part of our efforts in this direction, we are now requesting that all authors identified as 'corresponding author' on published papers create and link their Open Researcher and Contributor Identifier (ORCID) with their account on the Manuscript Tracking System (MTS), prior to acceptance. ORCID helps the scientific community achieve unambiguous attribution of all scholarly contributions. You can create and link your ORCID from the home page of the MTS by clicking on 'Modify my Springer Nature account'. For more information please visit [please visit www.springernature.com/orcid](http://www.springernature.com/orcid).

Sincerely,

Ioana Visan, Ph.D.
Senior Editor
Nature Immunology

Tel: 212-726-9207
Fax: 212-696-9752
www.nature.com/ni

Reviewers' Comments:

Reviewer #1:

Remarks to the Author:

In their revision, Yin et al. have expended considerable effort to address the multiplicity of concerns raised by the three reviewers and notably provided new data including confirmatory FACS experiments, scRNAseq evaluation of a subset of study participants as well as adjusted and expanded data analyses resulting in a substantial increase of supplementary figures. In regard to my critique points, the revision is for the most part satisfactory. Nevertheless, several specific issues may well remain a matter of debate (I am listing a few examples below) and I will leave it to the authors' discretion to address them or not.

I certainly agree with the shortcomings of AIM quantification for specific T cell identification but while CD40L surface detection (after stimulation in presence of CD40L antibodies akin to the CD107 degranulation assay) may qualify as an AIM readout, its simple intracellular detection after 6h peptide stimulation constitutes one of the best markers for virus-specific CD4+T cell detection and quantification - including SARS-CoV-2-specific CD4 T cells - due to the comparatively large fraction of specific CD4+T cells capable of rapidly synthesizing this TNFSF member (the cited author reference PMID 34260965 does not appear to employ CD40L detection)

the CD107 degranulation assay is indeed compatible with mass cytometry [e.g., PMID 27177653]; the fact that the authors did not intend to suggest that their intracellular CD107a detection as a functional readout is appreciated but what exactly is to be concluded from their intracellular CD107 data (why was it analyzed in the first place given, as the authors point out, the importance to carefully design staining panels limited to some 40 targets)?

the precise gating for Tfh cells (fig.s4a) appears to a bit "too generous" especially considering the otherwise convincing control stains for PD-1 and CXCR5 shown in fig.s18a (a more stringent gating would capture considerably fewer Tfh and therefore would affect the downstream calculations and statistics). similar considerations also apply to the Treg and Tscm gating in fig.s4a (and in particular comparison to fig.s18 control stains)

unclear what exactly is meant in the reply by "Of note, the perforin staining in the original Fig. S2 did not look robust because SARS-CoV-2-specific T cells did not specifically induce large amounts of perforin in response to cognate peptide stimulation." perforin expression is not induced by 6h peptide stimulation; if anything, degranulation may somewhat deplete perforin stores.

CD62L expression after recovery from cryostorage: my question was about the value of these analyses. that CD62L is re-expressed in o/n cultures of thawed PBMC is well known, as is the fact that expression levels are a function of culture conditions and in particular duration, and the authors have now amended the methods section accordingly (note, however, that the "classic methods paper" cited in the reply and as ref. 68 describes CD62L loss after thawing and not its recovery after culture). in contrast, the CD62L expression data now shown for virus-specific T cells in figs.s8 and s9 needs to be interpreted more carefully: following peptide stimulation, CD62L is rapidly shed so its absence on these cells is not reflective of their "after o/n culture and recovery" phenotype.

other issues: Ki67 gating by itself is somewhat problematic and cellular proliferation should have been assessed in conjunction with simple idu addition to the overnight sample culture; CD95 staining lacks

convincing controls; IL-4 detection after PMA/ionomycin is somewhat predictably not possible; a better choice of "exhaustion-focused" markers would have been helpful; FoxP3 detection by cytof is admittedly difficult but can be done somewhat better than shown in fig.s18 and could have been complemented with detection of other transcription factors; showing only msi values in figs.s6-s9 is quickly done yet doesn't leverage the richness of the data that includes the opportunity to define relative fractions of marker-expressing subsets in as much as that has not been done for selected markers in other figures etc.

Reviewer #2:

Remarks to the Author:

The authors have addressed my major comments. In its current version, the study represents a step towards understanding the immunological aspects of long COVID.

I would like to clarify two points;

1- In their response to point 1 in Data & methodology; my main concern was not using bulk RNAseq as a technique per se, but rather performing bulk RNAseq on "total PBMCs" that is composed of so many heterogeneous cell type and subsets. When performed on a well defined cell population with a degree of homogeneity bulk RNAseq is highly informative.

2- When discussing the point about whether the statistical significance was driven by a few patients: I cannot understand the authors' comment about "... if the editor and reviewer can suggest a good way to report on data where we selectively remove datapoints ...", as I do not believe such a thing exists. It is not acceptable under any circumstances to remove any data point "manually". Outliers can only be determined based on unbiased statistical algorithms, and it has to be clearly mentioned in the text, figure legend, Materials and Methods ... etc. I cannot find a single word in my review that mentions the word "outliers" or anything about removing datapoints. I think it was sufficient to only mention the first part of their response (highlighting that at least 6, 9, or more LC patients were higher for these parameters than the Non-LC and that the statistical significance is not driven by 1-2 patients).

Letter

A Letter discusses an important, novel research result, but is less substantial than an Article.

Format

Introductory paragraph (not abstract) up to 150 words, summarizing the background, rationale, main results and implications. This paragraph should be referenced, and should be considered part of the main text.

Main text – up to 2,500 words, excluding the introductory paragraph, online Methods, references and figure legends.

References – as a guideline, we typically recommend up to 30.

Display items – 5 items (figures and/or tables).

Letters are not divided by headings, except for the online Methods headings.

Letters include received/accepted dates.

Letters may be accompanied by supplementary information.

Letters are peer reviewed.

Author Rebuttal, first revision:

Dear Dr. Visan,

Thank you for the opportunity to delineate how we would rectify the remaining concerns brought forth by the reviewers. We are pleased that Reviewer 1 found our revision satisfactory, and that Reviewer 2 felt that we had addressed his/her major comments. As requested, we provide below a point-by-point response to the reviewers (in blue). (Please note line numbers refer to the “clean” version of the manuscript.) We would like to point out to the Reviewers that additional edits have been made in response to editorial requests to shorten the manuscript to a “Letter” format, which in addition to shortening the text, entailed combining the previous Figures 2 and 3 into one figure (now Figure 1) and consolidating supplementary figures to fit within the limit.

Reviewer #1

(Remarks to the Author)

In their revision, Yin et al. have expended considerable effort to address the multiplicity of concerns raised by the three reviewers and notably provided new data including confirmatory FACS experiments, scRNAseq evaluation of a subset of study participants as well as adjusted and expanded data analyses resulting in a substantial increase of supplementary figures. In regard to my critique points, the revision is for the most part satisfactory. Nevertheless, several specific issues may well remain a matter of debate (I am listing a few examples below).

We thank the Reviewer for recognizing our efforts, and finding our revision satisfactory. We appreciate the additional comments and have addressed them in detail below.

I certainly agree with the shortcomings of AIM quantification for specific T cell identification but while CD40L surface detection (after stimulation in presence of CD40L antibodies akin to the CD107 degranulation assay) may qualify as an AIM readout, its simple intracellular detection after 6h peptide stimulation constitutes one of the best markers for virus-specific CD4+T cell detection and quantification - including SARS-CoV-2-specific CD4 T cells - due to the comparatively large fraction of specific CD4+T cells capable of rapidly synthesizing this TNFSF member (the cited author reference PMID 34260965 does not appear to employ CD40L detection)

We thank the Reviewer for pointing out that CD40L, unlike other AIM markers, can be used in conjunction with a 6-hour intracellular cytokine staining (ICS) assay. As this study had set out to identify SARS-CoV-2-specific T cells via ICS and not AIM, we did not include CD40L. We have now stated in the Limitations section that CD40L was not included in our panel (Line 240).

the CD107 degranulation assay is indeed compatible with mass cytometry [e.g., PMID 27177653]; the fact that the authors did not intend to suggest that their intracellular CD107a detection as a functional readout is appreciated but what exactly is to be concluded from their

intracellular CD107 data (why was it analyzed in the first place given, as the authors point out, the importance to carefully design staining panels limited to some 40 targets)?

We thank the Reviewer for pointing out the previous CD107a degranulation study performed via mass cytometry, but would like to note that this study was looking at PMA/ionomycin-stimulated NK cells, and not peptide-stimulated as needed for our studies T cells. Furthermore, NK cell-based approach used a two-step fluorescent CD107a-APC (FACS antibody) followed by a secondary metal-conjugated antibody. We are not aware of a study to date validating use of a metal-conjugated CD107a antibody in a CyTOF-based degranulation assay for characterizing antigen-specific T cells. In any case, as stated in our original revision, we did not use CD107a as a degranulation marker, and did not draw conclusions based on this marker. We agree that in that regard it's not clear what value the CD107a data provide, and therefore we have removed all CD107a data from our manuscript. We now describe our panel as a 38-parameter panel instead of the original 39-parameter panel that included CD107a.

the precise gating for Tfh cells (fig.s4a) appears to a bit "too generous" especially considering the otherwise convincing control stains for PD-1 and CXCR5 shown in fig.s18a (a more stringent gating would capture considerably fewer Tfh and therefore would affect the downstream calculations and statistics). similar considerations also apply to the Treg and Tscm gating in fig.s4a (and in particular comparison to fig.s18 control stains)

As we mentioned in our prior rebuttal, we had implemented a Tfh gate to match our prior COVID-19 studies, so that the proportions of Tfh in PBMCs can be directly compared between studies. We agree that this gate doesn't directly match the control stains of PD1 and CXCR5 in the original Fig. S18A, which were implemented on tonsillar T cells which, unlike PBMCs, harbor distinct populations of cells expressing much higher levels of these two Tfh markers. As expected, when we use the Tfh gate shown below for the distinct population of PD1^{high}CXCR5^{high} cells:

we obtain lower frequencies of Tfh cells in our PBMC datasets as PBMC-derived T cells overall express lower levels of PD1 and CXCR5 compared to their tonsil counterparts. Importantly, however, even using this tonsil-driven gate, our overall conclusion that Tfh are higher in LC as compared to non-LC remains the same; the statistical change however is that while our original observation was that Tfh was significantly higher in LC as compared to non-LC as determined

by a p-value of 0.0373), the data now with the stringent gate only trends at $p=0.066$ (the old Fig. 3A, which is now Fig. 1H after request from Editor to consolidate figures).

In this second revision, we have opted to show both Tfh gates. Our original one we now refer to as the "pTfh" (for peripheral Tfh) gate, to allow for more generous PD1 and CXCR5 gates when analyzing PBMC specimens, an approach that has been implemented by us and others. We felt it valuable to keep this gate to enable direct comparisons with our prior COVID-19 studies reporting on percentages of Tfh (PMIDs 34636722, 34260965, 32839763). In parallel, we report data based on the PD1^{high}CXCR5^{high} gating strategy from tonsil cells, and refer to these cells as the "Tfh" population.

We have also made our Tscm gate tighter. This did not change our original finding that Tscm proportions were not different between LC and non-LC individuals (they are still not different). With regards to our Treg population, it was not clear how we would change the gate, as the population we had gated on was already rather conservative, but more importantly we had validated this gate by demonstrating preferential expression of Foxp3 in this population of cells (see below). Furthermore, this gate matches the Treg gates we had implemented in our previous COVID-19 CyTOF studies (PMIDs 34636722, 34260965, 32839763) to better enable comparison of Treg proportions between studies. We have therefore opted to keep our original Treg gates.

unclear what exactly is meant in the reply by "Of note, the perforin staining in the original Fig. S2 did not look robust because SARS-CoV-2-specific T cells did not specifically induce large amounts of perforin in response to cognate peptide stimulation." perforin expression is not induced by 6h peptide stimulation; if anything, degranulation may somewhat deplete perforin stores.

We apologize for the confusion. With this sentence, we were responding to what we interpreted as the original critique that our perforin antibody may not be working as we did not show any positive control of perforin staining. Therefore, in our first revision, we showed a positive control for perforin staining in Fig. S18 (now Fig. S10 after consolidating figures as requested by the Editor). We agree with the Reviewer that perforin is not induced by a 6h peptide stimulation, and that is indeed what we had observed.

CD62L expression after recovery from cryostorage: my question was about the value of these analyses. that CD62L is re-expressed in o/n cultures of thawed PBMC is well known, as is the fact that expression levels are a function of culture conditions and in particular duration, and the authors have now amended the methods section accordingly (note, however, that the "classic methods paper" cited in the reply and as ref. 68 describes CD62L loss after thawing and not its recovery after culture). in contrast, the CD62L expression data now shown for virus-specific T cells in figs.s8 and s9 needs to be interpreted more carefully: following peptide stimulation, CD62L is rapidly shed so its absence on these cells is not reflective of their "after o/n culture and recovery" phenotype.

We thank the Reviewer for recognizing our prior revisions, and agree with him/her that among all the MSI data for SARS-CoV-2-specific T cells shown in our previous Fig. S8 and S9, the CD62L data should be interpreted with caution given that CD62L is rapidly shed after peptide stimulation. At the request of the Editor to consolidate figures to fit the "Letter" format, the former Fig. S8 and S9 are now consolidated into one single Fig. S5. We now state in the figure legend of the new Fig. S5: "Of note, the low levels of CD62L may result from peptide stimulation which can downmodulate expression of this surface receptor," to caution on interpretation of the CD62L data in this figure. Importantly, all of the paper's analyses/discussions of CD62L were focused on total (not peptide-stimulated) T cells, so the fact that the CD62L signal is difficult to interpret among our peptide-stimulated cells doesn't affect any of our conclusions.

other issues: Ki67 gating by itself is somewhat problematic and cellular proliferation should have been assessed in conjunction with simple idu addition to the overnight sample culture;

We agree with this new point that deoxyuridine (IdU) in conjunction with Ki67 better identifies cycling cells, although Ki67 has also commonly been used on its own both in FACS and mass cytometry. We note that Idu needs to be read out on CyTOF channel 127, which we currently use to monitor iodine contamination, which is important for any clinical specimen analyzed by CyTOF. In the future, pending sufficient specimen availability, samples could be pre-screened for the ability to be subjected to idu-based experiments; however, that is beyond the scope of this study. Nevertheless, as we agree that despite technical and logistical challenges, having Idu data would have been valuable. We now cite the 2021 manuscript describing Idu as tool for monitoring proliferation by CyTOF (PMID 34747407) and acknowledge our lack of Idu experiments as a limitation of our study (Line 240).

CD95 staining lacks convincing controls;

We would like to point out that although the CD95 antibody does not stain robustly, the level of staining we observed is similar to what we saw in our recently published CyTOF-based COVID-19 studies, which had used this same antibody clone (PMIDs 34636722, 34260965, 32839763). We would also point out that the extent of CD95 staining is somewhat irrelevant in our study as none of our conclusions about differences between those with vs. without Long COVID involved CD95.

IL-4 detection after PMA/ionomycin is somewhat predictably not possible;

We agree with this statement, and indeed the data we depict in our previous Fig. S18 (now Fig. S10 after figure consolidations to fit "Letter format") are consistent with this. We note that our metal-conjugated IL-4 antibody is a commercially purchased one, and has been validated by the vendor Standard BioTools. We would also point out that none of our conclusions about differences between those with vs. without Long COVID involved IL-4.

a better choice of "exhaustion-focused" markers would have been helpful;

We assume the Reviewer is referring to his/her original comment about the lack of TOX on our CyTOF panel. As we had detailed in our original rebuttal, we had included 3 canonical exhaustion markers (PD1, CTLA4, and TIGIT) in our study, but unfortunately did not have TOX. This was due to limited room in our CyTOF panel, but also because TOX is not a validated CyTOF reagent commercially available from Standard BioTools (the only company that produces metal-conjugated antibodies for CyTOF). We fully recognize the importance of TOX in T cell exhaustion, and therefore have now added to our Limitations section (Line 239) that our assessment of T cell exhaustion did not include TOX, and have cited a 2019 key study by Wherry and colleagues (PMID 31207603) highlighting the importance of TOX for driving CD8+ T cell exhaustion.

FoxP3 detection by cytof is admittedly difficult but can be done somewhat better than shown in fig.s18 and could have been complemented with detection of other transcription factors;

We agree that Foxp3 is difficult to detect robustly as the Reviewer acknowledges. We have tried hard to obtain stronger staining by CyTOF but have been unable to, which is why we resorted to using Foxp3 to validate our identification of Tregs as CD45RA-CD45RO+CD127-CD25+ T cells, rather than as a gate, for our analyses comparing Long COVID to non-Long COVID patients (Fig. S10 in latest version). We have now added a statement that because Foxp3 staining was weak, we used the CD45RA-CD45RO+CD127-CD25+ gating strategy to identify Tregs (a commonly used approach) (Lines 472-476).

showing only msi values in figs.s6-s9 is quickly done yet doesn't leverage the richness of the data that includes the opportunity to define relative fractions of marker-expressing subsets in as much as that has not been done for selected markers in other figures etc.

We would respectfully like to point out that we had performed much more than the previous Fig. S6-S9 (now consolidated to Fig. S4 and S5) to leverage the richness of our datasets. In addition to these four supplementary figures, we also 1) performed unbiased clustering analyses leveraging all markers in our panel (which directly leverages the richness of the high-dimensional data) (the previous Fig. S11 and S12, now consolidated as Fig. S7), 2) examined cytokine effector functions and polyfunctionalities (the previous Fig. 2, now Fig. 1 F&G), 3) examined major subsets of CD4+ and CD8+ T cells (the previous Fig. 3, S5, now Fig. 1H&I and S3), 4) performed a formal analysis of activation markers as requested (the previous Fig. S10, now Fig. S6), 5) performed focused analyses on chemokine receptors (the previous Fig. 4, now Fig. 2), 6) performed focused analyses on exhaustion markers in our panel (the previous Fig. 5, now Fig. 3). We feel this is one of the most comprehensive analyses of total and SARS-CoV-2-specific T cells in the context of COVID-19, and that we have leveraged the richness of our datasets.

Reviewer #2

(Remarks to the Author)

The authors have addressed my major comments. In its current version, the study represents a step towards understanding the immunological aspects of long COVID.

We are pleased that the Reviewer is satisfied with our revisions, and thank him/her for clarifying his/her original points below, which we respond to below.

I would like to clarify two points;

1- In their response to point 1 in Data & methodology; my main concern was not using bulk RNAseq as a technique per se, but rather performing bulk RNAseq on "total PBMCs" that is composed of so many heterogeneous cell type and subsets. When performed on a well defined cell population with a degree of homogeneity bulk RNAseq is highly informative.

We thank the Reviewer for clarifying this aspect. Regardless, the suggestion from this Reviewer to perform scRNAseq was a great one (and in the end provided more meaningful data than would have been gotten by multiple bulk RNAseq analyses on well-defined populations), and we feel that our revised manuscript now containing scRNAseq is significantly improved from our original submission.

In our revised manuscript, we show both the bulk RNAseq and scRNAseq data. The scRNAseq data is presented in the context of both the bulk data which was used to select samples of interest for further analyses by scRNAseq, as well as its own discovery dataset. The scRNAseq data are shown as three main figure panels (Fig. 5D, E, F), Figure S9, and 3 supplementary tables (Tables S5, S6, S7). The new analyses performed since the previous version of the manuscript (where we first presented the scRNAseq datasets) are presented in Fig. 5F. Also since the previous version, we have now generated the new Table S5 which shows using a less stringent cutoff differentially expressed genes (DEGs) between the non-LC vs. the two groups of LC individuals combined together. When combining the two groups of LC together, there are fewer DEGs as compared to when the individual LC groups are compared to non-LC (Tables S6 and S7).

2- When discussing the point about whether the statistical significance was driven by a few patients: I cannot understand the authors' comment about "... if the editor and reviewer can suggest a good way to report on data where we selectively remove datapoints ...", as I do not believe such a thing exists. It is not acceptable under any circumstances to remove any data point "manually". Outliers can only be determined based on unbiased statistical algorithms, and it has to be clearly mentioned in the text, figure legend, Materials and Methods ... etc. I cannot find a single word in my review that mentions the word "outliers" or anything about removing datapoints. I think it was sufficient to only mention the first part of their response (highlighting that at least 6, 9, or more LC patients were higher for these parameters than the Non-LC and that the statistical significance is not driven by 1-2 patients).

We absolutely agree with these points and we would never manually remove datapoints. We were trying to use as many ways as possible to emphasize the point that our data were not driven by 1-2 patients. We agree that it would have been sufficient to just mention the first part of our response. Should our manuscript be published in Nature Immunology in a manner where the first set of Response to Reviewers is made public, we would request that we limit our

response to this point to the first part of our original rebuttal, which the Reviewer states is sufficient of a response.

Decision Letter, second revision:

11th Sep 2023

Dear Dr. Roan,

The comments of the referees on your Letter, "Long COVID manifests with T cell dysregulation, inflammation, and an uncoordinated adaptive immune response to SARS-CoV-2" are now in. While the work is of potential interest, the issues raised need to be addressed.

Please revise the manuscript to address these issues. We agree with referee #2 that the scRNA-seq data is underutilized. As previously mentioned, and outlined by referee #2, we suggest to refine your analysis of the scRNA-seq dataset to harness its full potential for discovery and validation at a granular level. We suggest to de-emphasize the bulk RNA-seq results regarding OR7D2 and ALAS2, and their analysis in the scRNA-seq. At resubmission, please include a "Response to referees" detailing, point-by-point, how you addressed each referee comment. If no action was taken to address a point, you must provide a compelling argument. This response will be sent back to the referees along with the revised manuscript.

Please include a revised version of any required reporting checklist. It will be available to referees to aid in their evaluation.

Reporting summary:

[REDACTED]

We hope to receive your revised manuscript within three-four weeks. If you cannot send it within this time, please let us know. We will be happy to consider your revision so long as nothing similar has been accepted for publication at Nature Immunology or published elsewhere.

Nature Immunology is committed to improving transparency in authorship. As part of our efforts in this direction, we are now requesting that all authors identified as ‘corresponding author’ on published papers create and link their Open Researcher and Contributor Identifier (ORCID) with their account on the Manuscript Tracking System (MTS), prior to acceptance. ORCID helps the scientific community achieve unambiguous attribution of all scholarly contributions. You can create and link your ORCID from the home page of the MTS by clicking on ‘Modify my Springer Nature account’. For more information please visit www.springernature.com/orcid.

Sincerely,

Ioana Visan, Ph.D.
Senior Editor
Nature Immunology

Tel: 212-726-9207
Fax: 212-696-9752
www.nature.com/ni

Reviewers' Comments:

Reviewer #2:

Remarks to the Author:

The authors made a very good effort in addressing additional reviewer comments, as well as reformatting the manuscript as a letter.

I believe the scRNAseq they conducted has lots of potential.

The analyses presented in Figure 5G and 5H, as well as in Figure S9 do not harness the full potential of the granularity of data within scRNAseq that could add much more to the information gained by bulk RNAseq.

It could be used to confirm many of the CyTOF data, and even shed light on many markers that are either difficult to stain or were not included in their panel for T cells. In addition, it could shed light on other immune cells that were not directly discussed because they were not the main focus of the study, but would still impact T cell function in the different settings of LC versus non-LC.

Author Rebuttal, second revision:

Dear Dr. Visan,

We are pleased that Reviewers 1 and 3 have found our previous revisions satisfactory, and that Reviewer #2 has stated that we made a “very good effort in addressing additional reviewer comments, as well as reformatting the manuscript as a letter.” In this revision, we address the remaining request from Reviewer #2 to elaborate on the scRNAseq analysis. We have also, in response to your editorial request, de-emphasized the OR7D2 and ALAS2 results – specifically, the previous Fig. 5E has been moved to the supplement, and the previous Fig. S9E and S9F have been deleted). As requested, we provide below a point-by-point response to the Reviewer #2 comments (in blue).

Reviewer #2

(Remarks to the Author)

The authors made a very good effort in addressing additional reviewer comments, as well as reformatting the manuscript as a letter.

I believe the scRNAseq they conducted has lots of potential.

The analyses presented in Figure 5G and 5H, as well as in Figure S9 do not harness the full potential of the granularity of data within scRNAseq that could add much more to the information gained by bulk RNAseq.

It could be used to confirm many of the CyTOF data, and even shed light on many markers that

are either difficult to stain or were not included in their panel for T cells. In addition, it could shed light on other immune cells that were not directly discussed because they were not the main focus of the study, but would still impact T cell function in the different settings of LC versus non-LC.

We assume the Reviewer was referring to our previous Figure 5D/5E/5F instead of Figure 5G/5H, since the latter were Olink and not scRNAseq analyses. We have now added additional analyses, including validation data and cellular pathway analyses, and made room for these analyses by removing some of the prior OR7D2 and ALAS2 analyses (the former S9E and S9F) as requested by the Editor. This includes:

- 1) Figure 5F showing 3 volcano plots highlighting all significant DEGs when comparing LC vs. non-LC individuals, for clusters that harbored significant DEGs
- 2) Figure S9F, where we manually assessed for relative expression levels of CXCR4, CXCR5, and CCR6 among the CD4⁺ T cell clusters from the scRNAseq data. This revealed that these chemokine receptors were more highly expressed in the LC individuals, thereby directly confirming the prior findings from CyTOF (Fig. 2) and FACS (Fig. S8). (Note that the other major CyTOF finding of preferential expression of PD1 and CTLA4 on SARS-CoV-2-specific CD8⁺ T cells, but not total CD8⁺ T cells, cannot be confirmed by scRNAseq analyses as there is no way to know the specificity of the CD8⁺ T cells from the scRNAseq datasets, and the frequencies of these antigen-specific T cells are in any case too low to be captured by scRNAseq analysis.) The fact that preferential expression of these chemokine receptors in CD4⁺ T cells from LC as compared to non-LC individuals has been found using 3 independent assays (CyTOF, FACS, and scRNAseq) has now been highlighted in the discussion section.
- 3) Figure S9G showing volcano plot analyses of scRNAseq clusters between OR7D2^{high} LC vs. non-LC individuals, highlighting significant DEGs
- 4) Figure S9H showing volcano plot analyses of scRNAseq clusters between ALAS2^{high} LC vs. non-LC individuals, highlighting significant DEGs
- 5) Table S6 of Gene Ontology (GO) pathway analysis of cellular pathways significantly different between monocytes in LC vs. non-LC individuals
- 6) Table S7 of GO pathway analysis of cellular pathways significantly different between CD8⁺ T cells / CTLs in LC vs. non-LC individuals
- 7) Table S10 of GO pathway analysis of cellular pathways significantly different between all clusters in OR7D2^{high} LC vs. non-LC individuals
- 8) Table S11 of GO pathway analysis of cellular pathways significantly different between all clusters in ALAS2^{high} LC vs. non-LC individuals

Combined with the previous scRNAseq analyses we had done in the prior revision, in total we now have 10 figure panels (Fig. 5E, 5F, S9A, S9B, S9C, S9D, S9E, S9F, S9G, S9H) and 7 tables (Tables S5, S6, S7, S8, S9, S10, S11) relating to scRNAseq data analyses. We feel we have now harnessed the full potential of the granularity of the data within these datasets. If there are additional specific analyses that can be performed that we missed and that the Reviewer/Editor would like to see, we would be happy to perform them.

Decision Letter, third revision:

16th Oct 2023

Dear Dr. Roan,

Thank you for submitting your revised manuscript "Long COVID manifests with T cell dysregulation, inflammation, and an uncoordinated adaptive immune response to SARS-CoV-2" (NI-LE35392C). It has now been seen by the original referees and their comments are below. We are happy to inform you that if you revise your manuscript appropriately according to our editorial requirements, your manuscript should be publishable in Nature Immunology.

I will now pre-edit the current version of your paper. We will also perform detailed checks on your paper and will send you a checklist detailing our editorial and formatting requirements in about two weeks. Please do not upload the final materials and make any revisions until you receive this additional information from us.

In the meantime however, please deposit all omics and code data into public repositories so that the accession codes are readily available to be added in the revised manuscript. We cannot accept the paper without them. In addition, please check that the ORCID of all corresponding authors is linked to their Nature account, as this frequently causes delays at acceptance. Should you have any query or comments about ORCID, please do not hesitate to contact our editorial assistant at immunology@us.nature.com.

If you had not uploaded a Word file for the current version of the manuscript, we will need one before beginning the editing process; please email that to immunology@us.nature.com at your earliest convenience.

Thank you again for your interest in Nature Immunology. Please do not hesitate to contact me if you have any questions.

Sincerely,

Ioana Visan, Ph.D.
Senior Editor
Nature Immunology

Tel: 212-726-9207
Fax: 212-696-9752
www.nature.com/ni

Reviewer #2 (Remarks to the Author):

I would like to thank and congratulate the authors for their efforts to get the manuscript to its current version. I think it represents an excellent step forward in starting to understand Long-COVID.

I apologize for confusing the figure panels representing scRNAseq analysis in my comments to Version B of the manuscript.

Final Decision Letter:

Dear Dr. Roan,

I am delighted to accept your manuscript entitled "Long COVID manifests with T cell dysregulation, inflammation, and an uncoordinated adaptive immune response to SARS-CoV-2" for publication in an upcoming issue of Nature Immunology.

Over the next few weeks, your paper will be copyedited to ensure that it conforms to Nature Immunology style. Once your paper is typeset, you will receive an email with a link to choose the appropriate publishing options for your paper and our Author Services team will be in touch regarding any additional information that may be required.

Please note that *Nature Immunology* is a Transformative Journal (TJ). Authors may publish their research with us through the traditional subscription access route or make their paper immediately open access through payment of an article-processing charge (APC). Authors will not be required to make a final decision about access to their article until it has been accepted. [Find out more about Transformative Journals](https://www.springernature.com/gp/open-research/transformative-journals).

Authors may need to take specific actions to achieve [compliance](https://www.springernature.com/gp/open-research/funding/policy-compliance-faqs) with funder and institutional open access mandates. If your research is supported by a funder that requires immediate open access (e.g. according to [Plan S principles](https://www.springernature.com/gp/open-research/plan-s-compliance))

then you should select the gold OA route, and we will direct you to the compliant route where possible. For authors selecting the subscription publication route, the journal's standard licensing terms will need to be accepted, including [self-archiving policies](https://www.springernature.com/gp/open-research/policies/journal-policies). Those licensing terms will supersede any other terms that the author or any third party may assert apply to any version of the manuscript.

Your paper will be published online soon after we receive your corrections and will appear in print in the next available issue. Content is published online weekly on Mondays and Thursdays, and the embargo is set at 16:00 London time (GMT)/11:00 am US Eastern time (EST) on the day of publication. Now is the time to inform your Public Relations or Press Office about your paper, as they might be interested in promoting its publication. This will allow them time to prepare an accurate and satisfactory press release. Include your manuscript tracking number (NI-LE35392D) and the name of the journal, which they will need when they contact our office.

About one week before your paper is published online, we shall be distributing a press release to news organizations worldwide, which may very well include details of your work. We are happy for your institution or funding agency to prepare its own press release, but it must mention the embargo date and Nature Immunology. Our Press Office will contact you closer to the time of publication, but if you or your Press Office have any enquiries in the meantime, please contact press@nature.com.

Also, if you have any spectacular or outstanding figures or graphics associated with your manuscript - though not necessarily included with your submission - we'd be delighted to consider them as candidates for our cover. Simply send an electronic version (accompanied by a hard copy) to us with a possible cover caption enclosed.

If you have not already done so, we strongly recommend that you upload the step-by-step protocols used in this manuscript to the Protocol Exchange. Protocol Exchange is an open online resource that allows researchers to share their detailed experimental know-how. All uploaded protocols are made freely available, assigned DOIs for ease of citation and fully searchable through nature.com. Protocols can be linked to any publications in which they are used and will be linked to from your article. You can also establish a dedicated page to collect all your lab Protocols. By uploading your Protocols to Protocol Exchange, you are enabling researchers to more readily reproduce or adapt the methodology you use, as well as increasing the visibility of your protocols and papers. Upload your Protocols at

www.nature.com/protocolexchange/. Further information can be found at www.nature.com/protocolexchange/about .

Please note that we encourage the authors to self-archive their manuscript (the accepted version before copy editing) in their institutional repository, and in their funders' archives, six months after publication. Nature Portfolio recognizes the efforts of funding bodies to increase access of the research they fund, and strongly encourages authors to participate in such efforts. For information about our editorial policy, including license agreement and author copyright, please visit www.nature.com/ni/about/ed_policies/index.html

Sincerely,

Ioana Visan, Ph.D.
Senior Editor
Nature Immunology

Tel: 212-726-9207
Fax: 212-696-9752
www.nature.com/ni